# The influence of sample size and covariate distributions on neuroanatomical normative modeling

Camille Elleaume[1,2]*, Bruno Hebling Vieira[1], Dorothea L Floris[1], Nicolas Langer[1,2]

[1]Methods of Plasticity Research, Department of Psychology, University of Zürich, Zürich, Switzerland; [2]Neuroscience Center Zürich (ZNZ), Zürich, Switzerland

## eLife Assessment

This **important** manuscript evaluates how sample size and demographic balance of reference cohorts affect the reliability of normative models. The evidence supporting the conclusions is **convincing**. This work will be of interest to clinicians and scientists working with normative models.

**Abstract** Normative models are increasingly used to characterize individual-level brain deviations in neuroimaging studies, but their performance depends heavily on the reference sample used for training or adaptation. In this study, we systematically investigated how sample size and covariate composition of the reference cohort influence model fit, deviation estimates, and clinical readouts in Alzheimer's disease (AD). Using a discovery dataset (OASIS-3, $n$ = 1032), we trained models on healthy control (HC), subsamples ranging from 5 to 600 individuals, while varying age and sex distributions to simulate biases in reference populations. We further assessed the use of adaptive transfer learning by pre-training models on the UK Biobank ($n$ = 42,747) and adapting them to the clinical dataset applying the same subsampling strategies. We evaluated model performance on a fixed HC test set and quantified deviation score errors, outlier detection, and classification accuracy in both the HC test set and the AD cohort. The findings were replicated in an external validation sample (AIBL, $n$ = 463). Across all settings, model performance improved with increasing sample size, but demographic alignment of the covariates, particularly in age, was essential for reliable deviation estimates. Models trained directly within the dataset achieved stable fit with approximately 200 HCs, while adapted models reached comparable performance with as few as 50 individuals when pre-trained on large-scale data. These results show that robust individual-level modeling can be achieved using moderately sized but demographically matched cohorts, supporting broader application of normative modeling in aging and neurodegeneration research.

*For correspondence:
camille.elleaume@psychologie.
uzh.ch

Competing interest: The authors declare that no competing interests exist.

## Introduction

Normative modeling is a statistical framework for quantifying individual deviations from typical brain structure or function (*Marquand et al., 2016*). It estimates covariate-adjusted percentiles of a given measurement in a reference population. These percentiles serve as a basis for quantifying individual deviations from population norms, typically accounting for covariates such as age and sex. Unlike traditional case–control analyses that rely on group-level means, normative models generate subject-specific deviation scores that preserve inter-individual variability. This allows the characterization of brain atypicality at the individual level, without relying on the assumption that individuals within the same group share common patterns (*Marquand et al., 2016*; *Rutherford et al., 2022a*; *Rutherford et al., 2023*). Warped Bayesian Linear Regression (BLR) (*Fraza et al., 2021*) is a widely used normative

modeling approach that combines flexibility and scalability, making it particularly well-suited for large neuroimaging datasets (*Corrigan et al., 2024*; *Holz et al., 2023*; *Meijer et al., 2024*; *Rutherford et al., 2023*; *Savage et al., 2024*; *Verdi et al., 2024*). Long-term, these models can be envisioned as interpretable and quantitative neuroradiology tools to support clinical decision making (*Bozek et al., 2023*; *Goodkin et al., 2019*).

Among clinical applications of normative models, Alzheimer's disease (AD) represents a particularly relevant use case. AD is the most common cause of dementia and remains challenging to characterize due to its marked heterogeneity in clinical presentation and neuroanatomical patterns (*Duara and Barker, 2022*; *Lam et al., 2013*; *Rajan et al., 2021*). In clinical practice, structural imaging already plays a central role in diagnosis, and hippocampal volume is routinely evaluated in patients relative to expectations built from reference data (*Vernooij et al., 2019*). Normative modeling builds on this approach by enabling a more fine-grained characterization of neuroanatomical variability, supporting more precise and personalized assessments in AD. Recent applications to AD research have demonstrated its potential to reveal individualized patterns of cortical atrophy that correlate with cognitive performance and key biomarkers, including CSF $A\beta_{42}$ and phosphorylated tau (*Loreto et al., 2024*; *Verdi et al., 2021*; *Verdi et al., 2023*; *Verdi et al., 2024*).

However, the performance of normative models depends critically on the reference population used to fit them (*Bethlehem et al., 2022*; *Bozek et al., 2023*). The size of the training data, and the extent to which relevant covariates (e.g., age and sex) are adequately represented can substantially influence model fit and the accuracy of the resulting deviation scores. Large-scale samples have been recommended to accurately estimate outlying percentiles for clinical use (*Bozek et al., 2023*). However, collecting sufficiently large neuroimaging datasets is particularly challenging, as it often requires multi-site data aggregation and extensive efforts to harmonize differences in acquisition protocols, scanner hardware. To address the challenge of limited sample sizes, studies often employ adaptive transfer learning strategies that allow recalibrating pre-trained models on smaller samples (*Bayer et al., 2022*; *Gaiser et al., 2023*; *Kia et al., 2021*). But transfer learning applications face a similar challenge. While it offers a pragmatic solution, its success ultimately depends on the quality and representativeness of the available adaptation sample used for re-calibration. Whether models are directly trained within cohorts or adapted from large-scale references, both strategies raise concerns about robustness of clinical interpretations, particularly when the available data are limited, as is commonly observed in neuroimaging studies. Beyond sample size, careful consideration of covariates is critical, especially in applications to AD. The disease typically emerges in older adulthood and is more commonly diagnosed in women (*Ferretti et al., 2018*; *Riedel et al., 2016*), making accurate modeling of age and sex covariates especially important in this context. Despite growing adoption of normative modeling, systematic evaluations of how performance is affected by the composition and size of the training or adaptation sample remain limited.

To address this gap, we conducted a systematic investigation of how reference sample size and covariate composition affect normative model performance and clinical readouts. Using a single-site dataset (OASIS-3), we varied the size of the healthy control (HC) reference cohort (from 5 to 600 individuals) and manipulated age and sex distributions to simulate biases in reference populations. We replicated the approach in a two-site cohort (AIBL) and further assessed whether models pre-trained on a large external dataset (UK Biobank; *n* = 42,747) could be effectively adapted through transfer learning (*Rutherford et al., 2022b*), using the same subsampling strategies for the adaptation set. Adapted models were then applied to the same fixed test set, allowing direct comparison between models trained within each cohort and UKB-adapted models across all sampling conditions. Models' performances were evaluated using standard normative modeling evaluation metrics in an independent fixed HC test set. We assessed the error in deviation scores relative to those obtained using the full, theoretically optimal, reference sample in both the HC test set and the clinical sample diagnosed with AD. To assess clinical implications at the application level, we further quantified differences in outlier detection and classification performance when distinguishing AD from HC groups.

This work provides a systematic analysis of how reference sample characteristics influence model fit, deviation estimates, and their clinical interpretability, offering practical guidance for applying normative modeling in aging and neurodegeneration research and for maximizing the value of existing, deeply phenotyped cohorts when assembling reference datasets.

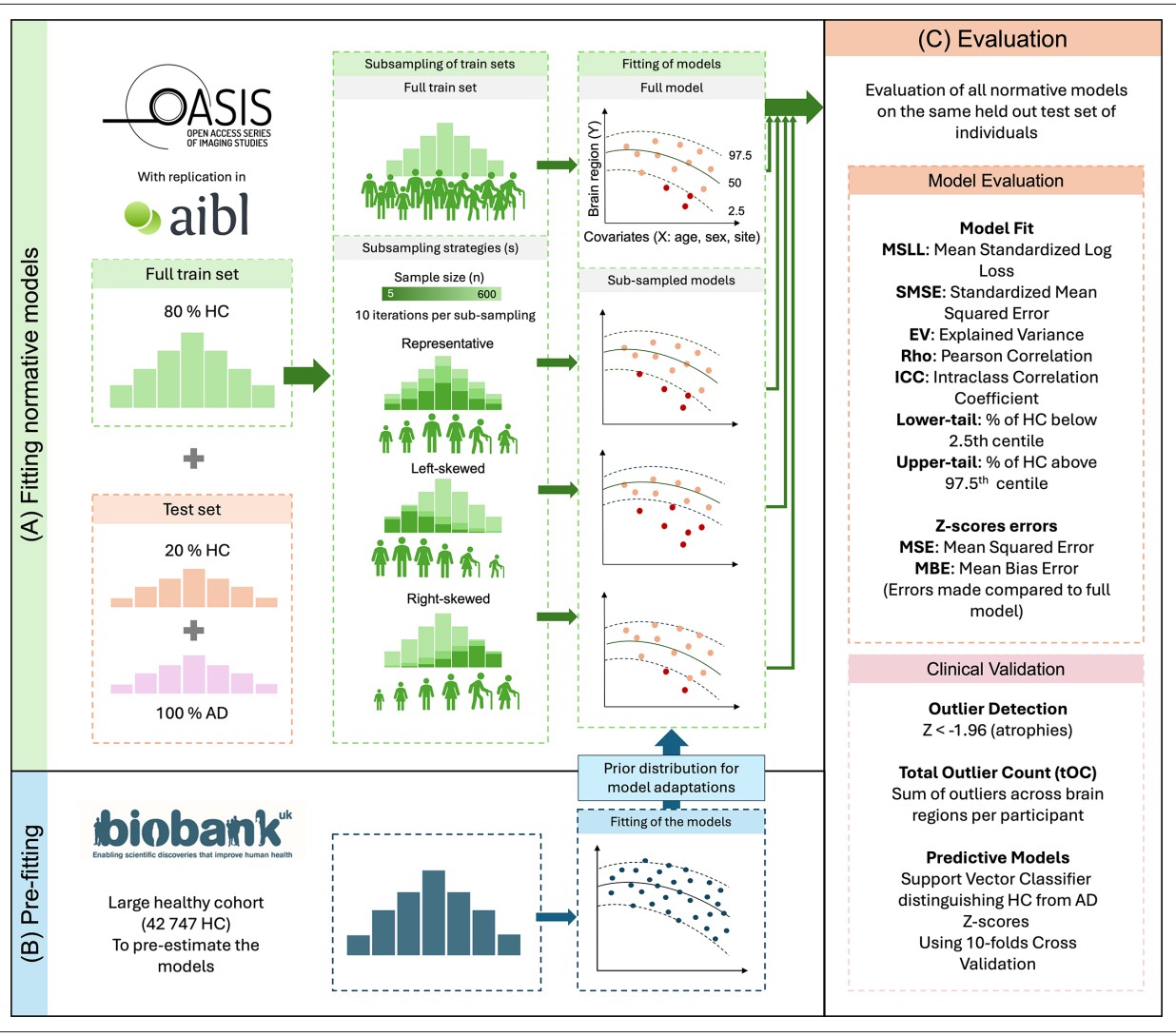

**Figure 1.** Methodology diagram for evaluating normative model estimation under different sampling scenarios. (**A**) Analyses were conducted on the OASIS-3 dataset, with replication in AIBL. Normative models were first fitted using the entire training set (i.e., 80% of HC from the cohort). For each sampling strategy (s), models were re-fitted on randomly drawn subsamples of the training set, with sample sizes (n) ranging from 5 to the maximum available (n = 692 for OASIS, n = 322) and repeated across 10 iterations per sample size. Representative sampling: subsamples preserved the original age distribution using 10 age bins with an equal number of individuals per bin, while also ensuring balanced sex distributions. Left-skewed sampling: overrepresented younger age ranges by applying a beta distribution (α = 2, β = 5), across 10 equally spaced bins. Right-skewed sampling: overrepresented older age ranges using a beta distribution (α = 5, β = 2). Each model was evaluated on a fixed test set composed of 20% of HC from the same cohort and all AD individuals. Example normative fits illustrate how identical test values are interpreted under different models: red dots represent values outside the 95% centile range (outliers), while orange dots fall within the normative range. (**B**) In a parallel analysis, normative models were pre-trained on the UK Biobank dataset. Adaptive transfer learning was used to adapt these pre-trained models to the clinical cohorts. The same subsampling strategies and sample sizes described in panel A were used here to define the adaptation sets. Adapted models were then applied to the same fixed test set as in panel A, allowing direct comparison between within-cohort models and adapted UKB models across all sampling conditions. (**C**) Each resulting model was evaluated using standard performance metrics to assess model fit. Z-score errors were additionally computed and compared to those obtained from the model fitted using the full training data. Clinical validation was then performed by analyzing outlier detection and predictive performance.

## Results

We first evaluated how reference population characteristics influenced normative model accuracy. Using Warped BLR (*Fraza et al., 2021*), we trained normative models for 167 neuroanatomical regions of interest (ROI) using two independent datasets, OASIS-3 and AIBL (*Figure 1*, *Table 1*, *Appendix 4—table 1*). Training subsets ranged from five participants to the entire training cohort (n = 692), obtained

**Table 1.** Demographics of HC and AD participants across datasets and sites summary of age ranges, mean ages, standard deviations, and sex ratios (F/M) for HC and AD groups within the UKB, OASIS-3, and AIBL datasets, including site-specific subgroups.

| Dataset | Site | N | Age range (mean ± sd) | Sex ratio F/M (%) | N | Age range (mean ± sd) | Sex ratio F/M (%) |
|---|---|---|---|---|---|---|---|
| | | Healthy controls group | | | Alzheimer's disease group | | |
| UKB | Full dataset | 42,747 | 44.61 – 82.79 (64.50 ± 7.67) | 52.75/47.25 | | | |
| | UKB 1 | 25,536 | 44.61 – 82.34 (63.68 ± 7.57) | 51.78/48.22 | | | |
| | UKB 2 | 10,914 | 48.68 – 82.79 (66.33 ± 7.87) | 54.20/45.80 | | | |
| | UKB 3 | 6297 | 47.92 – 81.93 (65.36 ± 7.52) | 54.19/45.81 | | | |
| OASIS-3 | Full dataset | 865 | 45.41 – 82.69 (68.21± 7.83) | 58.96/41.04 | 167 | 50.79 – 82.67 (73.79 ± 5.99) | 44.91/55.09 |
| AIBL | Full dataset | 403 | 60.13 – 82.61 (72.24 ± 5.40) | 56.67/43.33 | 60 | 58.69 – 82.75 (73.59 ± 6.62) | 56.67/43.33 |
| | AIBL 1 | 274 | 60.99 – 82.61 (72.64 ± 5.26) | 58.03/41.97 | 39 | 60.33 –82.75(73.05 ± 6.35) | 53.85/46.15 |
| | AIBL 2 | 129 | 60.13 – 82.56 (71.39 ± 5.63) | 60.16/39.84 | 21 | 58.69 – 81.43 (74.60 ± 7.14) | 61.90/38.10 |

by subsampling to reproduce realistic age- and sex-related skews. For each subsampling scenario, we generated 10 random draws and benchmarked their performance against the model trained on the full training sample.

## Model fit evaluation

First, to quantify training-set sample size effects, we trained normative models on HC subsamples with representative age distributions (mirroring the full training set age distribution) and balanced sex ratios. We evaluated these models on a fixed HC test set and benchmarked their performance against corresponding full training sample models (*Figure 1A*). Normative model fits were assessed using *Mean Standardized Log Loss (MSLL)*, *Standardized Mean Squared Error (SMSE)*, *Explained Variance (EV)*, *Pearson Correlation (Rho)*, and *Intraclass Correlation Coefficient (ICC)*. In addition to standard fit metrics, calibration was assessed by estimating the percentage of HC test individuals falling outside the central 95% normative interval (2.5–97.5%) (*Figure 1C*).

Sample size demonstrated a strong influence on model fit for cortical thicknesses and subcortical volumes (*Figure 2*). Specifically, model performance improved with larger sample sizes, as reflected by reductions in *MSLL* ($\beta$=–0.496, p<0.001) and *SMSE* ($\beta$ = –0.427, p < 0.001) (*Figure 2A, B*) and increases in *EV* ($\beta$ = 0.499, p < 0.001), *Rho* ($\beta$ = 0.527, p < 0.001), and *ICC* ($\beta$ = 0.728, p < 0.001) (*Figure 2C, D, G*, *Table 2*). Increasing sample size was consistently associated with a strong reduction in upper-tail HC deviations across both modeling frameworks ($\beta$ = −0.603, p < 0.001) (*Figure 2E, F*, *Table 2*). In contrast, effects on the lower tail were small and not consistently observed across models (*Tables 2 and 3*). Illustrative centile overlays for selected cortical and subcortical regions across sampling strategies and sample sizes are provided in the Figure Supplements of Figure 4 (*Figure 4—figure supplements 2 and 5*).

The standardized effect sizes ($\beta$ values) allow for direct comparisons across metrics and sampling strategies. Performance improved consistently from $n$ = 10 to the full training sample size ($n$ = 692), with $n$ = 5 showing unstable values and falling outside the overall trend. Rapid improvements were observed between $n$ = 10 and 50, capturing ~70–80% of total gains across metrics. Gains continued more gradually between $n$ = 50 and 200, accounting for most of the remaining improvements (~15–25%). Beyond $n$ = 200, all metrics reached ~92–95% of their final values. After $n$ = 300, changes became minimal, and metrics plateaued near 97–99% (*Figure 2*). Notably, ICC reaches excellent reliability from $n$ = 50 across all lobes (*Koo and Li, 2016*).

Next, to isolate the effect of age distribution, we compared models trained on HC subsamples with representative age distributions (as used above) to those trained using left-skewed (younger-biased)

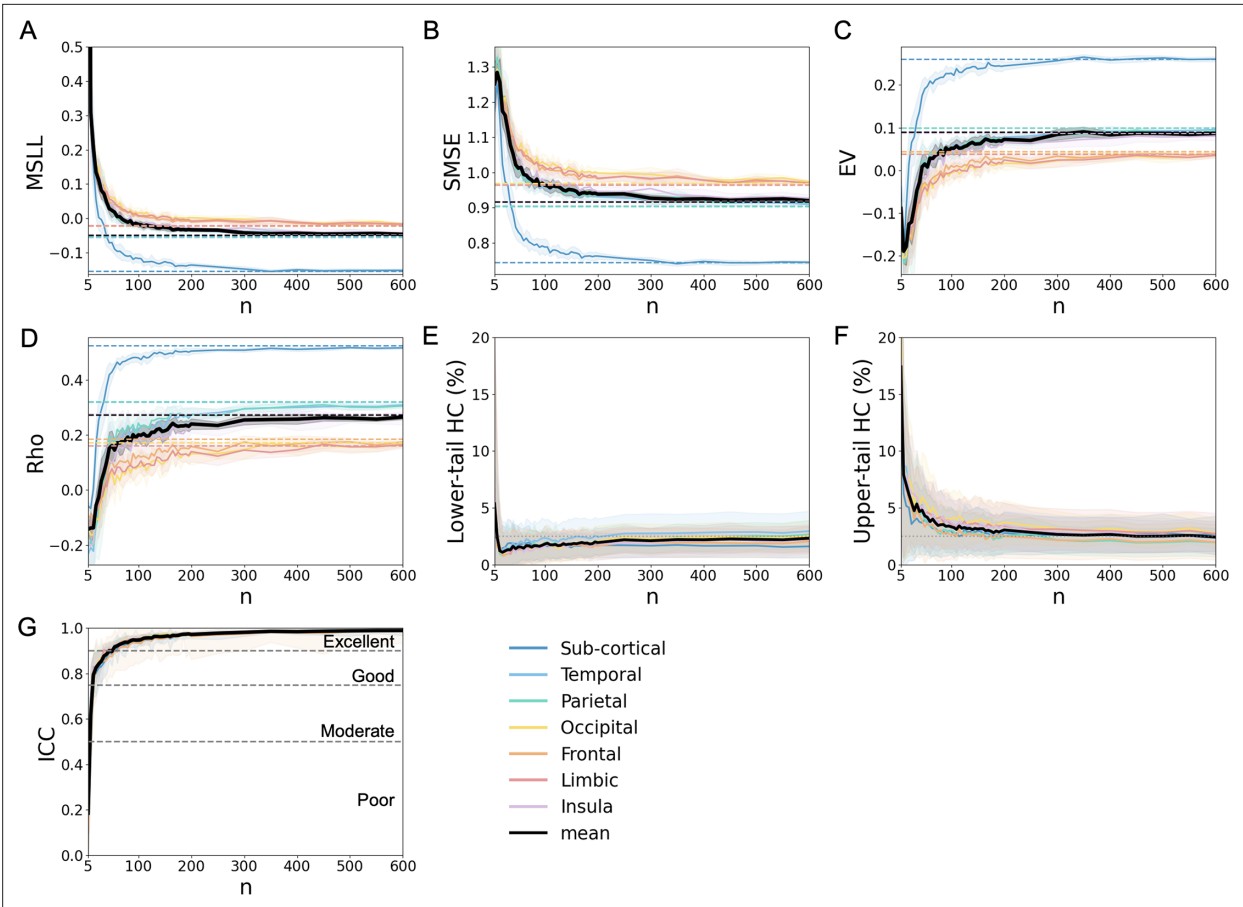

**Figure 2.** Model fit evaluation in the HC test set. Performance evaluation of models as a function of the number of participants used for fitting the models (*n*), calculated using the HC test set from the OASIS-3 dataset. Depicted are in (**A**) the MSLL; (**B**) SMSE; (**C**) EV; (**D**) Rho; (**E**) lower-tail HC percentage (below the 2.5% bound); (**F**) upper-tail HC percentage (above the 97.5% bound); and (**G**) ICC. For all plots, solid lines represent the mean performance of the evaluation metric across all cortical region models grouped by lobe and the mean performance across subcortical region models. The thick black line indicates the overall mean performance across all cortical and subcortical models. In panels (**A–F**), dashed lines denote the performance using the full training set (*n* = 692) and the shaded areas indicate the standard deviation, reflecting variability across 10 iterations of the same sample size. In panel (**G**), gray dotted lines indicate commonly accepted reliability thresholds: below 0.5 (poor), up to 0.75 (moderate), up to 0.9 (good), and above 0.9 (excellent) (*Koo and Li, 2016*).

The online version of this article includes the following figure supplement(s) for figure 2:

**Figure supplement 1.** Model fit evaluation of normative models trained within OASIS-3.

**Figure supplement 2.** Model fit evaluation of normative models pre-trained on the UKB and adapted to OASIS-3.

**Figure supplement 3.** Model fit evaluation of normative models trained within AIBL.

**Figure supplement 4.** Model fit evaluation of normative models pre-trained on the UKB and adapted to AIBL.

and right-skewed (older-biased) age distributions, while maintaining balanced sex ratios. All models used the same incremental sample sizes and were evaluated on the same fixed HC test set (see *Figure 1A, C*, and *Table 2*).

Left-skewed sampling (i.e., oversampling of younger subjects) had the most pronounced adverse effect on model fit, with a marked increase in *SMSE* ($\beta$ = 1.323, p < 0.001), increase in *MSLL* ($\beta$ = 0.219, p < 0.001), decreases in *EV* ($\beta$ = –0.927, p < 0.001), *Rho* ($\beta$ = –0.642, p < 0.001), and in *ICC* ($\beta$ = –0.300, p < 0.001). Right-skewed sampling (i.e., oversampling of older subjects) had a smaller effect on model fits compared to left-skewed sampling. It increased *SMSE* ($\beta$ = 0.585, p<0.001) and *MSLL* ($\beta$ = 0.115, p < 0.001) and decreased *EV* ($\beta$ = –0.692, p < 0.001) and *Rho* ($\beta$ = –0.567, p < 0.001). The effect on *ICC* was minimal ($\beta$ = –0.024, p = 0.026). Age-skewed sampling also affected calibration: left-skewed training distributions were associated with increased deviations beyond the lower bound of the nominal 95% prediction interval ($\beta$ = 1.410, p < 0.001) and a reduction in upper-tail deviations

**Table 2.** Linear mixed model results for evaluation metrics under age-skewed sampling conditions.

Models assess the influence of standardized and log-transformed sample size ($n$) and age sampling strategy (Representative, Left-skewed, Right-skewed) on model performance metrics (MSLL, SMSE, EV, Rho, ICC, the lower and upper tail HC%). All variables were standardized to allow comparison of effect sizes. Sample size was log-transformed to account for the non-linear association between sample size and model performance. Representative sampling serves as the reference level. Reported coefficients and corresponding p-values indicate the direction and significance of each effect.

| | MSLL $\beta$ (p-value) | SMSE $\beta$ (p-value) | EV $\beta$ (p-value) | Rho $\beta$ (p-value) | ICC $\beta$ (p-value) | Lower tail HC % $\beta$ (p-value) | Upper tail HC % $\beta$ (p-value) |
|---|---|---|---|---|---|---|---|
| Intercept (representative) | –0.031 ($p$ = 0.022) | –0.318 (p < 0.001) | 0.260 (p < 0.001) | 0.183 (p < 0.001) | 0.106 (p < 0.001) | –0.203 (p < 0.001) | –0.035 ($p$ = 0.322) |
| Log($n$) | –0.496 (p < 0.001) | –0.427 (p < 0.001) | 0.499 (p < 0.001) | 0.527 (p < 0.001) | 0.728 (p < 0.001) | –0.015 ($p$ = 0.126) | –0.603 (p < 0.001) |
| Left-skewed | 0.219 (p < 0.001) | 1.323 (p < 0.001) | –0.927 (p < 0.001) | –0.642 (p < 0.001) | –0.300 (p < 0.001) | 1.410 (p < 0.001) | –0.205 (p < 0.001) |
| Right-skewed | 0.115 (p < 0.001) | 0.585 (p < 0.001) | –0.692 (p < 0.001) | –0.567 (p < 0.001) | –0.024 ($p$ = 0.026) | –0.068 (p < 0.001) | 0.364 (p < 0.001) |
| Log($n$):Left-skewed | –0.064 (p < 0.001) | –0.636 (p < 0.001) | 0.019 ($p$ = 0.047) | –0.112 (p < 0.001) | –0.032 ($p$ = 0.003) | –0.983 (p < 0.001) | 0.340 (p < 0.001) |
| Log($n$): Right-skewed | –0.084 (p < 0.001) | –0.139 (p < 0.001) | 0.047 (p < 0.001) | –0.034 (p < 0.001) | –0.048 (p < 0.001) | 0.056 (p < 0.001) | –0.295 (p < 0.001) |

($\beta$ = −0.205, p < 0.001), whereas right-skewed sampling showed reduced lower-tail deviations ($\beta$ = −0.068, p < 0.001) and increased upper-tail deviations ($\beta$ = 0.364, p < 0.001).

A consistent dip in performance was observed around $n$ = 300 for the left-skewed sampling condition in the original analysis (**Figure 3**). To assess whether this reflected sensitivity to the specific subsampling or stochastic sampling variability, we repeated the analysis for this specific sample using 20 independent random seeds (**Appendix 5—figure 1**); the absence of a consistent effect across repetitions indicates that the original pattern was driven by sampling variability rather than a systematic model artifact.

Model fit generally improved with increasing sample size across all metrics and sampling strategies. Interaction effects, however, revealed that the rate of improvement varied depending on the sampling strategy. Under left-skewed sampling, *SMSE* improved substantially with larger sample sizes (interaction $\beta$ = –0.636, p < 0.001), suggesting a steeper recovery. *Rho* and *ICC* improved, but at a slower rate of improvement than under representative sampling (interaction $\beta$ = –0.112, p < 0.001 and $\beta$ = –0.032, p = 0.003). Right-skewed sampling showed uniform improvements across metrics, with interaction effects indicating modest gains in *EV* ($\beta$ = –0.047, p < 0.001), Rho ($\beta$ = –0.034, p < 0.001), and ICC ($\beta$ = –0.048, p < 0.001), and reductions in *MSLL* ($\beta$ = –0.084, p < 0.001) and SMSE ($\beta$ = –0.139, p < 0.001). Interaction effects were also observed for calibration. Under left-skewed sampling, increasing sample size was associated with a strong reduction in lower-tail HC deviations (interaction $\beta$ = –0.983, p<0.001) and an increase in upper-tail deviations (interaction $\beta$ = 0.340, p < 0.001). Under right-skewed sampling, the pattern was reversed, with lower-tail deviations increasing (interaction $\beta$ = 0.056, p < 0.001) and upper-tail deviations decreasing (interaction β = −0.295, p < 0.001).

These results highlight that the effect of increasing sample size varies depending on the sampling strategy, with less consistent improvements observed under left-skewed conditions. All results for age-skewed samplings are summarized in **Table 2**.

Finally, to examine the effect of sex ratio imbalance, we trained models on HC subsamples with representative age distributions and varying sex ratios: balanced (1F:1M), 1F:10M, 1F:4M, 4F:1M, and 10F:1M, where F denotes females and M denotes males. As before, all models used the same incremental sample sizes and were evaluated on the fixed HC test set.

Sex distribution imbalances had a comparatively smaller effect on model fit than age distribution shifts (**Table 3**, **Figure 3**). Across metrics, the extent of deviation increased with the degree of imbalance. More extreme ratios (1F:10M and 10F:1M) were associated with moderate increases in *SMSE* ($\beta$ = 0.166 and $\beta$ = 0.076, respectively; both p < 0.001) and reductions in *EV* ($\beta$ = –0.124 and $\beta$ = –0.033,

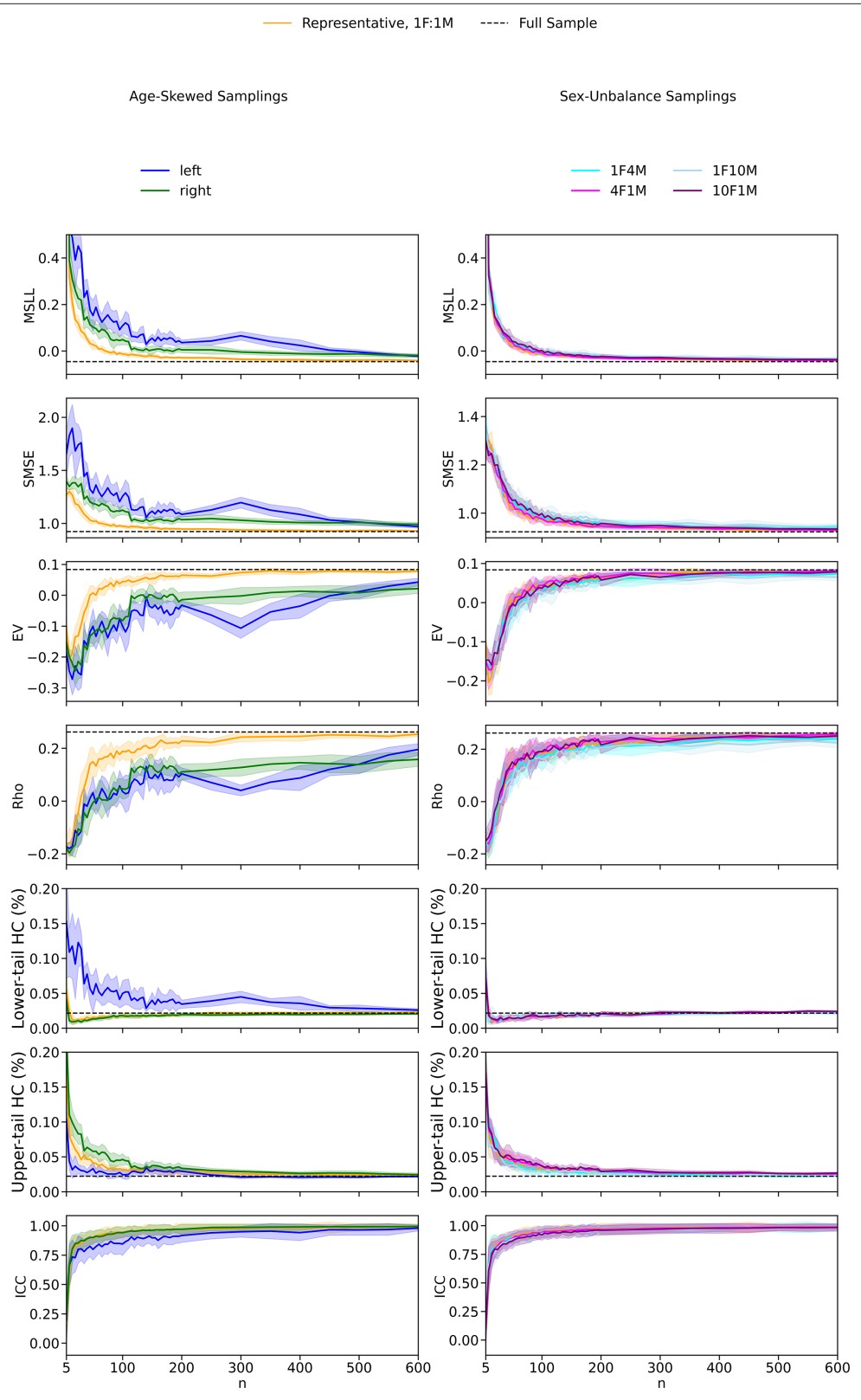

**Figure 3.** Evaluation of model fits in the HC test set across different sampling strategies of the training set, shown for varying sample sizes (*n*). Age-skewed sampling strategies include representative (matching the initial age distribution with balanced sex, 1F:1M), left-skewed (favoring younger individuals), and right-skewed (favoring older individuals), each with balanced sex distributions. Sex-imbalanced sampling strategies include female-to-

*Figure 3 continued on next page*

*Figure 3 continued*

male ratios of 1:1, 1:4, 1:10, 4:1, and 10:1, all with representative age distributions. Depicted metrics include MSLL, SMSE, EV, Rho, ICC, the lower and upper tail HC%. Solid lines represent the mean metric values across ROIs and iterations, with shaded areas indicating the standard deviation across iterations for MSLL, SMSE, EV, Rho, and tail percentages. For ICC, shaded areas represent variation across ROIs, as ICC already reflects variability across iterations. Dashed lines indicate the mean metric values for models trained with the full sample for MSLL, SMSE, EV, Rho, and tail percentages.

The online version of this article includes the following figure supplement(s) for figure 3:

**Figure supplement 1.** Regional linear mixed-effects results for model-fit metrics in models trained in OASIS-3 with age-skewed samples.

**Figure supplement 2.** Regional linear mixed-effects results for model-fit metrics in models trained in OASIS-3 with sex-imbalanced samples.

**Figure supplement 3.** Model fit evaluation of normative models pre-trained on UKB and adapted to OASIS-3 across different sampling strategies.

**Figure supplement 4.** Regional linear mixed-effects results for model-fit metrics in models pre-fitted in the UKB and adapted to OASIS-3 with age-skewed samples.

**Figure supplement 5.** Regional linear mixed-effects results for model-fit metrics in models pre-fitted in the UKB and adapted to OASIS-3 with sex-imbalanced samples.

**Figure supplement 6.** Model fit evaluation of normative models trained on AIBL across different sampling strategies.

**Figure supplement 7.** Regional linear mixed-effects results for model-fit metrics in models trained in AIBL with age-skewed samples.

**Figure supplement 8.** Regional linear mixed-effects results for model-fit metrics in models trained in AIBL with sex-imbalanced samples.

**Figure supplement 9.** Model fit evaluation of normative models pre-trained on the UKB and adapted to AIBL across different sampling strategies.

**Figure supplement 10.** Regional linear mixed-effects results for model-fit metrics in models pre-fitted in the UKB and adapted to AIBL with age-skewed samples.

**Figure supplement 11.** Regional linear mixed-effects results for model-fit metrics in models pre-fitted in the UKB and adapted to AIBL with sex-imbalanced samples.

**Figure supplement 12.** Training age-distribution coverage with respect to the HC test cohorts in OASIS-3 (left) and AIBL (right).

respectively; both p < 0.001), while more moderate imbalances (1F:4M and 4F:1M) produced minimal changes. *ICC* was reduced in all imbalanced conditions, with the largest effects observed under the most extreme ratios ($\beta$ = –0.152 for 1F:10M and $\beta$ = –0.162 for 10F:1M; both p < 0.001). Effects on calibration across sex ratios were small, with only modest changes in HC percentages in the lower and upper tails. Interaction effects with sample size were statistically significant in some cases but remained of small magnitude. All results for sex-unbalanced samplings are summarized in *Table 3*.

In summary, these findings demonstrate that representative sampling generally provided the best performance in terms of model fit across metrics on the test set, while skewed distributions introduced substantial model fit degradation. The left-skewed sampling particularly exacerbated errors, as shown by the larger effect sizes. Increasing the sample size improved fits in all configurations. Sex ratio imbalance had a statistically significant but limited effect on model fit, with only minor deviations observed across metrics. These findings underscore the dominant influence of age distribution and sample size in model fit. Individual effects of sample size for the different age and sex distributions are presented in *Figure 2—figure supplement 1*, *Figure 3—figure supplements 1 and 2*, and importantly, we replicated these findings in an independent dataset (AIBL) (*Figure 2—figure supplement 3*, *Figure 3—figure supplements 7 and 8*, *Appendix 1—tables 1 and 2*).

## Z-score errors

Subsequently, we assessed the direct impact of age and sex distributions and sample size on normative models' outcomes, quantified by the *mean squared error* (*MSE*) and *mean bias error* (*MBE*) in Z-scores relative to models trained with the full training set.

**Table 3.** Linear mixed model results for evaluation metrics under sex-imbalanced sampling conditions.

Models assess the influence of standardized and log-transformed sample size (*n*) and sex ratio in the training set (1F:1M, 1F:4M, 1F:10M, 4F:1M, 10F:1M; F = female, M = male) on model performance metrics (MSLL, SMSE, EV, Rho, ICC, the lower and upper tail HC%). All variables were standardized to allow comparison of effect sizes. Sample size was log-transformed to account for the non-linear association between sample size and model performance. Representative sampling (1F:1M) serves as the reference level. Reported *β* coefficients and corresponding p-values indicate the direction and significance of each effect.

| | MSLL $\beta$ (p-value) | SMSE $\beta$ (p-value) | EV $\beta$ (p-value) | Rho $\beta$ (p-value) | ICC $\beta$ (p-value) | Lower tail HC % $\beta$ (p-value) | Upper tail HC % $\beta$ (p-value) |
|---|---|---|---|---|---|---|---|
| Intercept (1F1M) | –0.031 (p = 0.005) | –0.318 (p < 0.001) | 0.260 (p < 0.001) | 0.183 (p < 0.001) | NA | –0.203 (p < 0.001) | –0.035 (p = 0.308) |
| Log(*n*) | –0.496 (p < 0.001) | –0.427 (p < 0.001) | 0.498 (p < 0.001) | 0.527 (p < 0.001) | 0.728 (p < 0.001) | –0.015 (p = 0.004) | –0.603 (p < 0.001) |
| 10F1M | 0.001 (p = 0.965) | 0.076 (p < 0.001) | –0.023 (p < 0.001) | 0.044 (p < 0.001) | –0.152 (p < 0.001) | 0.010 (p = 0.194) | 0.151 (p < 0.001) |
| 4F1M | –0.015 (p = 0.252) | 0.010 (p = 0.030) | 0.010 (p = 0.108) | 0.042 (p < 0.001) | –0.055 (p < 0.001) | 0.015 (p = 0.043) | 0.091 (p < 0.001) |
| 1F4M | –0.039 (p = 0.003) | 0.068 (p < 0.001) | –0.062 (p < 0.001) | –0.062 (p < 0.001) | –0.048 (p < 0.001) | 0.020 (p = 0.007) | –0.028 (p = 0.002) |
| 1F10M | –0.062 (p < 0.001) | 0.166 (p < 0.001) | –0.121 (p < 0.001) | –0.098 (p < 0.001) | –0.162 (p < 0.001) | 0.031 (p < 0.001) | –0.127 (p < 0.001) |
| Log(*n*):10F1M | 0.057 (p < 0.001) | –0.018 (p < 0.001) | 0.005 (p = 0.428) | –0.034 (p < 0.001) | 0.073 (p < 0.001) | –0.027 (p < 0.001) | –0.033 (p < 0.001) |
| Log(*n*):4F1M | 0.060 (p < 0.001) | –0.009 (p = 0.051) | 0.015 (p = 0.018) | –0.011 (p = 0.114) | 0.048 (p < 0.001) | –0.033 (p < 0.001) | –0.022 (p = 0.014) |
| Log(*n*):1F4M | 0.159 (p < 0.001) | –0.020 (p < 0.001) | 0.001 (p = 0.874) | –0.011 (p = 0.134) | –0.013 (p = 0.221) | –0.022 (p = 0.003) | –0.038 (p < 0.001) |
| Log(*n*):1F10M | 0.279 (p < 0.001) | –0.032 (p < 0.001) | –0.019 (p = 0.002) | –0.049 (p < 0.001) | –0.017 (p = 0.120) | 0.033 (p < 0.001) | 0.163 (p < 0.001) |

Increasing the sample size significantly reduced *MSE* in both HC and AD groups ($\beta$ = –0.443, p<0.001), with no significant interaction, indicating similar effects across groups. Age distribution sampling (*Figure 4A*) had a significant effect on *Z*-score errors: left-skewed sampling (i.e., oversampling of younger individuals) led to a larger increase in *MSE* ($\beta$ = 1.045, p<0.001) compared to right-skewed sampling (i.e., oversampling of older individuals) ($\beta$ = 0.101, p<0.001) (*Figure 4B*, *Table 4*).

Because the MSE quantifies the magnitude of errors but does not provide information about their direction, we additionally computed the *MBE* to assess whether errors reflected systematic over- or underestimation of deviation scores. Left-skewed sampling consistently resulted in negative *MBE* values, indicating overestimation of deviations, while right-skewed sampling produced positive *MBE* values, indicating underestimation, particularly at smaller sample sizes (*Figure 4B*).

Cubic regression analyses on age revealed that, under the representative distribution, *Z*-score errors were lowest in mid-range ages and increased toward both extremes. Left-skewed sampling amplified errors in older individuals, with deviations persisting even at larger sample sizes (e.g., *n* = 100). In contrast, right-skewed sampling led to elevated errors in younger individuals, which progressively decreased with increasing sample size (*Figure 4D*).

Age-related trends in *MSE* were exemplified in centile estimation maps for the left hippocampus (*Figure 4E*), where left-skewed sampling led to higher centiles in older individuals (indicating overestimation), while right-skewed sampling produced lower centiles in younger individuals (indicating underestimation) (*Figure 4E*).

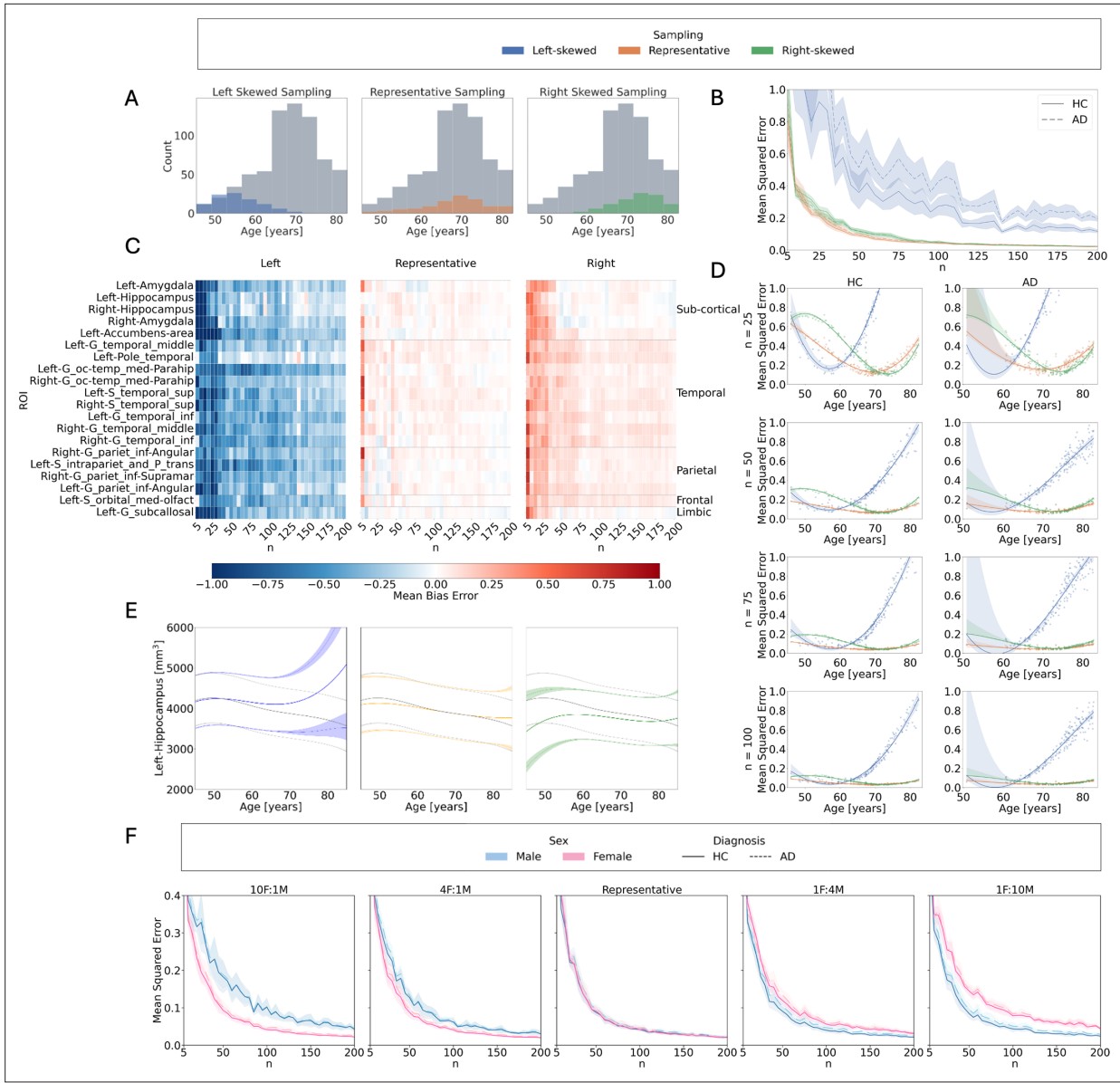

**Figure 4.** *Z*-score errors: Influence of sample size, age distribution, and sex imbalance on normative model outcomes. (**A**) Age distributions for left-skewed (younger-biased), representative, and right-skewed (older-biased) sampling strategies, compared to the full training set in the OASIS-3 dataset. (**B**) Mean squared error (MSE) of *Z*-scores relative to the full training set model across sample sizes, age distributions, and diagnostic groups. (**C**) Mean bias error (MBE) per region across sample sizes and age distributions in the test set. Shown are the 20 brain regions with the highest Cohen's *d* effect sizes distinguishing between HC and AD calculated from models trained on the full training set. From left to right: results for left-skewed, representative, and right-skewed training sets. Blue indicates negative MBE (underestimation), red indicates positive MBE (overestimation), and white indicates close alignment with the full training set model. (**D**) Cubic regression of MSE as a function of age, across sample sizes and sampling strategies. Left-skewed sampling shows increased errors in older individuals; right-skewed sampling shows increased errors in younger individuals. (**E**) Centile curves for the left hippocampus as a function of age, derived from models trained on left-skewed, representative, and right-skewed sampling (*n* = 100). Colored lines represent the 5th, 50th, and 95th percentiles; gray lines show centiles from the full training set model. (**F**) MSE across sample sizes and test set sex, obtained using sex-imbalanced training sets (female-to-male ratios: 10:1, 4:1, 1:1, 1:4, 1:10), all with representative age distributions.

The online version of this article includes the following figure supplement(s) for figure 4:

**Figure supplement 1.** Cohen's *d* effect sizes computed on full models' *Z*-scores for discriminating HC from AD groups for each ROI.

**Figure supplement 2.** Centile curve overlays for selected cortical and subcortical regions across sampling strategies and sample sizes in OASIS-3 dataset.

**Figure supplement 3.** *Figure 5.*

*Figure 4 continued on next page*

*Figure 4 continued*

**Figure supplement 4.** *Z*-score errors in AIBL using models trained directly on the dataset: Influence of sample size, age distribution, and sex imbalance on normative model outcomes.

**Figure supplement 5.** Centile curve overlays for selected cortical and subcortical regions across sampling strategies and sample sizes in AIBL dataset.

**Figure supplement 6.** *Z*-score errors in AIBL using models pretrained in UKB and adapted to AIBL: Influence of sample size, age distribution, and sex imbalance on normative model outcomes.

Sex imbalances in the training set had a smaller but statistically significant effect on *Z*-score errors (*Table 5*). Extreme imbalances led to higher *MSE* compared to the representative configuration, with the largest increases observed for the 10F:1M ($\beta$ = 0.089, p < 0.001) and 1F:10M ($\beta$ = 0.083, p < 0.001) ratios and smaller increases for 4F:1M ($\beta$ = 0.031, p < 0.001) and 1F:4M ($\beta$ = 0.046, p < 0.001). Errors did not statistically differ for the HC and AD groups ($\beta$=0.021, p = 0.137). Errors tended to be lower for individuals matching the overrepresented sex in the training set, with discrepancies increasing with the degree of imbalance (*Figure 4F*).

Supplementary analysis (*Appendix 4—tables 5 and 7*) also showed that main effect of age was not significant for either MSE or tOC, and no significant age × sex-ratio interactions were observed. While some higher-order interactions involving age, diagnosis, and sex ratio reached statistical significance, all associated effect sizes were very small and inconsistent across outcomes, indicating that the observed error changes are not driven by residual age confounding.

Highly similar results were found for the AIBL dataset, with the exception that the right-skewed distribution led to a larger effect than in the OASIS-3 dataset (*Figure 4—figure supplement 4*, *Appendix 2—tables 3 and 4*).

## Clinical validation

Clinical validity was evaluated for every sampling strategy and sample size using three metrics: (1) the proportion of subjects with atrophy outliers in each ROI ($Z < -1.96$); (2) the *total outlier count* (*tOC*) per subject, defined as the number of ROIs in which a subject's deviation score exceeds the *Z*

**Table 4.** Linear mixed model results for MSE and total outlier count (tOC) under age-skewed sampling.

Models evaluate the influence of diagnosis (HC, AD), log-transformed and standardized sample size (*n*), and age sampling strategy (Representative, Left-skewed, Right-skewed) on standardized deviation score outcomes. Continuous variables were standardized to allow comparison of effect sizes. Sample size was log-transformed to account for the non-linear association between sample size and model performance. Representative sampling and HC are used as reference levels. Reported $\beta$ coefficients and corresponding p-values indicate the direction and significance of the effects. Age was not included as an additional predictor to keep the modeling approach aligned with the sex-based analysis (*Table 5*) and to limit the complexity of the models.

| | MSE $\beta$ (p-value) | tOC $\beta$ (p-value) |
|---|---|---|
| Intercept (HC, Representative) | –0.245 (p < 0.001) | –0.443 (p < 0.001) |
| Log(*n*) | –0.443 (p < 0.001) | –0.004 (p = 0.405) |
| AD | 0.021 (p = 0.570) | 0.665 (p < 0.001) |
| Left-skewed | 1.045 (p < 0.001) | 0.461 (p < 0.001) |
| Right-skewed | 0.101 (p < 0.001) | –0.021 (p = 0.004) |
| Log(*n*):AD | –0.007 (p = 0.530) | 0.096 (p < 0.001) |
| Log(*n*):Left-skewed | –0.752 (p < 0.001) | –0.323 (p < 0.001) |
| Log(*n*):Right-skewed | –0.148 (p < 0.001) | 0.019 (p = 0.008) |
| Left-skewed:AD | 0.575 (p < 0.001) | 0.721 (p < 0.001) |
| Right-skewed:AD | –0.077 (p < 0.001) | 0.016 (p = 0.136) |
| Log(*n*):Left-skewed:AD | –0.234 (p < 0.001) | –0.290 (p < 0.001) |
| Log(*n*):Right-skewed:AD | 0.100 (p < 0.001) | –0.004 (p = 0.674) |

**Table 5.** Linear mixed model results for MSE and total outlier count (tOC) under sex-imbalanced sampling.

Models evaluate the influence of diagnosis (HC, AD), log-transformed and standardized sample size ($n$), and sex ratio in the training set (1:1, 1F:4M, 1F:10M, 4F:1M, 10F:1M) on standardized deviation score outcomes. Continuous variables were standardized to allow comparison of effect sizes. Sample size was log-transformed to account for the non-linear association between sample size and model performance. The 1:1 ratio and HC are used as reference levels. Reported $\beta$ coefficients and corresponding p-values indicate the direction and significance of the effects. Sex was not included as an additional predictor to keep the modeling approach aligned with the age-based analysis (*Table 4*) and to limit the complexity of the models.

| | MSE<br>$\beta$ (p-value) | tOC<br>$\beta$ (p-value) |
|---|---|---|
| Intercept (HC, 1 F 1 M) | –0.245 (p < 0.001) | –0.443 (p < 0.001) |
| Log($n$) | –0.443 (p < 0.001) | –0.004 (p = 0.084) |
| AD | 0.021 (p = 0.137) | 0.665 (p < 0.001) |
| 10F1M | 0.089 (p < 0.001) | 0.005 (p = 0.173) |
| 4F1M | 0.031 (p < 0.001) | 0.006 (p = 0.076) |
| 1F4M | 0.046 (p < 0.001) | 0.004 (p = 0.212) |
| 1F10M | 0.083 (p < 0.001) | 0.005 (p = 0.170) |
| Log($n$):AD | –0.007 (p = 0.153) | 0.096 (p < 0.001) |
| Log($n$):10F1M | –0.021 (p < 0.001) | –0.010 (p = 0.005) |
| Log($n$):4F1M | –0.012 (p = 0.018) | –0.012 (p = 0.001) |
| Log($n$):1F4M | –0.015 (p = 0.003) | –0.005 (p = 0.148) |
| Log($n$):1F10M | 0.037 (p < 0.001) | 0.014 (p < 0.001) |
| 10F1M:AD | 0.025 (p<0.001) | –0.006 (p = 0.258) |
| 4F1M:AD | 0.013 (p = 0.073) | 0.009 (p = 0.079) |
| 1F4M:AD | –0.011 (p = 0.133) | –0.002 (p = 0.718) |
| 1F10M:AD | –0.013 (p = 0.072) | –0.013 (p = 0.012) |
| Log($n$):10F1M:AD | –0.013 (p = 0.078) | 0.009 (p = 0.092) |
| Log($n$):4F1M:AD | –0.013 (p = 0.081) | 0.001 (p = 0.900) |
| Log($n$):1F4M:AD | 0.013 (p = 0.081) | –0.006 (p = 0.220) |
| Log($n$):1F10M:AD | –0.002 (p = 0.750) | 0.000 (p = 0.930) |

threshold; and (3) the *ROC–AUC* of support vector classifiers (SVC) distinguishing HC vs. AD based on deviation scores (*Figure 1C*).

In the model fitted using the full training set, the highest percentages of volume outliers in the AD group were observed in subcortical regions such as the left hippocampus (30.5%) and left amygdala (28.1%). Among cortical areas, the middle temporal gyrus and parahippocampal gyrus also showed frequent deviations (e.g., 22.8% in the right middle temporal gyrus, 19.2% in parahippocampal gyrus).

To illustrate how outlier estimation varies with training sample size, example iterations are shown in *Figure 5*. At $n$ = 25, outlier percentages were substantially lower than in the full sample model, for example, 8.4% in the left hippocampus (–22.1% compared to the full sample) and 0% in the left amygdala (–28.1%). At $n$ = 50, estimates became closer to the full model in some regions (e.g., 24.0% in the left hippocampus, –6.5%), while others showed higher deviation rates (e.g., 41.3% in the left amygdala, +13.2%). Outlier estimates in the example at $n$ = 75 closely matched those of the full model across many regions; for example, the left hippocampus showed 30.5% (–0.0%). At $n$ = 100, values showed minor variability, with the left hippocampus slightly above the reference at 34.7% (+4.2%). This residual fluctuation across runs highlights some instability in outlier estimation even at higher sample sizes. Similar variability was observed in cortical areas. For instance, the percentage of outliers in the right middle temporal gyrus increased from 12.6% at $n$ = 25 (–10.2%) to 18.0% at $n$ = 100

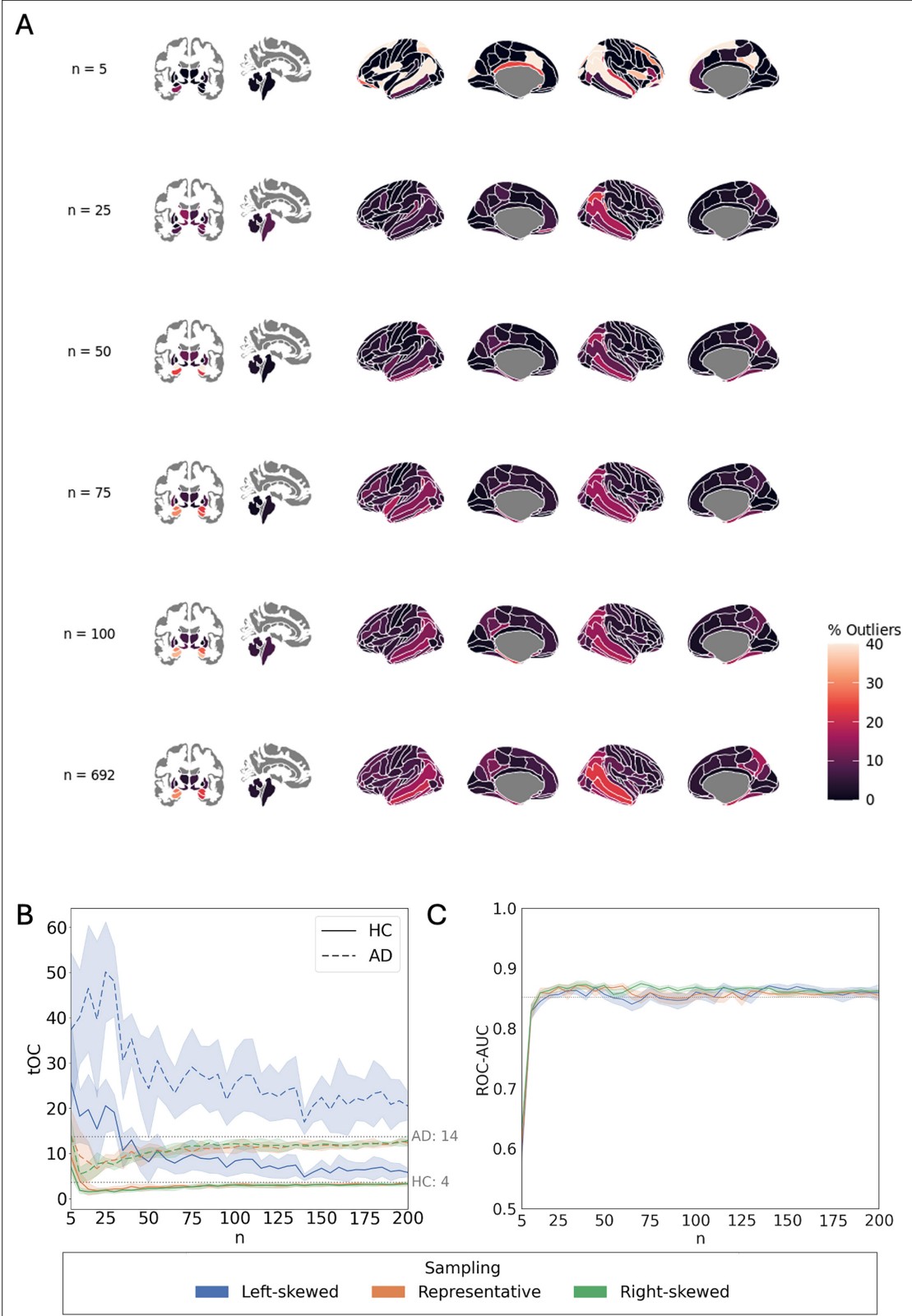

**Figure 5.** Clinical validation: effect of sample size and age distributions in outlier detection and classification performance. (**A**) Percentage of participants with extreme negative deviation (Z<−1.96) in each brain region in the AD group, shown for independent example iterations at different sample sizes with representative age distribution. (**B**) Average total Outlier Count (tOC) in the AD and HC is represented with dashed and solid lines respectively as a function of sample size (*n*). The shaded areas indicate the standard deviation across iterations. Dotted gray lines correspond to the

*Figure 5 continued on next page*

*Figure 5 continued*

estimations of tOC for HC and AD groups obtained with full sample size. (**C**) ROC-AUC of support vector classifier with 10-fold cross-validation as a function of sample size in the training set. The solid lines represent the average AUC across iterations for each sampling strategy. The dotted line shows the performance of the models trained with full sample size.

The online version of this article includes the following figure supplement(s) for figure 5:

**Figure supplement 1.** Regional linear mixed-effects results for deviation-score metrics in models trained in OASIS-3 with age-skewed samples.

**Figure supplement 2.** Regional linear mixed-effects results for deviation-score metrics in models trained in OASIS-3 with sex-imbalanced samples.

**Figure supplement 3.** Clinical validation for models pre-fitted in UKB and adapted to OASIS-3: effect of sample size and age distributions in outlier detection and classification performance.

**Figure supplement 4.** Regional linear mixed-effects results for deviation-score metrics in models pre-fitted in the UKB and adapted to OASIS-3 with age-skewed samples.

**Figure supplement 5.** Regional linear mixed-effects results for deviation-score metrics in models pre-fitted in the UKB and adapted to OASIS-3 with sex-imbalanced samples.

**Figure supplement 6.** Clinical validation for models trained in AIBL: effect of sample size and age distributions in outlier detection and classification performance.

**Figure supplement 7.** Regional linear mixed-effects results for deviation-score metrics in models trained in AIBL with age-skewed samples.

**Figure supplement 8.** Regional linear mixed-effects results for deviation-score metrics in models trained in AIBL with sex-imbalanced samples.

**Figure supplement 9.** Clinical validation for models pre-fitted in UKB and adapted to AIBL: effect of sample size and age distributions in outlier detection and classification performance.

**Figure supplement 10.** Regional linear mixed-effects results for deviation score in models pre-fitted in the UKB and adapted to AIBL with age-skewed samples.

**Figure supplement 11.** Regional linear mixed-effects results for deviation-score metrics in models pre-fitted in the UKB and adapted to AIBL with sex-imbalanced samples.

(–4.8%), compared to 22.8% in the full model. In the right parahippocampal gyrus, values rose from 3.6% at *n* = 25 (–15.6%) to 18.0% at *n* = 100 (–1.2%), relative to 19.2% in the full model.

tOC analysis (*Figure 5B*, *Tables 3 and 4*) showed consistently higher estimated outlier counts in AD compared to HC. Models trained on the full training set estimated an average of 14 outliers per individual in AD compared to 4 in HC. This group difference was confirmed by the statistical models (*Tables 3 and 4*), which reported a significant main effect of diagnosis ($\beta$ = 0.665, p < 0.001), independent of sample size or sampling distributions.

No significant main effect of sample size was found in HC, as outlier counts quickly converged to full-model estimates. In AD, a significant interaction with sample size ($\beta$ = 0.096, p < 0.001) indicated that tOC was underestimated at small sample sizes and increased with larger sample sizes, with representative and right-skewed sampling converging to full-model estimates around *n* = 100 (*Figure 5B*).

Left-skewed sampling strongly increased tOC ($\beta$ = 0.461, p < 0.001), an effect amplified in AD (interaction $\beta$ = 0.721, p < 0.001). Under this sampling strategy, tOC only slowly decreased with larger sample sizes ($\beta$ = –0.323, p < 0.001 for the interaction of sample size with left-skewed sampling; $\beta$ = –0.290, p < 0.001 for the three-way interaction with AD).

Sex-imbalanced samplings showed no significant main effects on tOC (all p > 0.05). Several interactions with sample size were significant for unbalanced sex ratios (e.g., 4F1M $\beta$ = –0.012, p = 0.001; 1F10M $\beta$ = 0.014, p < 0.001), but these effects were very small compared to those observed for age distributions. The only interaction with AD reaching significance was under the extreme male-dominated condition (1F10M:AD $\beta$ = –0.013, p = 0.012), indicating a slight moderation of tOC in the AD group in that scenario.

Finally, the influence of training sample size and age distribution on classification performance was assessed by evaluating the ability to distinguish AD and HC groups based on deviation scores from the normative models (i.e., Z-scores). ROC–AUC increased when increasing sample size for all sampling strategies, reaching a performance comparable to the models trained with the full training sample size (AUC = 0.86) at around 15 samples and stabilized from that point onwards (5 C). This suggests that small training sets may already capture relevant group-level patterns.

Importantly, all findings from the clinical validation analyses were replicated in the independent AIBL dataset (*Figure 5—figure supplement 6*, *Appendix 2—Tables 3 and 4*).

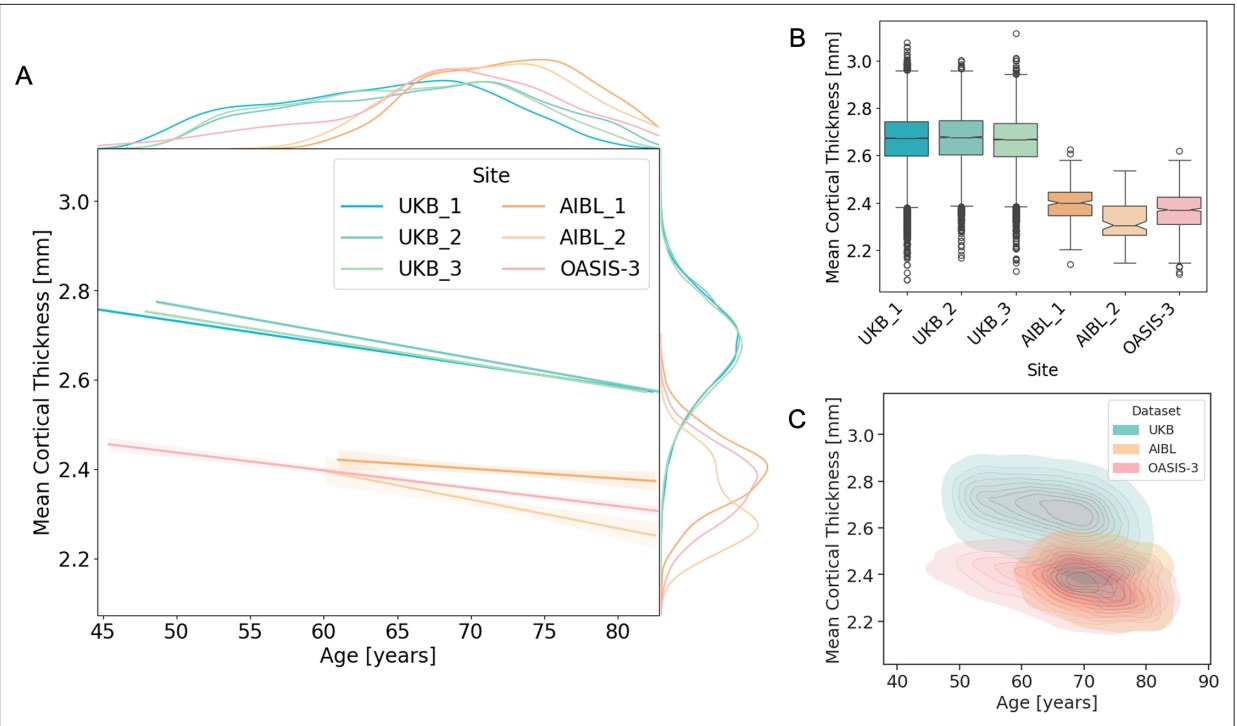

**Figure 6.** Site effects on mean cortical thickness (whole-brain average) in the UK Biobank (UKB), AIBL, and OASIS-3 datasets for HC. (**A**) Regression plots showing the relationship between age and mean cortical thickness across different datasets and imaging sites (UKB: three sites, AIBL: two sites, OASIS-3: one site). Marginal density plots are displayed on the sides of each axis, illustrating the distribution of mean cortical thickness and age for each dataset's imaging sites. The UKB exhibits notable differences in mean cortical thickness estimations compared to AIBL and OASIS-3, unrelated to the age distribution of participants. (**B**) Boxplots representing the mean cortical thickness for participants in each dataset and imaging site. (**C**) Bivariate kernel density estimates plots showing the joint distribution of age and mean cortical thickness for each dataset. The contour plots represent the density of data points, with filled areas reflecting higher concentrations of data.

The online version of this article includes the following figure supplement(s) for figure 6:

**Figure supplement 1.** Site effects on sub-cortical gray matter Volume in the UK Biobank (UKB), AIBL, and OASIS-3 datasets for healthy controls.

## Adaptation from large dataset

To motivate the need for model adaptation across cohorts (*Figure 1B*), we examined differences in mean cortical thickness between UKB and the clinical datasets. Systematic differences were evident, with UKB consistently showing higher mean cortical thickness compared to AIBL and OASIS-3, regardless of age (*Figure 6*). These differences most likely reflect acquisition-related variability and highlight the necessity of adapting models when applying the method to new datasets.

Adapted models based on pre-trained UKB data and transferred to OASIS-3 rapidly achieved the performance level of models trained directly within the cohort (*Figure 7*). Across all evaluation metrics (*MSLL, SMSE, EV, Rho*, and *MSE*) adapted models plateaued around *n* = 50, whereas models trained directly within cohort required approximately *n* = 200 to reach comparable performance. Explained variance (EV) and correlation (Rho) remained stable across all adaptation sample sizes, as adaptation modifies only the mean of the predictions without changing their variance or rank structure.

Despite the convergence in evaluation metrics, outlier detection at full adaptation sample size (*n* = 692) revealed differences between the two modeling strategies, with and without adaptive transfer learning. In the left amygdala, the adapted model detected 15.0% outliers compared to 28.1% for the models trained directly without adaptation (–13.1%); in the left hippocampus, 17.4% vs 30.5% (–13.1%); in the right amygdala, 11.4% vs 23.4% (–12.0%); and in the right hippocampus, 15.0% vs 22.2% (–7.2%). Cortical regions also showed discrepancies, such as the right middle temporal gyrus (17.4% adapted vs. 22.8% within-cohort, –5.4%) and the left parahippocampal gyrus (14.4% vs 16.2%, –1.8%).

Adapted models, while converging faster to the full adaptation set model estimation compared to within-cohort trainings, still showed instability in outlier detection at small adaptation sample sizes.

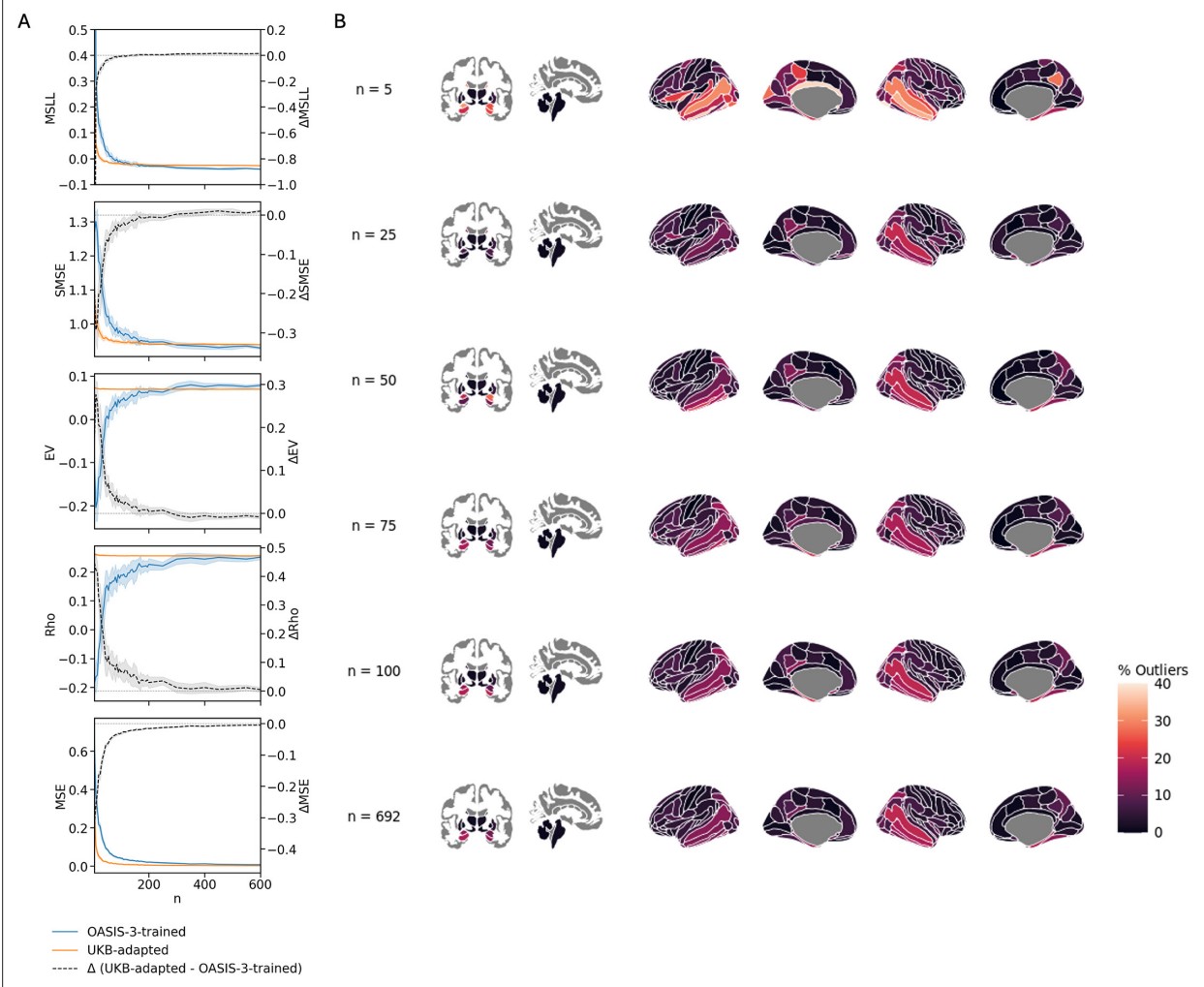

**Figure 7.** Adaptive transfer learning evaluation. (**A**) Comparison of model performance between direct training models (OASIS-3-trained) and models pretrained on UKB then adapted to OASIS-3 (UKB-adapted). The gray curve shows the difference in performance (within-cohort minus adapted), plotted on a separate *y*-axis (right). For MSLL, SMSE, and MSE, lower values indicate better performance; therefore, positive differences indicate better performance of the within-cohort models. For EV and Rho, higher values indicate better performance; hence, positive differences reflect better performance of the adapted models. EV and Rho values remain constant across models, as the adaptation procedure only modifies mean of models and does not affect the shape. (**B**) Example adaptation iteration showing the percentage of individuals identified as outliers per region. The last row shows the outlier detection obtained using the full adaptation sample (*n* = 692).

The online version of this article includes the following figure supplement(s) for figure 7:

**Figure supplement 1.** Adaptive transfer learning evaluation in AIBL.

In the example iterations in *Figure 7B*, at *n* = 25, the percentage of outliers was 7.2% in the left amygdala and 1.8% in the left hippocampus (–7.8 and –15.6%, respectively); at *n* = 50, values reached 19.2% and 9.6% (+4.2 and –7.8%). At *n* = 100, percentages were 15.0% in the left amygdala and 18.0% in the left hippocampus, closely aligning with the full model (differences under ±1%).

Similar to the direct training case, these values correspond to individual example iterations and are shown for illustration. The full replication of the previous analysis using UKB-adapted models to OASIS-3 and AIBL is provided in the Figure Supplements of *Figures 2 and 5* and Appendices 1 and 3.

## Discussion

In this study, we systematically examined how sample size and covariate distributions impact the fit and clinical utility of normative models to AD datasets. To ensure the robustness and generalizability, we leveraged a large-scale discovery sample (OASIS-3) and a replication sample (AIBL). Models were

first fitted on 80% of the HC participants from the OASIS-3, representing the optimal scenario in which the largest available dataset is leveraged to maximize accuracy and reliability. These full-sample training models served as a benchmark against which we evaluated the impact of reduced sample sizes and covariate misalignment. We systematically subsampled the full training set to vary sample sizes and manipulate covariate distributions (i.e., age and sex) and then assessed their effects on model fit in the HC test set as well as on deviation scores and outlier detection in both the HC test set and the AD cohorts. Finally, we applied the same sampling and evaluation framework to an adaptive transfer learning setting. In which pre-trained UK Biobank models were adapted using the same clinical training subsets method previously used for direct training.

Across all analyses, increasing sample size yielded more stable model fits and more consistent results. In contrast, misaligned covariate distributions, particularly with respect to age, introduced systematic distortions in model predictions and outlier detection. In the following sections, we provide a detailed analysis of how sample size and covariate distributions affected model evaluation and practical application.

Across all analyses, models trained on the full training set of available HC yielded the best overall fit, demonstrating superior performance when all data are utilized. While we acknowledge that this model does not represent an absolute ground truth, it served as a robust reference for evaluating how variations in sample size and covariate distribution affect model fit. Models trained directly on OASIS-3 achieved model fit comparable to UKB-pretrained models, indicating that training solely on the target cohort was sufficient for reliable estimation. The models fitted with the full training set also provided a strong reference for biological plausibility and interpretability, producing results consistent with established AD pathology (*Igarashi, 2023*; *Planche et al., 2022*). Specifically, in the HC test set, full training set models performed as expected, with the average *tOC* per individual around 4 in both datasets, aligning with expectations given the defined threshold for outlier detection (bottom 2.5% among 167 ROIs). In contrast, individuals with AD exhibited elevated *tOC* values, averaging at 14 in OASIS-3 and 20 in AIBL. Importantly, regions surrounding the entorhinal cortex, hippocampus, and amygdala exhibited the highest outlier percentage, highlighting their well-established involvement in AD pathology (*Igarashi, 2023*).

Subsampling analyses revealed that the largest improvements in model performance occurred up to approximately 50 subjects, at which point metrics such as the intraclass correlation coefficient (ICC) reached excellent reliability and 60–90% of total performance gains were achieved. Beyond 200 subjects, performance began to plateau with diminishing returns up to 300 and with only marginal improvements up to 600 subjects.

Sample-size requirements depend strongly on the intended use of a normative model. In the present study, we exemplified two applications commonly used in recent normative-modeling work: (1) outlier detection through regional deviation mapping across the brain and the tOC per individual (*Bhome et al., 2024*; *Floris et al., 2021*; *Loreto et al., 2024*; *Rutherford et al., 2023*; *Verdi et al., 2023*; *Verdi et al., 2024*; *Zabihi et al., 2020*); (2) case–control classification to assess whether the relative population ranks were preserved. Subsampling of real data showed that approximately 100–200 well-characterized HC were sufficient to generate clinically robust deviation maps and *tOC* values, while adding additional participants primarily reduced residual variance at a high acquisition cost. For the classification task, performance in OASIS-3 was indistinguishable from the full model with only fifteen training samples.

Sample size had no significant effect on *tOC* in the HC group, likely due to the low number of deviations expected in healthy aging. In contrast, *tOC* estimates in the AD group stabilized with increasing sample size. This was particularly evident in regions such as the hippocampus and amygdala, where outlier detection was inconsistent at lower sample sizes despite their known vulnerability in AD (*Barnes et al., 2009*; *Qu et al., 2023*). Previous work using simulated hippocampal-volume data in order to suggest that several thousand subjects are needed to stabilize the extreme lower percentiles (*Bozek et al., 2023*). In practice, achieving cohorts of that scale is hindered by high data acquisition costs, privacy constraints, and the logistical challenges of harmonizing data from multiple sites. Moreover, our clinical-validation tasks illustrated that much lower sample sizes saturate clinical readouts estimation. Our results, therefore, do not prescribe a universal threshold; rather, they provide a reference for the precision that can reasonably be expected from sample sizes commonly attainable in clinical studies. Roughly 200–300 participants emerge as a pragmatic baseline for deviation mapping

in aging cohorts, while larger data sets are strongly recommended for accurate percentile screening. Notably, broader lifespan applications may require larger samples to ensure coverage across the full age range. This is further supported by the coverage analysis (*Figure 3—figure supplement 12*), which shows that under representative sampling, the point of full age coverage closely coincides with the saturation of model fit and stability metrics. Rather than proposing a universal sample size threshold, we therefore encourage readers to perform learning-curve analyses, complemented by age coverage assessments, in their own datasets to empirically assess when performance approaches saturation for their specific age range and population.

This is further supported by our analysis of age-skewed distributions, which highlights the importance of aligning the training cohort with the target population. Indeed, beyond sample size, skewed age distributions affected both models' fit and the accuracy of deviation scores. Deviation scores were systematically overestimated in older individuals when models were trained on younger (left-skewed) samples and underestimated in younger individuals when trained on older (right-skewed) samples. These patterns align well with well-documented age-related brain changes, including cortical thinning and subcortical atrophy with increasing age (*Lemaitre et al., 2012*). Left-skewed samplings inflated normative means, leading to more pronounced negative deviations in older individuals, whereas right-skewed samplings lowered references, underestimating deviations in the young. As also observed in the simulation study by *Bozek et al., 2023*, errors were most pronounced at the extremes of the age distribution, where data were sparse. Centile analyses confirmed that models performed reliably across well-sampled age ranges but produced unstable estimates at underrepresented ages, even under representative sampling. These edge effects persisted despite covariate alignment and were attenuated with larger sample sizes.

Our findings highlight the critical importance of age distribution in clinical applications. When the age range of the training data failed to align with that of the target population, deviation estimates became systematically biased, affecting both global outlier counts and regional detection patterns. This was most notably under the left-skewed sampling, where younger-biased training sets inflated deviation scores across both HC and AD groups, with pronounced effects in regions such as the hippocampus and amygdala. Importantly, these biases were not fully resolved until relatively large sample sizes were reached. While error under right-skewed sampling gradually diminished with increasing sample size, overestimation persisted even at large sample sizes in left-skewed distributions.

Importantly, apparent increases in HC–AD separation in total outlier count should not be interpreted as evidence of superior model quality. Age-mismatched training can rescale deviation magnitudes and inflate tOC in specific subgroups without improving true case–control separability, as shown by classification task (*Figure 5C*). Model fit metrics and outlier-based measures, therefore, capture complementary but distinct aspects of normative model behavior and should be interpreted jointly rather than in isolation.

The left-skewed sampling had overall a greater effect than right-skewed sampling in both model evaluation and clinical validation, likely due to (1) the dataset's original bias toward older individuals, making younger-skewed samples less representative, and (2) the older age structure of the AD population, which exacerbates mismatch when younger HC are used to calibrate models in the clinical population. This asymmetry is also reflected in the coverage analysis, where left-skewed sampling resulted in poorer age coverage of the target population at the same sample size (*Figure 3—figure supplement 12*). Although we generated skewed distributions artificially, they mirror real-world imbalances frequently observed in neuroimaging studies, including in our own dataset, where recruitment constraints often result in age-biased cohorts (*Bethlehem et al., 2022*; *LeWinn et al., 2017*).

Together, these findings underscore the need for careful covariate matching, when developing normative models for clinical application, not only in terms of sample size but also distributional structure. This is particularly important in disorders such as AD, where age is tightly linked to disease progression and anatomical change. Deviations from a representative age distribution can propagate errors that directly affect individual-level interpretation. However, recruiting very old healthy individuals is inherently challenging. With increasing age, clinical and biological heterogeneity also increases, due in part to the greater inter-individual variability in brain structure (*Fjell and Walhovd, 2010*; *Lemaitre et al., 2012*), accumulation of comorbidities, and the difficulty of confidently excluding preclinical or undiagnosed conditions. Additionally, selective survival effects may introduce sampling

biases, further complicating the construction of representative reference cohorts in the oldest age ranges.

Given the higher prevalence of AD in women and documented sex-related differences in brain anatomy (*Ferretti et al., 2018*; *Küchenhoff et al., 2024*; *Riedel et al., 2016*; *Ruigrok et al., 2014*), we examined whether imbalances in sex distribution influenced model fit or deviation scores. Overall, sex imbalances had minimal effects in both model evaluation and clinical validation. Model fit remained stable, and only small improvements in deviation scores accuracy were observed. Deviation score accuracy was higher for individuals whose sex was more represented in the training data, and the strength of this association increased with the degree of sex imbalance. While sex imbalances had little impact on global model performance, some interaction effects with diagnosis were significant in deviation scores and outlier detection, particularly under extreme sex ratios. However, these effects were small and inconsistent, indicating limited practical impact overall. This limited sensitivity suggests that sex-related anatomical variation may generalize more robustly than age-related changes, especially in aging populations. Moreover, unlike age, which introduces continuous, nonlinear variability, sex is a discrete covariate, which makes it easier to model. These findings indicate that moderate sex imbalances are unlikely to affect overall model performance, although sex-matched training data may still be beneficial for improving individual-level assessments.

Our findings highlight the potential of adaptive transfer for efficient model fitting in clinical cohorts if large datasets are available. Normative models pre-trained on the external large-scale UKB dataset rapidly achieved performance levels comparable to models trained directly within the cohort, requiring substantially fewer individuals. Evaluation metrics plateaued around 50 participants for adapted models, whereas within-cohort models required approximately 200 to achieve similar fit. This suggests that pre-trained models can be effectively recalibrated with modest sample sizes, offering practical advantages in clinical or low-resource contexts where large datasets are often unavailable. Notably, this stabilization threshold of 50 participants was higher than the 25 participants reported by *Gaiser et al., 2023*, who used Hierarchical Bayesian Regression (HBR). HBR's hierarchical structure allows information sharing across sites, which may facilitate convergence with fewer samples. In contrast, our study used BLR, a method that does not account for cross-site dependencies and may require more data to accommodate population heterogeneity. Nevertheless, BLR remains widely used in recent normative modeling applications (*Bhome et al., 2024*; *Corrigan et al., 2024*; *Duara and Barker, 2022*; *Fraza et al., 2025*; *Holz et al., 2023*; *Maccioni et al., 2025*; *Meijer et al., 2024*; *Rutherford et al., 2022b*) because of its computational efficiency, ease of implementation, and flexibility. Another factor likely contributing to this difference in stabilization thresholds is the demographic composition of the studied populations. *Gaiser et al., 2023* examined a younger, developmentally homogeneous cohort (ages 6–17), whereas our sample spanned older adults (44–82 years), a range characterized by greater inter-individual variability in brain structure (*Fjell and Walhovd, 2010*; *Lemaitre et al., 2012*). This increased heterogeneity may necessitate larger adaptation sets to achieve stable and reliable deviation estimates.

From a methodological perspective, the choice between warped BLR and HBR should primarily be guided by the structure of site effects and by computational constraints. HBR explicitly models site-level variation through hierarchical random effects, enabling information sharing across sites and supporting federated-learning implementations in which site-specific updates can be combined without sharing raw data (*Bayer et al., 2022*; *Kia et al., 2021*; *Maccioni et al., 2025*). This structure provides more stable estimates when site-specific sample sizes are small or acquisition differences are substantial. In contrast, Warped BLR treats site as a fixed-effect covariate when site adjustment is required and does not implement hierarchical pooling, but offers simpler inference and substantially lower computational cost while accommodating non-Gaussian data distributions through the warping transformation (*Fraza et al., 2021*). These properties make Warped BLR practical in settings where site heterogeneity is limited or adequately controlled, whereas HBR may be preferable in strongly multi-site contexts or when federated learning is required for privacy-preserving data integration.

While evaluation metrics indicated convergence in overall model performance, differences remained in outlier detection between adapted and within-cohort models. In several regions commonly affected in AD, adapted models consistently identified fewer outliers, even at the largest adaptation sample size. These discrepancies suggest that, although adaptation effectively aligns prediction accuracy, subtle differences in the underlying normative distributions may persist and influence deviation-based

measures. Furthermore, the age distribution of the adaptation set continued to influence both model fit and outlier detection, despite the pre-trained UKB model already covering the same age range extensively. In contrast, sex distribution had minimal influence on the adaptation outcomes.

Overall, these results support the use of adaptive transfer as a resource-efficient alternative to full retraining, particularly when pre-trained models are available for the targeted demographics.

Some limitations should be considered. This study relied on datasets in which adults of European ancestry were overrepresented. Although the UK Biobank, which was used to pre-train the model, provided a robust reference distribution for the targeted populations of OASIS-3 and AIBL in this study, expanding reference datasets to include populations with more diverse socioeconomic and ethnic backgrounds would further improve representativeness and ensure broader applicability of normative models (*Harnett et al., 2024*). In addition, the observed stabilization of model performance around 200–300 participants was evaluated within the specific age ranges and cohorts examined here and may shift in broader lifespan settings or in populations with different sources of biological variability. Moreover, we did not simultaneously manipulate age and sex distributions in the training and adaptation sets. Exploring these factors jointly would have required a substantially more complex experimental design and statistical modeling, as their interactions could introduce confounding effects that are difficult to disentangle within the current framework. Finally, our analysis was restricted to cortical thickness for cortical regions and volumes for subcortical structures. Thickness was preferred in the cortex as it is more sensitive to age-related cellular alterations (*Lemaitre et al., 2012*), while volumetric measures were used for subcortical regions, where the absence of a clear laminar organization and current MRI resolution limit reliable thickness estimation (*Dima et al., 2022*; *Lemaitre et al., 2012*). Future work could incorporate additional structural or microstructural metrics to enhance the characterization of brain changes and improve the applicability of normative models across conditions.

Collectively, our results indicate that the benefits of enlarging a reference cohort depend critically on how closely its demographic profile mirrors that of the clinical sample. Gains from additional participants plateau when age alignment is poor, whereas even moderate-sized cohorts perform well when demographic concordance is preserved. This nuance underscores the importance of balancing sample size with representativeness when assembling reference datasets for normative modeling in aging and neurodegeneration research. Our findings provide practical guidelines for maximizing the value of moderately sized but deeply phenotyped datasets, thereby reducing costs and limiting the need for additional large-scale data collection without compromising accuracy.

## Methods

### Data

#### Cohorts

We used data from three publicly available datasets: two clinical datasets including individuals diagnosed with AD and HC, namely the Open Access Series of Imaging Studies-3 (OASIS-3) and the Australian Imaging Biomarkers and Lifestyle Study of Ageing (AIBL) (*Fowler et al., 2021*) for replication. In addition, the UKB (*Sudlow et al., 2015*) was used for pre-training of normative models for the adaptive transfer learning experiments. In the following, we provide more details on each dataset (see also *Tables 1 and 4*).

#### OASIS-3

The OASIS-3 sample included 1098 participants from the United States. For the present project, we included participants from the OASIS-3 whose ages overlapped with the UKB age range, had available T1-weighted MRI scans, passed quality control (QC) checks, and were diagnosed with either AD or identified as HC. Only baseline scans were included to ensure independence across observations, and individuals with mild cognitive impairment or other dementia-related diagnoses were excluded. The final OASIS-3 sample consisted of 1,032 individuals aged from 45 to 82 years (58% female), including 865 HC participants and 167 diagnosed with AD (*LaMontagne et al., 2019*). Image acquisitions were performed at a single site. Detailed acquisition protocols are provided in here. Ethics approval was obtained from the institutional human research ethics committees of Austin Health, St Vincent's

Health, Hollywood Private Hospital, and Edith Cowan University, and all participants provided written informed consent.

## AIBL

AIBL is a longitudinal Australian cohort study comprising 2359 participants. Data was collected by the AIBL study group. AIBL study methodology has been reported previously (*Ellis et al., 2009*). Participants were included if they had available T1-weighted MRI scans that passed QC checks (see MRI pre-processing and ROI extraction section), were aged 44–82 years (overlapping with the UKB range), and were classified as HC or diagnosed with AD. Individuals with other dementia-related diagnoses were excluded, and only baseline time points were retained, consistent with OASIS-3. The final AIBL sample consisted of 462 individuals aged 58–82 years (58% female), including 403 HC and 60 AD participants. MRI data were acquired at two sites; acquisition protocols are described in detail elsewhere (*Ayton et al., 2017*; *Dima et al., 2022*). Ethics approval was obtained from the institutional human research ethics committees of Austin Health, St Vincent's Health, Hollywood Private Hospital, and Edith Cowan University, and all participants provided written informed consent.

## UKB

The UKB is a large-scale health and imaging databank of 500,000 participants, from which 42,747 healthy individuals aged 44–82 years (53% female) were selected based on the availability of T1-weighted MRI scans and their QC assessment (see MRI pre-processing and ROI extraction section for more details on QC and preprocessing steps of all datasets). Only baseline time points were included when longitudinal data were available. MRI data were collected at three different sites using harmonized protocols. The detailed acquisition protocols are provided in https://www.fmrib.ox.ac.uk/ukbiobank/protocol/. All participants provided full informed consent, and the study received ethics approval from the National Health Service National Research Ethics Service (Ref. 11/NW/0382). This research was conducted using the UKB resource under application number 96841.

## MRI pre-processing and ROI extraction

All datasets underwent processing using FreeSurfer software, with QC performed primarily after segmentation (*Fischl, 2012*). The standard FreeSurfer pipeline included intensity normalization, skull stripping, and segmentation of gray matter (GM) and white matter (WM) surfaces, ultimately producing a tessellated representation of the GM/WM boundary. These surfaces were then corrected for topological errors and aligned to a spherical atlas based on individual cortical folding patterns. Cortical thickness was defined as the shortest distance between the vertices of the GM/WM boundary and the pial surface, and cortical thickness was measured for each individual using the Destrieux parcellation scheme, which included 74 regions per hemisphere (*Destrieux et al., 2010*). For a comprehensive overview of the pipeline and its documentation, refer to the official FreeSurfer website https://surfer.nmr.mgh.harvard.edu/ (https://surfer.nmr.mgh.harvard.edu/).

In this study, we focused on cortical thickness and on subcortical volumes. Cortical thickness was included as it is a sensitive biomarker of age-related atrophy, particularly in the parietal cortex, and provides valuable insights into neurodegeneration in AD (*Lemaitre et al., 2012*; *McGinnis et al., 2011*). Cortical volumes, which account for both cortical folding and thickness, may be less sensitive to specific cellular changes such as neuronal, dendritic, and synaptic changes (*Lemaitre et al., 2012*). Subcortical volumes were also considered, as AD is known to particularly affect the entorhinal cortex and surrounding regions (*Qu et al., 2023*; *Yan et al., 2019*; *Zhao et al., 2019*). These structures lack the laminar organization of the cortex, making thickness estimates unreliable. As a result, volumetric measures are typically used for subcortical regions (*Dima et al., 2022*; *Lemaitre et al., 2012*).

For labeling subcortical tissue classes, the volume-based stream was utilized. After affine registration to MNI305 space, initial volumetric labeling, and B1 bias field intensity correction, a high-dimensional nonlinear volumetric alignment to the MNI305 atlas was performed, with the volume labeled according to the ASEG atlas (*Fischl et al., 2002*). From this atlas, we retained 19 subcortical regions.

## Quality control

QC procedures were implemented across all datasets. For the OASIS-3 and AIBL datasets, the QC process was performed internally by our research team (BHV and CE). We used the quality assessment tools bundled with FreeSurfer which included several automated QC measures, such as signal-to-noise ratio, topological defects, Euler number, and rotation parameters, to identify potential issues with MRI data. For each scan flagged by these measures, we conducted a thorough visual inspection. Scans with poor contrast, orientation, or significant artifacts were excluded. Consequently, 39 scans were excluded from the OASIS-3 dataset, while no exclusion was necessary for the AIBL dataset. For the UKB, QC was conducted as part of the original data release using a semi-automated pipeline. T1-weighted images were scored using an automated classifier, with manual review performed for cases close to the 'bad data' threshold. Data deemed seriously problematic were excluded. Free-Surfer outputs were also QC checked using the Qoala-T approach (*Klapwijk et al., 2019*), supplemented by manual review, ensuring only high-quality data were used. For more details, refer to the documentation available at here.

## Normative modeling

### Estimation of normative models

To capture individual deviations from typical aging patterns, normative models were fitted independently within each dataset (OASIS-3 and AIBL) for each ROI (*Figure 1A*). Each dataset was stratified by site, sex, and age (5-year bins), and split into a training set (80% of HC; $n = 692$ for OASIS-3, $n = 322$ for AIBL) and a test set (20% of HC plus all AD participants; $n = 174$ HC, $n = 167$ AD for OASIS-3, and $n = 81$ HC, $n = 60$ AD for AIBL) (*Table 1*).

We employed Warped BLR with a B-spline basis expansion, using Python 3.9.16 and PCNtoolkit version 0.29 (*Fraza et al., 2021*). All procedures followed the recommendations for BLR models, as detailed in *Rutherford et al., 2022b*. The covariates (age, sex, and a dummy-coded site variable) were incorporated into the model. Age and sex were modeled using a cubic B-spline basis with three evenly spaced knots, allowing for the creation of a smooth curve that can flexibly represent the changes in brain structure across different ages and sexes. The Sinh–Arcsinh function and the respective warped likelihood were used to handle non-Gaussian distributions in the data (*Fraza et al., 2021*; *Jones and Pewsey, 2009*) and optimized the model by minimizing the negative log-likelihood using Powell's conjugate direction method.

From normative models, deviation scores (*Z*-scores) were derived to quantify how much an individual's measurement deviates from the predicted normative range. These scores were computed as the difference between the observed value and the model's predicted conditional mean, scaled by the estimated variance. Unlike traditional *Z*-scores, which assume a fixed population variance, normative modeling *Z*-scores account for two sources of uncertainty commonly defined in machine learning: (1) aleatoric uncertainty (irreducible uncertainty), which captures true underlying variability across individuals that cannot be reduced with more data; and (2) epistemic uncertainty (reducible uncertainty), which reflects parameter uncertainty due to limited data and can be minimized as more data becomes available (*Fraza et al., 2021*; *Marquand et al., 2016*). By integrating these components, *Z*-scores provide a standardized measure of deviation, indicating whether an individual's measurement falls within or outside the normative range and how atypical it is relative to the reference population.

## Subsampling

We first fitted models for each ROI using 80% of the HC participants from OASIS-3 or AIBL for the replication analysis. These models represent the optimal scenario, leveraging the maximum available data to achieve the highest accuracy and reliability. These models trained on the full sample then served as a reference benchmark for evaluating the impact of reduced sample sizes and covariate misalignment. To assess how systematic variations in (1) sample size and (2) covariate distributions influenced model fit performance, we applied a series of subsampling strategies to generate training sets with varying sizes and age or sex distributions. For each subsampling condition, a set of individuals was selected to form the training set, which was then used to fit separate models for each ROI. This approach allowed us to systematically assess how variations in sample composition affected model fit across brain regions. Across all subsampling strategies, the test set remained identical (described in Estimation of

normative models), ensuring comparability. All primary results are presented for the OASIS-3 dataset, with replication in the AIBL dataset detailed in the corresponding Figure Supplements.

## Sample size variations

Subsamples were drawn in increments of 5 from size 5–200, and in increments of 50 beyond 200 up to 600. Finer steps at smaller sample sizes allowed closer track of changes in model performance at smaller sample sizes, while larger steps were applied beyond 200 to reduce computational demands, after convergence was expected. For each sample size, 10 random iterations were performed to estimate variability and reduce the influence of individual draws. For replication in AIBL, samples were drawn from 5 to 100 per site, due to more limited available data.

## Subsampling with different covariates distributions

We assessed the influence of age and sex distributions in the training set by using a representative age distribution with balanced sex ratios and then altering either age distribution (using skewed subsampling) or sex balance (using uneven ratios of females and males) while also systematically varying sample size as described above.

To generate subsamples that mimicked the original age distributions of the data, we partitioned the age distribution into 10 bins using a quantile discretizer, which ensured equal numbers of participants in each bin. An equal number of males and females (±1) was then randomly selected across bins. This approach, referred to as *representative sampling*, ensured that the subsampled training sets closely matched the original age distribution of the dataset while maintaining a balanced sex ratio.

To create training sets with skewed (non-representative) age distributions, we applied two strategies that preferentially selected either younger (left-skewed) or older (right-skewed) individuals. More specifically, the age distribution was divided into 10 equally spaced age bins, using a uniform discretizer (which divides the age range into equal-width intervals but does not ensure an equal number of participants per bin as in the representative subsampling). Samples were randomly drawn from these bins, and a beta distribution was applied to weight the probability of drawing samples from each bin. For the left-skewed sampling, we used parameters $\alpha = 2$ and $\beta = 5$, increasing the likelihood of selecting younger samples. Conversely, for the right-skewed sampling, parameters $\alpha = 5$ and $\beta = 2$ were used, concentrating more samples in the older age bins.

To examine the influence of sex imbalance, we adapted the representative age sampling strategy but altered the male-to-female ratio according to predefined levels (1:4; 1:10; 4:1, 10:1). This enables assessment of how sex distribution imbalances affected model performance and clinical interpretation of deviations.

Age distributions were summarized separately for males and females in the original training and test sets (*Appendix 4—table 1*) and the expected age distributions resulting from the skewed-age sampling and the sex-imbalance sampling procedures were obtained by repeated simulations at a fixed sample size and are reported in *Appendix 4—tables 2 and 3*.

## Age-distribution coverage between sampled training sets and test cohorts

To quantify how well each sampling strategy aligned the age distribution of the sampled training sets with the empirical age distributions of the HC test cohorts, we computed an age-bin coverage metric based on distribution intersection. Age was discretized into 20 quantile-based bins using the full training set of each dataset (OASIS-3 and AIBL) as reference.

For each sampling strategy (Representative, Left-skewed, Right-skewed), sample size, and dataset, we generated 1000 independent training samples using the same sampling procedures as in the main analyses. For each sampled training set, age-bin count distributions were computed and compared to the corresponding HC test-set age-bin counts.

Coverage was defined as:

$$\text{Coverage} = \frac{\sum\limits_{i} \min\left(n_{\text{train}}\left(i\right), n_{\text{test}}\left(i\right)\right)}{\sum\limits_{i} n_{\text{test}}\left(i\right)},$$

where $i$ indexes age bins, $n_{\text{train}}$ and $n_{\text{test}}$ are the numbers of individuals in bin i in the sampled training set and HC test set, respectively. This metric quantifies the fraction of the test-set age distribution that is 'covered' by the sampled training set and ranges from 0 (no test-set ages covered) to 1 (complete coverage of the test-set age distribution). For each condition, the mean and standard deviation of the coverage across repetitions were computed.

This analysis quantifies how the imposed sampling-induced age profiles align with the empirical age structure of the target populations and how this alignment evolves with increasing sample size.

## Model evaluation

Subsequently, models trained on subsampled data were evaluated on the same fixed test set of participants, assessing model fit in the HC test set and Z-score errors in both HC and AD test sets (*Figure 1C*). The performance of the subsampled models was then compared with that of the models trained on the full training set across three subsampling scenarios: (1) representative age distribution with balanced sex ratios, (2) skewed age subsampling with balanced sex ratios, and (3) representative age sampling with unbalanced sex ratios.

### Evaluation of normative models fitting

Every derived model was evaluated using state-of-the-art model evaluation metrics including: MSLL, SMSE, EV, and Rho (*Fraza et al., 2021*; *Rutherford et al., 2022b*) and *ICC* (*Koo and Li, 2016*). Evaluation was restricted to HC individuals, as model fit is intended to assess performance relative to the reference population. The AD cohort was instead considered for Z-score errors and clinical validation. *MSLL*, standardized by the mean loss of the training dataset (*Bayer et al., 2022*; *Rasmussen and Williams, 2008*), indicates the predictive performance of a model compared to the mean of the training set, with more negative values being preferable as they indicate better predictive accuracy. *SMSE* assesses the average squared differences between the predicted and actual values, standardized by the variance of the test data. Lower *SMSE* values indicate better predictive performance, as they signify smaller errors. *EV* measures the proportion of variance in the test data that is explained by the model. Higher values of EV indicate that the model accounts for a greater portion of the variability in the data, reflecting better model performance. *Rho* measures the linear correlation between the predicted and actual values. A Pearson correlation close to 1 indicates a strong positive correlation, implying that the model's predictions are closely aligned with the actual values. The *ICC* quantifies the reliability of the models' outputs across iterations. The *ICC* was computed on the *Z*-scores of HC test set for each model using a two-way random-effects model for single measurements, with absolute agreement, to assess the consistency of *Z*-scores across iterations (where iterations refer to repeated random sampling of the same sample size within a subsampling strategy). In addition, calibration was assessed by estimating the percentage of HC test individuals falling outside the nominal 2.5–97.5% prediction interval, separately for the lower (below 2.5%) and upper (above 97.5%) tails.

### Evaluation of Z-score estimations

Z-score errors were computed for each sampling strategy and sample size by comparing models trained on subsampled data to the ones trained on the full training set. We used *MSE* to summarize overall model accuracy of estimates by averaging the squared deviations across ROIs for each individual. *MSE* captures the overall magnitude of errors, penalizing larger deviations more strongly due to the squaring operation, which makes it well suited for assessing general model accuracy at the individual level. To reduce the influence of rare iterations producing atypical very large errors, outliers were excluded using a conservative interquartile range threshold (3 × IQR) to exclude only extreme values while preserving most of the data. Age effects were evaluated using cubic regressions across the age range; sex effects were assessed by comparing *MSEs* between females and males under imbalanced sampling.

We additionally computed *MBE* to assess the direction of Z-score differences, indicating whether estimates tend to be systematically over- or underestimated. Because opposing errors can cancel each other out, *MBE* may underestimate the overall level of error. *MSE* and *MBE* together provide complementary information: *MSE* captures the total error magnitude, while *MBE* reveals directional biases.

## Clinical validation

To evaluate clinical validity, we assessed whether deviation patterns aligned with known AD-related neurobiological evidence and whether they could reliably distinguish HC from AD. Specifically, we examined whether the most deviating regions (i.e., outliers) corresponded to established patterns of AD pathology (i.e., biological plausibility) and whether classification based on deviation scores could distinguish HC from AD. For both analyses, we compared the performance of the model fitted with the full training set with subsampled training sets models to determine whether deviation patterns and classification performance remained consistent across sampling strategies and sample sizes.

### Outlier detection and biological plausibility

To assess biological plausibility, we computed the percentage of outliers per ROI in the test set to assess the consistency and group-specific patterns of deviations in HC and AD, across the model trained on the full set and those trained on each subsampled set. A data point was defined as an outlier if its deviation score was $Z < -1.96$. This threshold, representing the bottom 2.5% of the normative range, has been commonly used to report extreme deviations in previous works (*Bhome et al., 2024*; *Loreto et al., 2024*; *Verdi et al., 2023*; *Verdi et al., 2024*). Only negative outliers are considered here, as AD is mainly characterized by diffuse, progressive atrophy of brain regions (*Pini et al., 2016*; *Sabuncu et al., 2011*; *Ten Kate et al., 2018*). Using this framework, a properly fitted model is expected to show outlier distributions that align with known neuroanatomical patterns of AD, with regions commonly affected by the disease exhibiting higher outlier counts.

To further capture the overall trends in outlier detection, we analyzed the mean *tOC* per individual in both HC and AD groups. The *tOC*, defined as the sum of ROI with negative outliers for each participant (*Loreto et al., 2024*; *Verdi et al., 2021*; *Verdi et al., 2023*; *Verdi et al., 2024*), was computed within each group (i.e., HC and AD) for every sample size and subsampling iteration.

### Classification performance

We evaluated classification performance to determine whether deviations from normative models (i.e., *Z*-scores) retained their clinical validity and preserved the relative ranking of individuals for each sampling configuration. SVCs were trained using the ROI-level deviation scores from the test set. Classification was performed using 10-fold cross-validation and the *receiver operating characteristic – area under the curve* (*ROC-AUC*) was computed for each fold. For each sampling configuration, *ROC-AUC* values were averaged across folds and iterations to obtain a robust measure of classification performance.

## Statistical analysis

We used linear mixed-effects models (LMMs) to evaluate the effects of sample size and subsampling strategy on model fit metrics (*MSLL, EV, SMSE, Rho, ICC, the lower and upper tail HC%*) and deviation scores (*MSE* and *tOC*) (i.e., outcome). Subsampling strategies included age-skewed (left-skewed, right-skewed) and sex-imbalanced (1:4, 1:10, 4:1, 10:1 female-to-male ratios). The representative sampling strategy, which preserved the original age and sex distribution, served as the reference.

For model fit, LMMs included sample size (*n*), sampling strategy, and their interaction as fixed effects, with a random intercept for ROI to account for variability across regions:

$$\text{Outcome} \sim \log(n) \times \text{Sampling Strategy} + (1 \mid \text{ROI})$$

The relationship between sample size and model performance is characterized by steep improvements at small *n* that gradually level off with larger samples (*Figures 2 and 3*). To capture this non-linear pattern and avoid overemphasizing differences at high *n*, we applied a logarithmic transformation to sample size. This transformation linearizes the effect, resulting in more appropriate estimates of the influence of *n* and its interactions with sampling strategies. Both performance metrics and sample size were standardized prior to modeling to facilitate comparability.

For deviation scores, LMMs additionally included group (HC or AD) and its interactions with sample size and sampling strategy as fixed effects, with a random intercept for individuals to account for subject-level variability. As before, sample size was log-transformed, and both outcomes and sample size were standardized prior to modeling:

$$\text{Outcome} \sim \log(n) \times \text{Sampling Strategy} \times \text{Group} + (1 \mid \text{Individual})$$

Again, the representative sampling strategy and HC group served as baseline levels for comparison. Fixed effects and interaction terms captured the influence of sample size, sampling strategy, and group on the outcomes.

For completion, we extended the statistical error models in Appendix 4 to explicitly include age, sex, and all higher-order interactions with diagnosis, sample size, and sampling strategy.

To illustrate the effect of sampling strategies and sample size for specific brain regions, additional LMMs were computed independently for each ROI. These results are presented in the *Figure 3— figure supplements 1 and 2* and *Figure 5—figure supplements 1 and 2* and were corrected using the false discovery rate to account for multiple comparisons.

## Adaptive transfer learning

To evaluate whether models trained on large external datasets can generalize to clinical cohorts, we additionally compared our models trained within cohorts (i.e., OASIS-3 or AIBL) with models adapted from a large independent cohort using adaptive transfer learning. To motivate the need for model adaptation across cohorts, we first examined differences in mean cortical thickness between UKB and the clinical datasets. We compared mean cortical thickness and mean subcortical gray volumes for HC from the UKB, OASIS-3, and AIBL datasets. Differences in mean cortical thickness across datasets were assessed using boxplots and bivariate density plots (mean cortical thickness vs. age). Separate linear regression models were fitted for each dataset, with mean cortical thickness as the dependent variable and age as the independent variable.

Adaptive transfer learning was applied to adjust the UKB-based normative models to the clinical cohorts (*Figure 1B*). Models were pre-fitted using 80% of UKB HC, with 20% held out for internal evaluation. Adaptation sets and within-cohort training sets were sampled from the same training cohort and tested on the same evaluation set, ensuring direct comparability between models. This approach procedure assumes that biological effects, such as age-related trajectories, are shared across datasets, and only cohort-specific shifts require adjustment. The adaptation assumes that normative patterns learned from a large reference cohort remain valid across datasets, and that residual differences, such as scanner or cohort biases, can be captured through distributional adjustments at the mean level.

To evaluate the two modeling strategies, either directly trained on the target dataset (i.e., OASIS-3 or AIBL) or adapted from a pre-trained UKB model, we compared model fit using the same metrics as previously (*MSLL, SMSE, EV, Rho, ICC, the lower and upper tail HC%*), assessed *Z*-score errors relative to the full adaptation model, and illustrated the percentage of outliers per brain region across sample sizes. To facilitate direct comparison, we also computed the difference in performance between the two approaches (direct training minus adapted) for each metric. Additionally, we replicated all analyses conducted in the direct training setting, including covariate skewing, and present the adapted versions in the Figures supplements of each corresponding main figure.

## Acknowledgements

This research was supported by the Dementia Research Switzerland – Synapsis Foundation [No. 2021-PI02] and the Swiss National Science Foundation (SNSF) under the following grants [10,001C 197480 and IC00I0 227750]. Data were provided in part by OASIS-3: Longitudinal Multimodal Neuroimaging: Principal Investigators: T Benzinger, D Marcus, J Morris; NIH P30 AG066444, P50 AG00561, P30 NS09857781, P01 AG026276, P01 AG003991, R01 AG043434, UL1 TR000448, R01 EB009352. AV-45 doses were provided by Avid Radiopharmaceuticals, a wholly owned subsidiary of Eli Lilly.

## Additional information

### Funding

| Funder | Grant reference number | Author |
| --- | --- | --- |
| Swiss National Science Foundation | 10001C_197480 | Nicolas Langer<br>Bruno Hebling Vieira |

| Funder | Grant reference number | Author |
|---|---|---|
| UZH PostDoc Grant | grant no. FK-23-086 | Bruno Hebling Vieira |
| Swiss National Science Foundation | 10001C_220048 | Nicolas Langer |
| Swiss National Science Foundation | IC00I0-227750 | Nicolas Langer |
| Dementia Research Synapsis Foundation | grant no. 2021-PI02 | Nicolas Langer |

The funders had no role in study design, data collection, and interpretation, or the decision to submit the work for publication.

## Author contributions

Camille Elleaume, Conceptualization, Data curation, Formal analysis, Validation, Investigation, Visualization, Methodology, Writing – original draft, Writing – review and editing; Bruno Hebling Vieira, Conceptualization, Data curation, Methodology, Writing – review and editing; Dorothea L Floris, Conceptualization, Methodology, Writing – review and editing; Nicolas Langer, Conceptualization, Supervision, Funding acquisition, Methodology, Project administration, Writing – review and editing

## Author ORCIDs

Camille Elleaume https://orcid.org/0009-0001-4162-320X
Bruno Hebling Vieira https://orcid.org/0000-0002-8770-7396
Dorothea L Floris https://orcid.org/0000-0001-5838-6821
Nicolas Langer https://orcid.org/0000-0002-6038-9471

## Ethics

This study used de-identified human neuroimaging data from publicly available research databases. All participants provided written informed consent, and ethical approval was obtained by the original studies from their respective institutional review boards. The present analyses were conducted in accordance with the data use agreements and ethical guidelines of each dataset, and no new data were collected for this study.

Reviewer #1 (Public review): https://doi.org/10.7554/eLife.108952.3.sa1
Reviewer #2 (Public review): https://doi.org/10.7554/eLife.108952.3.sa2
Author response https://doi.org/10.7554/eLife.108952.3.sa3

# Additional files

## Supplementary files

MDAR checklist

## Data availability

This study used data from the UK Biobank, OASIS-3, and Australian Imaging, Biomarkers and Lifestyle (AIBL) datasets. These datasets contain human participant data and are subject to ethical and legal restrictions; therefore, they are not publicly available without approval. OASIS-3 data are available to qualified researchers upon application through the Washington University Knight Alzheimer Disease Research Center (https://www.oasis-brains.org). UK Biobank data are available to bona fide researchers upon application via the UK Biobank Access Management System (https://www.ukbiobank.ac.uk). AIBL data are available to approved researchers through the AIBL data access process (https://aibl.csiro.au). Processed summary-level data underlying the figures and tables, together with the code used to generate the analyses, are publicly available at https://github.com/Elleaume/RobustNM (copy archived at *Elleaume, 2026*).

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

# Appendix 1

## Replication with adaptive transfer learning (OASIS-3)

**Appendix 1—table 1.** Linear mixed model results for evaluation metrics under age-skewed sampling conditions using normative models pre-trained on the UKB and adapted to the OASIS-3 dataset.

Models assess the influence of standardized and log-transformed sample size ($n$) and age sampling strategy (Representative, Left-skewed, Right-skewed) on model performance metrics (MSLL, SMSE, EV, Rho, ICC, the lower and upper tail HC%). All variables were standardized to allow comparison of effect sizes. Representative sampling serves as the reference level. Reported $\beta$ coefficients and corresponding p-values indicate the direction and significance of each effect.

| | MSLL $\beta$ (p-value) | SMSE $\beta$ (p-value) | EV $\beta$ (p-value) | Rho $\beta$ (p-value) | ICC $\beta$ (p-value) | Lower tail HC % $\beta$ (p-value) | Upper tail HC % $\beta$ (p-value) |
|---|---|---|---|---|---|---|---|
| Intercept (representative) | −0.195 (p < 0.001) | −0.567 (p < 0.001) | 0.644 (p < 0.001) | 0.701 (p < 0.001) | 0.504 (p < 0.001) | −0.219 (p < 0.001) | −0.373 (p < 0.001) |
| Log($n$) | −0.054 (p < 0.001) | −0.102 (p < 0.001) | 0.000 (p = 1.000) | 0.000 (p = 1.000) | 0.245 (p < 0.001) | −0.228 (p < 0.001) | −0.210 (p < 0.001) |
| Left-skewed | 0.062 (p < 0.001) | 0.220 (p < 0.001) | 0.000 (p = 1.000) | −0.000 (p = 1.000) | 0.041 (p < 0.001) | 0.186 (p < 0.001) | 0.778 (p < 0.001) |
| Right-skewed | −0.005 (p < 0.001) | 0.006 (p = 0.049) | −0.000 (p = 1.000) | −0.000 (p = 1.000) | 0.020 (p < 0.001) | 0.076 (p < 0.001) | −0.057 (p < 0.001) |
| Log($n$):Left-skewed | −0.037 (p < 0.001) | −0.019 (p < 0.001) | 0.000 (p = 1.000) | 0.000 (p = 1.000) | −0.002 (p = 0.645) | −0.010 (p = 0.133) | −0.161 (p < 0.001) |
| Log($n$):Right-skewed | 0.003 (p = 0.087) | 0.008 (p = 0.010) | −0.000 (p = 1.000) | 0.000 (p = 1.000) | −0.001 (p = 0.686) | 0.008 (p = 0.242) | 0.012 (p = 0.187) |

**Appendix 1—table 2.** Linear mixed model results for evaluation metrics under sex-imbalanced sampling conditions using normative models pre-trained on the UKB and adapted to the OASIS-3 dataset.

Models assess the influence of standardized and log-transformed sample size ($n$) and sex ratio in the training set (1F:1M, 1F:4M, 1F:10M, 4F:1M, 10F:1M; F=female, M=male) on model performance metrics (MSLL, SMSE, EV, Rho, ICC, the lower and upper tail HC%). All variables were standardized to allow comparison of effect sizes. Representative sampling (1F:1M) serves as the reference level. Reported $\beta$ coefficients and corresponding p-values indicate the direction and significance of each effect.

| | MSLL $\beta$ (p-value) | SMSE $\beta$ (p-value) | EV $\beta$ (p-value) | Rho $\beta$ (p-value) | ICC $\beta$ (p-value) | Lower tail HC % $\beta$ (p-value) | Upper tail HC % $\beta$ (p-value) |
|---|---|---|---|---|---|---|---|
| Intercept (1F1M) | −0.195 (p < 0.001) | −0.567 (p < 0.001) | 0.644 (p < 0.001) | 0.701 (p < 0.001) | 0.504 (p < 0.001) | −0.219 (p < 0.001) | −0.373 (p < 0.001) |
| Log($n$) | −0.054 (p < 0.001) | −0.102 (p < 0.001) | 0.000 (p = 1.000) | 0.000 (p = 1.000) | 0.245 (p < 0.001) | −0.228 (p < 0.001) | −0.210 (p < 0.001) |
| 10F1M | −0.011 (p < 0.001) | −0.005 (p = 0.002) | −0.000 (p = 1.000) | −0.000 (p = 1.000) | −0.000 (p = 0.919) | 0.088 (p < 0.001) | 0.059 (p < 0.001) |
| 4F1M | −0.009 (p < 0.001) | −0.006 (p < 0.001) | 0.000 (p = 1.000) | 0.000 (p = 1.000) | −0.000 (p = 0.995) | 0.050 (p < 0.001) | 0.036 (p < 0.001) |
| 1F4M | 0.002 (p = 0.031) | 0.013 (p < 0.001) | 0.000 (p = 1.000) | 0.000 (p = 1.000) | −0.002 (p = 0.595) | −0.006 (p = 0.266) | 0.036 (p < 0.001) |

*Appendix 1—table 2 Continued on next page*

*Appendix 1—table 2 Continued*

| | MSLL $\beta$ (p-value) | SMSE $\beta$ (p-value) | EV $\beta$ (p-value) | Rho $\beta$ (p-value) | ICC $\beta$ (p-value) | Lower tail HC % $\beta$ (p-value) | Upper tail HC % $\beta$ (p-value) |
|---|---|---|---|---|---|---|---|
| 1F10M | –0.001 (p = 0.593) | 0.020 (p < 0.001) | –0.000 (p = 1.000) | –0.000 (p = 1.000) | –0.005 (p = 0.138) | 0.011 (p = 0.032) | 0.036 (p < 0.001) |
| Log(*n*):10F1M | 0.004 (p < 0.001) | 0.006 (p < 0.001) | –0.000 (p = 1.000) | –0.000 (p = 1.000) | –0.004 (p = 0.280) | 0.006 (p = 0.267) | 0.001 (p = 0.855) |
| Log(*n*):4F1M | 0.004 (p < 0.001) | 0.004 (p = 0.014) | –0.000 (p = 1.000) | –0.000 (p = 1.000) | –0.003 (p = 0.366) | 0.008 (p = 0.133) | 0.000 (p = 0.970) |
| Log(*n*):1F4M | 0.000 (p = 0.975) | –0.001 (p = 0.479) | –0.000 (p = 1.000) | 0.000 (p = 1.000) | 0.001 (p = 0.825) | 0.007 (p = 0.208) | –0.046 (p < 0.001) |
| Log(*n*):1F10M | 0.006 (p < 0.001) | –0.002 (p = 0.175) | –0.000 (p = 1.000) | –0.000 (p = 1.000) | 0.004 (p = 0.192) | 0.001 (p = 0.793) | 0.015 (p = 0.001) |

**Appendix 1—table 3.** Linear mixed model results for MSE and total outlier count (tOC) under age-skewed sampling for models adapted from the UK Biobank to OASIS-3.
Models evaluate the influence of diagnosis (HC, AD), log-transformed and standardized sample size (*n*), and age sampling strategy (Representative, Left-skewed, Right-skewed) on standardized deviation score outcomes. Continuous variables were standardized to allow comparison of effect sizes. Representative sampling and HC are used as reference levels. Reported $\beta$ coefficients and corresponding p-values indicate the direction and significance of the effects.

| | MSE $\beta$ (p-value) | tOC $\beta$ (p-value) |
|---|---|---|
| Intercept (HC, Representative) | –0.313 (p < 0.001) | –0.462 (p < 0.001) |
| Log(*n*) | –0.451 (p < 0.001) | 0.026 (p = 0.015) |
| AD | 0.070 (p = 0.243) | 0.909 (p < 0.001) |
| Left-skewed | 0.930 (p < 0.001) | 0.281 (p < 0.001) |
| Right-skewed | 0.560 (p < 0.001) | –0.068 (p < 0.001) |
| Log(*n*):AD | –0.073 (p = 0.039) | 0.144 (p < 0.001) |
| Log(*n*):Left-skewed | –0.359 (p < 0.001) | –0.142 (p < 0.001) |
| Log(*n*):Right-skewed | –0.521 (p < 0.001) | 0.012 (p = 0.427) |
| Left-skewed:AD | 0.211 (p < 0.001) | 0.670 (p < 0.001) |
| Right-skewed:AD | –0.061 (p = 0.224) | –0.192 (p < 0.001) |
| Log(*n*):Left-skewed:AD | –0.030 (p = 0.555) | –0.187 (p < 0.001) |
| Log(*n*):Right-skewed:AD | 0.056 (p = 0.264) | 0.005 (p = 0.819) |

**Appendix 1—table 4.** Linear mixed model results for MSE and total outlier count (tOC) under sex-imbalanced sampling for models adapted from the UK Biobank to OASIS-3.
Models evaluate the influence of diagnosis (HC, AD), log-transformed and standardized sample size (*n*), and sex ratio in the training set (1:1, 1F:4M, 1F:10M, 4F:1M, 10F:1M) on standardized deviation score outcomes. Continuous variables were standardized to allow comparison of effect sizes. The 1:1 ratio and HC are used as reference levels. Reported $\beta$ coefficients and corresponding p-values indicate the direction and significance of the effects.

| | MSE<br>$\beta$ (p-value) | tOC<br>$\beta$ (p-value) |
|---|---|---|
| Intercept (HC, 1 F 1 M) | NA | –0.443 (p < 0.001) |
| Log($n$) | 0.010 (p < 0.001) | 0.029 (p < 0.001) |
| AD | 0.003 (p = 0.168) | 0.016 (p < 0.001) |
| 10F1M | 0.010 (p < 0.001) | –0.003 (p = 0.202) |
| 4F1M | 0.014 (p < 0.001) | 0.003 (p = 0.119) |
| 1F4M | NA | 0.576 (p < 0.001) |
| 1F10M | 0.002 (p = 0.576) | 0.055 (p < 0.001) |
| Log($n$):AD | 0.001 (p = 0.809) | 0.033 (p < 0.001) |
| Log($n$):10F1M | –0.000 (p = 0.964) | –0.012 (p < 0.001) |
| Log($n$):4F1M | 0.001 (p = 0.701) | –0.007 (p = 0.011) |
| Log($n$):1F4M | –0.142 (p < 0.001) | –0.075 (p < 0.001) |
| Log($n$):1F10M | 0.004 (p = 0.052) | 0.003 (p = 0.185) |
| 10F1M:AD | 0.005 (p = 0.010) | 0.003 (p = 0.100) |
| 4F1M:AD | –0.003 (p = 0.106) | 0.004 (p = 0.060) |
| 1F4M:AD | 0.001 (p = 0.628) | 0.002 (p = 0.404) |
| 1F10M:AD | –0.005 (p = 0.014) | –0.044 (p < 0.001) |
| Log($n$):10F1M:AD | –0.002 (p = 0.604) | 0.002 (p = 0.558) |
| Log($n$):4F1M:AD | –0.001 (p = 0.749) | 0.003 (p = 0.260) |
| Log($n$):1F4M:AD | 0.002 (p = 0.560) | 0.006 (p = 0.025) |
| Log($n$):1F10M:AD | –0.001 (p = 0.817) | 0.002 (p = 0.514) |

# Appendix 2

## Replication of direct training in independent dataset (AIBL)

**Appendix 2—table 1.** Linear mixed model results for evaluation metrics under age-skewed sampling conditions using normative models trained on AIBL dataset.

Models assess the influence of standardized and log-transformed sample size ($n$) and age sampling strategy (Representative, Left-skewed, Right-skewed) on model performance metrics (MSLL, SMSE, EV, Rho, ICC, the lower and upper tail HC%). All variables were standardized to allow comparison of effect sizes. Representative sampling serves as the reference level. Reported $\beta$ coefficients and corresponding p-values indicate the direction and significance of each effect.

| | MSLL $\beta$ (p-value) | EV $\beta$ (p-value) | SMSE $\beta$ (p-value) | Rho $\beta$ (p-value) | ICC $\beta$ (p-value) | Lower tail HC % $\beta$ (p-value) | Upper tail HC % $\beta$ (p-value) |
|---|---|---|---|---|---|---|---|
| Intercept (Representative) | –0.293 (p < 0.001) | –0.332 (p < 0.001) | 0.175 (p < 0.001) | 0.067 (p = 0.181) | 0.100 (p < 0.001) | –0.163 (p = 0.004) | –0.026 (p = 0.626) |
| Log($n$) | –0.426 (p < 0.001) | –0.308 (p < 0.001) | 0.262 (p < 0.001) | 0.219 (p < 0.001) | 0.755 (p < 0.001) | 0.103 (p < 0.001) | –0.307 (p < 0.001) |
| Left-skewed | 1.080 (p < 0.001) | 1.382 (p < 0.001) | –0.517 (p < 0.001) | –0.212 (p < 0.001) | 0.043 (p = 0.011) | 1.096 (p < 0.001) | –0.457 (p < 0.001) |
| Right-skewed | 0.465 (p < 0.001) | 0.438 (p < 0.001) | –0.338 (p < 0.001) | –0.127 (p < 0.001) | –0.052 (p = 0.002) | –0.265 (p < 0.001) | 0.562 (p < 0.001) |
| Log($n$):Left-skewed | –0.484 (p < 0.001) | –0.670 (p < 0.001) | 0.196 (p < 0.001) | 0.002 (p = 0.897) | –0.208 (p < 0.001) | –0.555 (p < 0.001) | 0.182 (p < 0.001) |
| Log(n):Right-skewed | –0.411 (p < 0.001) | –0.284 (p < 0.001) | 0.072 (p < 0.001) | 0.006 (p = 0.703) | 0.002 (p = 0.912) | 0.047 (p = 0.010) | –0.572 (p < 0.001) |

**Appendix 2—table 2.** Linear mixed model results for evaluation metrics under sex-imbalanced sampling conditions using normative models trained on AIBL dataset.

Models assess the influence of standardized and log-transformed sample size ($n$) and sex ratio in the training set (1F:1M, 1F:4M, 1F:10M, 4F:1M, 10F:1M; F = female, M = male) on model performance metrics (MSLL, SMSE, EV, Rho, ICC, the lower and upper tail HC%). All variables were standardized to allow comparison of effect sizes. Representative sampling (1F:1M) serves as the reference level. Reported $\beta$ coefficients and corresponding p-values indicate the direction and significance of each effect.

| | MSLL $\beta$ (p-value) | SMSE $\beta$ (p-value) | EV $\beta$ (p-value) | Rho $\beta$ (p-value) | ICC $\beta$ (p-value) | Lower tail HC % $\beta$ (p-value) | Upper tail HC % $\beta$ (p-value) |
|---|---|---|---|---|---|---|---|
| Intercept (1F1M) | –0.294 (p < 0.001) | –0.333 (p < 0.001) | 0.177 (p < 0.001) | 0.067 (p = 0.285) | 0.100 (p < 0.001) | –0.163 (p < 0.001) | –0.026 (p = 0.635) |
| Log($n$) | –0.423 (p < 0.001) | –0.306 (p < 0.001) | 0.258 (p < 0.001) | 0.219 (p < 0.001) | 0.755 (p < 0.001) | 0.103 (p < 0.001) | –0.307 (p < 0.001) |
| 10F1M | 0.107 (p < 0.001) | 0.118 (p < 0.001) | –0.111 (p < 0.001) | –0.035 (p = 0.021) | –0.223 (p < 0.001) | 0.041 (p < 0.001) | 0.015 (p = 0.157) |
| 4F1M | 0.056 (p < 0.001) | 0.054 (p < 0.001) | –0.042 (p = 0.004) | –0.010 (p = 0.490) | –0.113 (p < 0.001) | 0.048 (p < 0.001) | –0.005 (p = 0.651) |
| 1F4M | 0.101 (p < 0.001) | 0.100 (p < 0.001) | –0.054 (p < 0.001) | –0.026 (p = 0.086) | –0.081 (p < 0.001) | 0.076 (p < 0.001) | 0.037 (p < 0.001) |
| 1F10M | 0.248 (p < 0.001) | 0.234 (p < 0.001) | –0.164 (p < 0.001) | –0.062 (p < 0.001) | –0.272 (p < 0.001) | 0.141 (p < 0.001) | 0.033 (p = 0.002) |
| Log($n$):10F1M | –0.056 (p < 0.001) | –0.069 (p < 0.001) | 0.029 (p = 0.048) | –0.022 (p = 0.146) | 0.095 (p < 0.001) | –0.040 (p < 0.001) | –0.067 (p < 0.001) |
| Log($n$):4F1M | –0.076 (p < 0.001) | –0.077 (p < 0.001) | 0.037 (p = 0.011) | –0.013 (p = 0.400) | 0.105 (p < 0.001) | –0.059 (p < 0.001) | –0.074 (p < 0.001) |

*Appendix 2—table 2 Continued on next page*

*Appendix 2—table 2 Continued*

| | MSLL β (p-value) | SMSE β (p-value) | EV β (p-value) | Rho β (p-value) | ICC β (p-value) | Lower tail HC % β (p-value) | Upper tail HC % β (p-value) |
|---|---|---|---|---|---|---|---|
| Log(*n*):1F4M | 0.041 (p < 0.001) | 0.026 (p = 0.005) | –0.073 (p < 0.001) | –0.011 (p = 0.461) | –0.069 (p < 0.001) | 0.040 (p < 0.001) | –0.007 (p = 0.501) |
| Log(*n*):1F10M | 0.036 (p = 0.001) | 0.060 (p < 0.001) | –0.076 (p < 0.001) | 0.002 (p = 0.914) | –0.112 (p < 0.001) | 0.080 (p < 0.001) | 0.018 (p = 0.083) |

**Appendix 2—table 3.** Linear mixed model results for MSE and total outlier count (tOC) under age-skewed sampling for models trained in AIBL.

Models evaluate the influence of diagnosis (HC, AD), log-transformed and standardized sample size (*n*), and age sampling strategy (Representative, Left-skewed, Right-skewed) on standardized deviation score outcomes. Continuous variables were standardized to allow comparison of effect sizes. Representative sampling and HC are used as reference levels. Reported β coefficients and corresponding p-values indicate the direction and significance of the effects.

| | MSE β (p-value) | tOC β (p-value) |
|---|---|---|
| Intercept (HC, Representative) | –0.313 (p < 0.001) | –0.462 (p < 0.001) |
| Log(*n*) | –0.451 (p < 0.001) | 0.026 (p = 0.015) |
| AD | 0.070 (p = 0.243) | 0.909 (p < 0.001) |
| Left-skewed | 0.930 (p < 0.001) | 0.281 (p < 0.001) |
| Right-skewed | 0.560 (p < 0.001) | –0.068 (p < 0.001) |
| Log(*n*):AD | –0.073 (p = 0.039) | 0.144 (p < 0.001) |
| Log(*n*):Left-skewed | –0.359 (p < 0.001) | –0.142 (p < 0.001) |
| Log(*n*):Right-skewed | –0.521 (p < 0.001) | 0.012 (p = 0.427) |
| Left-skewed:AD | 0.211 (p < 0.001) | 0.670 (p < 0.001) |
| Right-skewed:AD | –0.061 (p = 0.224) | –0.192 (p < 0.001) |
| Log(*n*):Left-skewed:AD | –0.030 (p = 0.555) | –0.187 (p < 0.001) |
| Log(*n*):Right-skewed:AD | 0.056 (p = 0.264) | 0.005 (p = 0.819) |

**Appendix 2—table 4.** Linear mixed model results for MSE and total outlier count (tOC) under sex-imbalanced sampling for model trained in AIBL.

Models evaluate the influence of diagnosis (HC, AD), log-transformed and standardized sample size (*n*), and sex ratio in the training set (1:1, 1F:4M, 1F:10M, 4F:1M, 10F:1M) on standardized deviation score outcomes. Continuous variables were standardized to allow comparison of effect sizes. The 1:1 ratio and HC are used as reference levels. Reported β coefficients and corresponding p-values indicate the direction and significance of the effects.

| | MSE β (p-value) | tOC β (p-value) |
|---|---|---|
| Intercept (HC, 1 F 1 M) | 0.052 (p < 0.001) | –0.462 (p < 0.001) |

*Appendix 2—table 4 Continued on next page*

*Appendix 2—table 4 Continued*

| | MSE<br>β (p-value) | tOC<br>β (p-value) |
|---|---|---|
| Log(n) | 0.093 (p < 0.001) | 0.011 (p = 0.207) |
| AD | 0.183 (p < 0.001) | 0.012 (p = 0.141) |
| 10F1M | 0.070 (p = 0.009) | 0.020 (p = 0.020) |
| *4F1M* | 0.011 (p = 0.404) | 0.036 (p < 0.001) |
| *1F4M* | 0.007 (p = 0.614) | 0.909 (p < 0.001) |
| *1F10M* | 0.002 (p = 0.896) | 0.019 (p = 0.148) |
| Log(n):AD | 0.003 (p = 0.823) | 0.031 (p = 0.017) |
| *Log(n):10F1M* | −0.451 (p < 0.001) | 0.012 (p = 0.346) |
| *Log(n):4F1M* | −0.055 (p < 0.001) | 0.006 (p = 0.667) |
| *Log(n):1F4M* | −0.070 (p < 0.001) | 0.026 (p < 0.001) |
| *Log(n):1F10M* | −0.010 (p = 0.264) | −0.010 (p = 0.221) |
| 10F1M:AD | 0.042 (p < 0.001) | −0.015 (p = 0.073) |
| 4F1M:AD | −0.073 (p < 0.001) | 0.010 (p = 0.225) |
| 1F4M:AD | −0.000 (p = 0.980) | 0.021 (p = 0.014) |
| 1F10M:AD | −0.004 (p = 0.771) | 0.144 (p < 0.001) |
| Log(n):10F1M:AD | 0.001 (p = 0.949) | −0.005 (p = 0.719) |
| Log(n):4F1M:AD | 0.004 (p = 0.765) | −0.017 (p = 0.188) |
| Log(n):1F4M:AD | 0.052 (p < 0.001) | −0.006 (p = 0.657) |
| Log(n):1F10M:AD | 0.093 (p < 0.001) | 0.021 (p = 0.101) |

# Appendix 3

## Tables for replication with adaptive transfer learning in independent dataset (AIBL)

**Appendix 3—table 1.** Linear mixed model results for evaluation metrics under age-skewed sampling conditions using normative models pre-trained on the UKB and adapted to the AIBL dataset.

Models assess the influence of standardized and log-transformed sample size (*n*) and age sampling strategy (Representative, Left-skewed, Right-skewed) on model performance metrics (MSLL, SMSE, EV, Rho, ICC, the lower and upper tail HC%). All variables were standardized to allow comparison of effect sizes. Representative sampling serves as the reference level. Reported *β* coefficients and corresponding p-values indicate the direction and significance of each effect.

| | MSLL β (p-value) | SMSE β (p-value) | EV β (p-value) | Rho β (p-value) | ICC β (p-value) | Lower tail HC % β (p-value) | Upper tail HC % β (p-value) |
|---|---|---|---|---|---|---|---|
| Intercept (Representative) | 0.792 (p < 0.001) | 1.040 (p < 0.001) | –2.562 (p < 0.001) | –0.076 (p < 0.001) | 0.502 (p < 0.001) | –0.010 (p = 0.869) | –0.168 (p = 0.009) |
| Log(n) | –0.436 (p < 0.001) | –0.211 (p < 0.001) | –0.000 (p = 1.000) | –0.000 (p = 1.000) | 0.515 (p < 0.001) | –0.364 (p < 0.001) | –0.295 (p < 0.001) |
| Left-skewed | 0.862 (p < 0.001) | 0.550 (p < 0.001) | 0.000 (p = 1.000) | 0.000 (p = 1.000) | 0.078 (p < 0.001) | –0.130 (p < 0.001) | 1.701 (p < 0.001) |
| Right-skewed | 0.505 (p < 0.001) | 0.461 (p < 0.001) | 0.000 (p = 1.000) | 0.000 (p = 1.000) | 0.093 (p < 0.001) | 0.946 (p < 0.001) | –0.244 (p < 0.001) |
| Log(n):Left-skewed | –0.445 (p < 0.001) | –0.012 (p = 0.261) | 0.000 (p = 1.000) | 0.000 (p = 1.000) | 0.062 (p < 0.001) | 0.087 (p < 0.001) | –0.303 (p < 0.001) |
| Log(n):Right-skewed | –0.214 (p < 0.001) | –0.009 (p = 0.374) | 0.000 (p = 1.000) | 0.000 (p = 1.000) | –0.009 (p = 0.267) | –0.180 (p < 0.001) | 0.013 (p = 0.537) |

**Appendix 3—table 2.** Linear mixed model results for evaluation metrics under sex-imbalanced sampling conditions using normative models pre-trained on the UKB and adapted to the AIBL dataset.

Models assess the influence of standardized and log-transformed sample size (*n*) and sex ratio in the training set (1F:1M, 1F:4M, 1F:10M, 4F:1M, 10F:1M; F = female, M = male) on model performance metrics (MSLL, SMSE, EV, Rho, ICC, the lower and upper tail HC%). All variables were standardized to allow comparison of effect sizes. Representative sampling (1F:1M) serves as the reference level. Reported *β* coefficients and corresponding p-values indicate the direction and significance of each effect.

| | MSLL β (p-value) | EV β (p-value) | SMSE β (p-value) | Rho β (p-value) | ICC β (p-value) | Lower tail HC % β (p-value) | Upper tail HC % β (p-value) |
|---|---|---|---|---|---|---|---|
| Intercept (1F1M) | 0.792 (p < 0.001) | 1.040 (p < 0.001) | –2.561 (p < 0.001) | –0.075 (p < 0.001) | 0.502 (p < 0.001) | –0.010 (p = 0.864) | –0.168 (p = 0.002) |
| Log(n) | –0.436 (p < 0.001) | –0.211 (p < 0.001) | –0.000 (p = 1.000) | –0.000 (p = 1.000) | 0.515 (p < 0.001) | –0.364 (p < 0.001) | –0.295 (p < 0.001) |
| *10F1M* | 0.056 (p < 0.001) | 0.060 (p < 0.001) | –0.000 (p = 1.000) | –0.000 (p = 1.000) | 0.002 (p = 0.781) | 0.097 (p < 0.001) | 0.021 (p = 0.018) |
| *4F1M* | 0.034 (p < 0.001) | 0.032 (p < 0.001) | 0.000 (p = 1.000) | –0.000 (p = 1.000) | 0.012 (p = 0.105) | 0.056 (p < 0.001) | 0.009 (p = 0.319) |

*Appendix 3—table 2 Continued on next page*

*Appendix 3—table 2 Continued*

| | MSLL β (p-value) | EV β (p-value) | SMSE β (p-value) | Rho β (p-value) | ICC β (p-value) | Lower tail HC % β (p-value) | Upper tail HC % β (p-value) |
|---|---|---|---|---|---|---|---|
| *1F4M* | 0.025 (p = 0.011) | 0.017 (p < 0.001) | –0.000 (p = 1.000) | 0.000 (p = 1.000) | 0.031 (p < 0.001) | 0.056 (p < 0.001) | 0.033 (p < 0.001) |
| *1F10M* | 0.030 (p = 0.002) | 0.034 (p < 0.001) | –0.000 (p = 1.000) | –0.000 (p = 1.000) | 0.036 (p < 0.001) | 0.063 (p < 0.001) | 0.085 (p < 0.001) |
| *Log(n):10F1M* | –0.032 (p = 0.001) | 0.003 (p = 0.462) | 0.000 (p = 1.000) | –0.000 (p = 1.000) | –0.025 (p < 0.001) | –0.014 (p = 0.189) | –0.027 (p = 0.002) |
| *Log(n):4F1M* | –0.037 (p < 0.001) | 0.007 (p = 0.128) | –0.000 (p = 1.000) | –0.000 (p = 1.000) | –0.039 (p < 0.001) | –0.023 (p = 0.029) | –0.046 (p < 0.001) |
| *Log(n):1F4M* | –0.021 (p = 0.030) | 0.017 (p < 0.001) | 0.000 (p = 1.000) | 0.000 (p = 1.000) | –0.049 (p < 0.001) | –0.016 (p = 0.130) | –0.002 (p = 0.836) |
| *Log(n):1F10M* | 0.008 (p = 0.410) | 0.016 (p < 0.001) | 0.000 (p = 1.000) | 0.000 (p = 1.000) | –0.037 (p < 0.001) | 0.007 (p = 0.538) | –0.014 (p = 0.113) |

**Appendix 3—table 3.** Linear mixed model results for MSE and total outlier count (tOC) under age-skewed sampling for models adapted from the UK Biobank to AIBL.

Models evaluate the influence of diagnosis (HC, AD), log-transformed and standardized sample size (*n*), and age sampling strategy (Representative, Left-skewed, Right-skewed) on standardized deviation score outcomes. Continuous variables were standardized to allow comparison of effect sizes. Representative sampling and HC are used as reference levels. Reported $\beta$ coefficients and corresponding p-values indicate the direction and significance of the effects.

| | MSE β (p-value) | tOC β (p-value) |
|---|---|---|
| Intercept (HC, Representative) | –0.572 (p < 0.001) | –0.415 (p < 0.001) |
| Log(n) | –0.277 (p < 0.001) | –0.096 (p < 0.001) |
| AD | 0.009 (p = 0.613) | 0.792 (p < 0.001) |
| Left-skewed | 0.712 (p < 0.001) | –0.039 (p < 0.001) |
| Right-skewed | 0.356 (p < 0.001) | 0.252 (p < 0.001) |
| Log(n):AD | –0.011 (p = 0.305) | –0.036 (p < 0.001) |
| Log(n):Left-skewed | –0.193 (p < 0.001) | 0.026 (p < 0.001) |
| Log(n):Right-skewed | –0.070 (p < 0.001) | –0.048 (p < 0.001) |
| Left-skewed:AD | –0.111 (p < 0.001) | –0.019 (p = 0.113) |
| Right-skewed:AD | 0.036 (p = 0.020) | 0.213 (p < 0.001) |
| Log(n):Left-skewed:AD | 0.002 (p = 0.878) | 0.005 (p = 0.682) |
| Log(n):Right-skewed:AD | –0.014 (p = 0.368) | 0.005 (p = 0.650) |

**Appendix 3—table 4.** Linear mixed model results for MSE and total outlier count (tOC) under sex-imbalanced sampling for models adapted from the UK Biobank to AIBL.

Models evaluate the influence of diagnosis (HC, AD), log-transformed and standardized sample size (*n*), and sex ratio in the training set (1:1, 1F:4M, 1F:10M, 4F:1M, 10F:1M) on standardized deviation score outcomes. Continuous variables were standardized to allow comparison of effect sizes. The 1:1 ratio and HC are used as reference levels. Reported $\beta$ coefficients and corresponding p-values indicate the direction and significance of the effects.

| | MSE<br>$\beta$ (p-value) | tOC<br>$\beta$ (p-value) |
|---|---|---|
| Intercept (HC, 1 F 1 M) | –0.572 (p < 0.001) | –0.415 (p < 0.001) |
| Log($n$) | 0.027 (p < 0.001) | 0.025 (p < 0.001) |
| AD | 0.010 (p = 0.132) | 0.014 (p = 0.002) |
| 10F1M | 0.025 (p < 0.001) | 0.014 (p = 0.002) |
| 4F1M | 0.046 (p < 0.001) | 0.016 (p < 0.001) |
| 1F4M | 0.009 (p = 0.201) | 0.792 (p < 0.001) |
| 1F10M | 0.003 (p = 0.746) | 0.049 (p < 0.001) |
| Log($n$):AD | 0.002 (p = 0.838) | 0.031 (p < 0.001) |
| Log($n$):10F1M | 0.002 (p = 0.844) | 0.009 (p = 0.188) |
| Log($n$):4F1M | 0.002 (p = 0.817) | 0.015 (p = 0.031) |
| Log($n$):1F4M | –0.277 (p < 0.001) | –0.096 (p < 0.001) |
| Log($n$):1F10M | –0.001 (p = 0.923) | –0.003 (p = 0.554) |
| 10F1M:AD | 0.002 (p = 0.750) | –0.005 (p = 0.281) |
| 4F1M:AD | 0.006 (p = 0.334) | –0.003 (p = 0.560) |
| 1F4M:AD | 0.002 (p = 0.699) | 0.003 (p = 0.477) |
| 1F10M:AD | –0.011 (p = 0.111) | –0.036 (p < 0.001) |
| Log($n$):10F1M:AD | –0.002 (p = 0.816) | –0.001 (p = 0.911) |
| Log($n$):4F1M:AD | –0.003 (p = 0.778) | –0.008 (p = 0.278) |
| Log($n$):1F4M:AD | –0.002 (p = 0.872) | –0.007 (p = 0.293) |
| Log($n$):1F10M:AD | 0.000 (p = 0.996) | –0.006 (p = 0.370) |

# Appendix 4

## Control analyses

### 1. Age and sex distributions and expected sampling effects

Age and sex distributions were first summarized for each dataset and split (train HC, test HC, test AD) separately for males and females and are reported as mean ± SD and interquartile range (P25–P75).

To characterize the expected age distributions induced by the sampling procedures, we simulated the sampling used in the main analyses by repeatedly drawing samples from the training set. Because the sampling strategy enforces the same quantile-based age stratification independently of sample size, the expected age distribution is theoretically invariant with respect to $n$, with smaller $n$ only introducing additional sampling noise. We therefore performed the simulation at a single representative sample size ($n = 200$). For each ratio, 1000 independent samples were drawn. Ages were pooled across repetitions within each ratio × sex combination and summarized as mean ± SD and P25–P75.

**Appendix 4—table 1.** Empirical age distributions by sex for the training and test splits of OASIS-3, AIBL, and UK Biobank.

Age is reported as mean ± SD and interquartile range (P25–P75) separately for males and females in each split (Train HC, Test HC, Test AD where available).

|  | Split | Sex | Mean ± sd | P25–P75 |
|---|---|---|---|---|
| OASIS-3 | Train HC | Male | 68.99 ± 7.55 | 65.6–74.1 |
|  |  | Female | 67.66 ± 7.97 | 63.2–73.7 |
|  | Test HC | Male | 69.05 ± 7.67 | 66.1–73.6 |
|  |  | Female | 67.69 ± 7.97 | 63.9–73.3 |
|  | Test AD | Male | 73.69 ± 6.06 | 69.6–78.6 |
|  |  | Female | 73.91 ± 5.93 | 70.4–77.9 |
| AIBL | Train HC | Male | 72.43 ± 5.19 | 67.9–76.0 |
|  |  | Female | 72.02 ± 5.56 | 67.7–76.2 |
|  | Test HC | Male | 72.38 ± 5.46 | 69.0–77.0 |
|  |  | Female | 72.47 ± 5.42 | 68.8–77.3 |
|  | Test AD | Male | 73.51 ± 6.59 | 69.3–79.0 |
|  |  | Female | 73.65 ± 6.75 | 71.5–79.8 |
| UKB | Train HC | Male | 65.22 ± 7.80 | 59.3–71.3 |
|  |  | Female | 63.87 ± 7.52 | 58.0–69.7 |
|  | Test HC | Male | 65.20 ± 7.72 | 59.3–71.2 |
|  |  | Female | 63.85 ± 7.51 | 58.0–69.6 |

**Appendix 4—table 2.** Expected age distributions under representative, left-skewed, and right-skewed age sampling for OASIS-3 and AIBL.

Expected distributions were estimated by repeated simulation of the age-skewed sampling procedure at $n = 200$ across 1000 independent samples and are reported as mean ± SD and P25–P75 separately for males and females.

|  | Sampling | Sex | Mean ± sd | P25–P75 |
|---|---|---|---|---|
| OASIS-3 | Representative | Male | 68.24 ± 7.75 | 64.4–73.8 |
|  |  | Female | 68.20 ± 7.82 | 64.1–73.8 |

*Appendix 4—table 2 Continued on next page*

*Appendix 4—table 2 Continued*

| | Sampling | Sex | Mean ± sd | P25–P75 |
|---|---|---|---|---|
| | Left | Male | 57.58 ± 5.23 | 54.0–60.0 |
| | | Female | 57.05 ± 5.40 | 53.3–60.1 |
| | Right | Male | 72.10 ± 6.08 | 67.9–76.8 |
| | | Female | 71.94 ± 5.89 | 68.2–75.8 |
| AIBL | Representative | Male | 72.19 ± 5.34 | 67.7–76.0 |
| | | Female | 72.18 ± 5.39 | 68.0–76.2 |
| | Left | Male | 67.54 ± 3.16 | 65.5–69.1 |
| | | Female | 67.29 ± 3.19 | 65.4–69.0 |
| | Right | Male | 76.04 ± 3.49 | 74.0–78.4 |
| | | Female | 75.66 ± 3.21 | 73.9–77.4 |

**Appendix 4—table 3.** Expected age distributions under sex-imbalance sampling for OASIS-3 and AIBL.

Sex ratios include 1F:1M (representative), 1F:4M, 1F:10M, 4F:1M, and 10F:1M. Expected distributions were estimated by repeated simulation of the sampling procedure at $n$ = 200 across 1000 independent samples and are reported as mean ± SD and P25–P75 separately for males and females.

| | Sampling | Sex | Mean ± sd | P25–P75 |
|---|---|---|---|---|
| OASIS-3 | 1F1M | Male | 68.24 ± 7.74 | 64.4–73.8 |
| | | Female | 68.19 ± 7.83 | 64.1–73.8 |
| | 1F4M | Male | 68.24 ± 7.74 | 64.4–73.8 |
| | | Female | 68.19 ± 7.83 | 64.1–73.8 |
| | 1F10M | Male | 68.23 ± 7.74 | 64.4–73.8 |
| | | Female | 68.2 ± 7.85 | 64.1–73.8 |
| | 4F1M | Male | 68.24 ± 7.73 | 64.4–73.8 |
| | | Female | 68.19 ± 7.83 | 64.1–73.8 |
| | 10F1M | Male | 68.22 ± 7.78 | 64.2–73.9 |
| | | Female | 68.19 ± 7.84 | 64.1–73.8 |
| AIBL | 1F1M | Male | 72.19 ± 5.34 | 67.6–76.0 |
| | | Female | 72.19 ± 5.39 | 68.1–76.2 |
| | 1F4M | Male | 72.19 ± 5.34 | 67.6–76.0 |
| | | Female | 72.18 ± 5.39 | 68.0–76.2 |
| | 1F10M | Male | 72.19 ± 5.34 | 67.6–76.0 |
| | | Female | 72.19 ± 5.39 | 68.0–76.2 |
| | 4F1M | Male | 72.2 ± 5.33 | 67.5–76.0 |
| | | Female | 72.18 ± 5.39 | 68.1–76.2 |
| | 10F1M | Male | 72.18 ± 5.33 | 67.6–76.0 |
| | | Female | 72.18 ± 5.39 | 68.0–76.2 |

## 2. Statistical models including age and sex

**Appendix 4—table 4.** Linear mixed model results for MSE and total outlier count (tOC) under age-skewed sampling for models trained with OASIS-3 or AIBL.

Models evaluate the influence of diagnosis (HC, AD), log-transformed and standardized sample size (*n*), age sampling strategy (Representative, Left-skewed, Right-skewed), age, sex, and their interactions on standardized deviation score outcomes. Continuous variables were standardized to allow comparison of effect sizes. Representative sampling, HC, and female sex are used as reference levels. Reported $\beta$ coefficients and corresponding p-values indicate the direction and significance of the effects.

| | OASIS-3 | | AIBL | |
| --- | --- | --- | --- | --- |
| | MSE $\beta$ (p-value) | tOC $\beta$ (p-value) | MSE $\beta$ (p-value) | tOC $\beta$ (p-value) |
| Intercept | 0.113 (p = 0.824) | 0.202 (p = 0.916) | 0.245 (p = 0.876) | 0.614 (p = 0.874) |
| n | −1.237 (p < 0.001) | 0.539 (p < 0.001) | −0.972 (p = 0.096) | 1.112 (p = 0.003) |
| AD | −0.964 (p = 0.242) | −0.795 (p = 0.798) | −0.132 (p = 0.952) | −7.493 (p = 0.165) |
| left | −6.233 (p < 0.001) | −3.919 (p < 0.001) | −12.217 (p < 0.001) | −5.242 (p < 0.001) |
| right | 1.270 (p < 0.001) | −0.604 (p < 0.001) | 8.295 (p < 0.001) | −1.852 (p < 0.001) |
| n:AD | 1.197 (p < 0.001) | 0.480 (p = 0.017) | −0.285 (p = 0.726) | −0.914 (p = 0.079) |
| n:left | 4.473 (p < 0.001) | 2.006 (p < 0.001) | 6.366 (p < 0.001) | 1.041 (p = 0.049) |
| n:right | −1.482 (p < 0.001) | 0.077 (p = 0.662) | −6.562 (p < 0.001) | −0.793 (p = 0.134) |
| left:AD | −2.744 (p < 0.001) | −4.194 (p < 0.001) | 0.434 (p = 0.705) | −4.832 (p < 0.001) |
| right:AD | −0.142 (p = 0.730) | 0.550 (p = 0.054) | −0.003 (p = 0.998) | 1.009 (p = 0.171) |
| n:left:AD | 0.558 (p = 0.175) | 0.389 (p = 0.172) | −1.107 (p = 0.335) | 0.962 (p = 0.192) |
| n:right:AD | −0.415 (p = 0.313) | 0.253 (p = 0.375) | −1.304 (p = 0.256) | 0.368 (p = 0.618) |
| age | −0.005 (p = 0.477) | −0.008 (p = 0.786) | −0.008 (p = 0.706) | −0.014 (p = 0.795) |
| left:age | 0.104 (p < 0.001) | 0.065 (p < 0.001) | 0.183 (p < 0.001) | 0.076 (p < 0.001) |
| right:age | −0.017 (p < 0.001) | 0.009 (p < 0.001) | −0.108 (p < 0.001) | 0.024 (p < 0.001) |
| AD:age | 0.014 (p = 0.228) | 0.017 (p = 0.685) | 0.003 (p = 0.922) | 0.108 (p = 0.141) |
| left:AD:age | 0.037 (p < 0.001) | 0.057 (p < 0.001) | −0.007 (p = 0.650) | 0.072 (p < 0.001) |
| right:AD:age | 0.002 (p = 0.686) | −0.007 (p = 0.064) | 0.002 (p = 0.884) | −0.014 (p = 0.167) |
| n:age | 0.012 (p < 0.001) | −0.007 (p < 0.001) | 0.008 (p = 0.330) | −0.014 (p = 0.006) |
| n:left:age | −0.075 (p < 0.001) | −0.034 (p < 0.001) | −0.094 (p < 0.001) | −0.016 (p = 0.030) |
| n:right:age | 0.019 (p < 0.001) | −0.001 (p = 0.690) | 0.084 (p < 0.001) | 0.010 (p = 0.152) |
| n:AD:age | −0.017 (p < 0.001) | −0.005 (p = 0.073) | 0.003 (p = 0.820) | 0.013 (p = 0.068) |
| n:left:AD:age | −0.006 (p = 0.299) | −0.005 (p = 0.166) | 0.016 (p = 0.294) | −0.015 (p = 0.132) |
| n:right:AD:age | 0.005 (p = 0.362) | −0.003 (p = 0.383) | 0.016 (p = 0.318) | −0.005 (p = 0.612) |

*Appendix 4—table 4 Continued on next page*

*Appendix 4—table 4 Continued*

|  | OASIS-3 |  | AIBL |  |
| --- | --- | --- | --- | --- |
| sex | 0.073 (p = 0.809) | –0.521 (p = 0.647) | –0.290 (p = 0.759) | –0.660 (p = 0.778) |
| left:sex | 0.002 (p = 0.990) | 0.722 (p < 0.001) | 0.057 (p = 0.909) | 1.052 (p = 0.001) |
| right:sex | 0.018 (p = 0.907) | 0.228 (p = 0.029) | 0.288 (p = 0.563) | 0.854 (p = 0.008) |
| AD:sex | 0.009 (p = 0.987) | 1.404 (p = 0.475) | 0.323 (p = 0.806) | 5.076 (p = 0.120) |
| left:AD:sex | 0.086 (p = 0.741) | 0.347 (p = 0.055) | 1.359 (p = 0.050) | 0.541 (p = 0.224) |
| right:AD:sex | –0.106 (p = 0.685) | –0.438 (p = 0.015) | –0.694 (p = 0.317) | –1.900 (p < 0.001) |
| n:sex | –0.034 (p = 0.751) | –0.200 (p = 0.007) | 0.187 (p = 0.595) | –0.302 (p = 0.182) |
| n:left:sex | –0.222 (p = 0.142) | –0.285 (p = 0.006) | –0.394 (p = 0.430) | 0.061 (p = 0.850) |
| n:right:sex | –0.089 (p = 0.554) | –0.024 (p = 0.817) | –0.555 (p = 0.266) | 0.240 (p = 0.453) |
| n:AD:sex | –0.067 (p = 0.716) | 0.259 (p = 0.042) | –0.178 (p = 0.717) | 1.452 (p < 0.001) |
| n:left:AD:sex | –0.176 (p = 0.499) | –0.067 (p = 0.712) | –0.170 (p = 0.806) | –0.605 (p = 0.174) |
| n:right:AD:sex | 0.433 (p = 0.096) | 0.070 (p = 0.699) | 1.254 (p = 0.071) | –0.708 (p = 0.112) |
| age:sex | –0.001 (p = 0.803) | 0.006 (p = 0.696) | 0.004 (p = 0.743) | 0.008 (p = 0.793) |
| left:age:sex | 0.002 (p = 0.466) | –0.011 (p < 0.001) | –0.002 (p = 0.767) | –0.015 (p < 0.001) |
| right:age:sex | –0.000 (p = 0.884) | –0.003 (p = 0.028) | –0.003 (p = 0.635) | –0.012 (p = 0.009) |
| AD:age:sex | –0.000 (p = 0.993) | –0.018 (p = 0.517) | –0.004 (p = 0.803) | –0.065 (p = 0.143) |
| left:AD:age:sex | –0.001 (p = 0.826) | –0.001 (p = 0.757) | –0.018 (p = 0.062) | –0.006 (p = 0.310) |
| right:AD:age:sex | 0.001 (p = 0.709) | 0.006 (p = 0.020) | 0.009 (p = 0.364) | 0.024 (p < 0.001) |
| n:age:sex | 0.000 (p = 0.787) | 0.002 (p = 0.026) | –0.003 (p = 0.538) | 0.004 (p = 0.245) |
| n:left:age:sex | 0.002 (p = 0.330) | 0.004 (p = 0.004) | 0.006 (p = 0.373) | –0.001 (p = 0.789) |
| n:right:age:sex | 0.001 (p = 0.540) | 0.000 (p = 0.764) | 0.007 (p = 0.300) | –0.003 (p = 0.510) |
| n:AD:age:sex | 0.001 (p = 0.717) | –0.004 (p = 0.047) | 0.003 (p = 0.697) | –0.019 (p < 0.001) |
| n:left:AD:age:sex | 0.003 (p = 0.460) | –0.000 (p = 0.931) | 0.002 (p = 0.834) | 0.008 (p = 0.183) |
| n:right:AD:age:sex | –0.006 (p = 0.111) | –0.001 (p = 0.707) | –0.016 (p = 0.088) | 0.010 (p = 0.111) |

**Appendix 4—table 5.** Linear mixed model results for MSE and total outlier count (tOC) under sex-imbalanced sampling for models trained with OASIS-3 or AIBL.
Models evaluate the influence of diagnosis (HC, AD), log-transformed and standardized sample size ($n$), sex ratio in the training set (1:1, 1F:4M, 1F:10M, 4F:1M, 10F:1M), age, sex, and their interactions on standardized deviation score outcomes. Continuous variables were standardized to allow comparison of effect sizes. The 1:1 ratio, HC, and female sex are used as reference levels. Reported $\beta$ coefficients and corresponding p-values indicate the direction and significance of the effects.

*Appendix 4—table 5 Continued on next page*

| | OASIS-3 | | AIBL | |
|---|---|---|---|---|
| | MSE $\beta$ (p-value) | tOC $\beta$ (p-value) | MSE $\beta$ (p-value) | tOC $\beta$ (p-value) |
| Intercept | 0.113 (p = 0.674) | 0.202 (p = 0.905) | 0.245 (p = 0.754) | 0.614 (p = 0.868) |
| n | −1.237 (p < 0.001) | 0.539 (p < 0.001) | −0.972 (p < 0.001) | 1.112 (p < 0.001) |
| AD | −0.964 (p = 0.027) | −0.795 (p = 0.771) | −0.132 (p = 0.903) | −7.493 (p = 0.144) |
| 10F1M | 0.461 (p = 0.002) | 0.052 (p = 0.618) | 0.649 (p = 0.086) | −0.027 (p = 0.934) |
| 4F1M | 0.129 (p = 0.383) | 0.055 (p = 0.595) | 0.381 (p = 0.313) | −0.040 (p = 0.901) |
| 1F4M | 0.080 (p = 0.587) | −0.155 (p = 0.135) | −0.033 (p = 0.931) | 0.295 (p = 0.361) |
| 1F10M | −0.061 (p = 0.677) | −0.101 (p = 0.331) | −0.097 (p = 0.797) | 0.474 (p = 0.142) |
| n:AD | 1.197 (p < 0.001) | 0.480 (p < 0.001) | −0.285 (p = 0.443) | −0.914 (p = 0.004) |
| n:10F1M | −0.150 (p = 0.308) | 0.035 (p = 0.732) | −0.587 (p = 0.120) | 0.067 (p = 0.836) |
| n:4F1M | 0.032 (p = 0.826) | 0.066 (p = 0.523) | −0.452 (p = 0.231) | 0.183 (p = 0.571) |
| n:1F4M | −0.178 (p = 0.228) | 0.020 (p = 0.844) | 0.285 (p = 0.450) | 0.062 (p = 0.849) |
| n:1F10M | 0.338 (p = 0.022) | −0.157 (p = 0.129) | 0.469 (p = 0.214) | −0.165 (p = 0.610) |
| 10F1M:AD | 0.729 (p = 0.002) | −0.558 (p < 0.001) | −0.102 (p = 0.845) | −0.071 (p = 0.874) |
| 4F1M:AD | 0.351 (p = 0.142) | −0.404 (p = 0.016) | 0.050 (p = 0.925) | −0.140 (p = 0.755) |
| 1F4M:AD | −0.535 (p = 0.025) | 0.190 (p = 0.257) | 0.102 (p = 0.846) | −0.201 (p = 0.654) |
| 1F10M:AD | −0.890 (p < 0.001) | 0.407 (p = 0.015) | −0.028 (p = 0.958) | −0.067 (p = 0.881) |
| n:10F1M:AD | −0.462 (p = 0.053) | −0.078 (p = 0.641) | −0.020 (p = 0.969) | 0.000 (p = 0.999) |
| n:4F1M:AD | −0.239 (p = 0.318) | −0.022 (p = 0.897) | −0.053 (p = 0.919) | 0.105 (p = 0.814) |
| n:1F4M:AD | 0.291 (p = 0.223) | −0.056 (p = 0.739) | −0.548 (p = 0.297) | 0.156 (p = 0.728) |
| n:1F10M:AD | 0.302 (p = 0.206) | −0.198 (p = 0.239) | −0.492 (p = 0.349) | −0.264 (p = 0.556) |
| age | −0.005 (p = 0.180) | −0.008 (p = 0.757) | −0.008 (p = 0.450) | −0.014 (p = 0.785) |
| 10F1M:age | −0.000 (p = 0.819) | −0.001 (p = 0.667) | −0.004 (p = 0.463) | 0.000 (p = 0.931) |
| 4F1M:age | 0.001 (p = 0.692) | −0.000 (p = 0.789) | −0.002 (p = 0.659) | 0.001 (p = 0.855) |
| 1F4M:age | −0.002 (p = 0.245) | 0.002 (p = 0.158) | −0.001 (p = 0.889) | −0.004 (p = 0.336) |
| 1F10M:age | −0.002 (p = 0.426) | 0.001 (p = 0.441) | −0.001 (p = 0.802) | −0.007 (p = 0.121) |
| AD:age | 0.014 (p = 0.023) | 0.017 (p = 0.645) | 0.003 (p = 0.844) | 0.108 (p = 0.122) |
| 10F1M:AD:age | −0.010 (p = 0.002) | 0.006 (p = 0.011) | 0.001 (p = 0.850) | −0.001 (p = 0.826) |
| 4F1M:AD:age | −0.005 (p = 0.137) | 0.005 (p = 0.035) | −0.001 (p = 0.928) | 0.001 (p = 0.906) |
| 1F4M:AD:age | 0.008 (p = 0.023) | −0.003 (p = 0.283) | −0.001 (p = 0.906) | 0.003 (p = 0.616) |
| 1F10M:AD:age | 0.013 (p < 0.001) | −0.005 (p = 0.032) | 0.001 (p = 0.879) | 0.002 (p = 0.802) |

| | OASIS-3 | | AIBL | |
|---|---|---|---|---|
| n:age | 0.012 (p < 0.001) | –0.007 (p < 0.001) | 0.008 (p = 0.033) | –0.014 (p < 0.001) |
| n:10F1M:age | –0.000 (p = 0.817) | –0.001 (p = 0.375) | 0.006 (p = 0.230) | –0.002 (p = 0.718) |
| n:4F1M:age | –0.002 (p = 0.328) | –0.002 (p = 0.250) | 0.004 (p = 0.432) | –0.003 (p = 0.469) |
| n:1F4M:age | 0.003 (p = 0.121) | –0.001 (p = 0.644) | –0.004 (p = 0.481) | –0.001 (p = 0.837) |
| n:1F10M:age | –0.003 (p = 0.130) | 0.002 (p = 0.162) | –0.006 (p = 0.259) | 0.002 (p = 0.679) |
| n:AD:age | –0.017 (p < 0.001) | –0.005 (p = 0.002) | 0.003 (p = 0.619) | 0.013 (p = 0.003) |
| n:10F1M:AD:age | 0.006 (p = 0.052) | 0.002 (p = 0.503) | 0.000 (p = 0.985) | –0.000 (p = 0.943) |
| n:4F1M:AD:age | 0.003 (p = 0.314) | 0.001 (p = 0.795) | 0.001 (p = 0.935) | –0.002 (p = 0.748) |
| n:1F4M:AD:age | –0.004 (p = 0.226) | 0.001 (p = 0.772) | 0.007 (p = 0.340) | –0.002 (p = 0.705) |
| n:1F10M:AD:age | –0.004 (p = 0.222) | 0.002 (p = 0.350) | 0.006 (p = 0.371) | 0.003 (p = 0.609) |
| sex | 0.073 (p = 0.648) | –0.521 (p = 0.603) | –0.290 (p = 0.539) | –0.660 (p = 0.767) |
| 10F1M:sex | –0.134 (p = 0.125) | –0.003 (p = 0.960) | –0.308 (p = 0.177) | 0.083 (p = 0.672) |
| 4F1M:sex | –0.019 (p = 0.826) | –0.012 (p = 0.849) | –0.135 (p = 0.554) | 0.071 (p = 0.716) |
| 1F4M:sex | 0.007 (p = 0.939) | 0.072 (p = 0.244) | 0.071 (p = 0.755) | –0.101 (p = 0.605) |
| 1F10M:sex | 0.042 (p = 0.629) | 0.046 (p = 0.455) | 0.219 (p = 0.337) | –0.158 (p = 0.419) |
| AD:sex | 0.009 (p = 0.975) | 1.404 (p = 0.417) | 0.323 (p = 0.623) | 5.076 (p = 0.102) |
| 10F1M:AD:sex | –0.519 (p < 0.001) | 0.369 (p < 0.001) | 0.092 (p = 0.773) | 0.438 (p = 0.107) |
| 4F1M:AD:sex | –0.267 (p = 0.078) | 0.241 (p = 0.024) | –0.026 (p = 0.935) | 0.340 (p = 0.211) |
| 1F4M:AD:sex | 0.300 (p = 0.047) | –0.194 (p = 0.068) | –0.036 (p = 0.909) | –0.024 (p = 0.930) |
| 1F10M:AD:sex | 0.552 (p < 0.001) | –0.333 (p = 0.002) | –0.002 (p = 0.995) | –0.136 (p = 0.616) |
| n:sex | –0.034 (p = 0.584) | –0.200 (p < 0.001) | 0.187 (p = 0.246) | –0.302 (p = 0.029) |
| n:10F1M:sex | 0.009 (p = 0.920) | –0.068 (p = 0.273) | 0.082 (p = 0.721) | –0.082 (p = 0.673) |
| n:4F1M:sex | –0.063 (p = 0.469) | –0.074 (p = 0.230) | –0.003 (p = 0.991) | –0.118 (p = 0.546) |
| n:1F4M:sex | 0.061 (p = 0.487) | –0.016 (p = 0.795) | –0.217 (p = 0.342) | –0.010 (p = 0.959) |
| n:1F10M:sex | –0.018 (p = 0.834) | 0.090 (p = 0.145) | –0.286 (p = 0.210) | 0.147 (p = 0.451) |
| n:AD:sex | –0.067 (p = 0.531) | 0.259 (p < 0.001) | –0.178 (p = 0.428) | 1.452 (p < 0.001) |
| n:10F1M:AD:sex | 0.305 (p = 0.044) | 0.044 (p = 0.680) | 0.011 (p = 0.972) | 0.162 (p = 0.551) |
| n:4F1M:AD:sex | 0.199 (p = 0.188) | 0.052 (p = 0.628) | 0.042 (p = 0.895) | 0.083 (p = 0.760) |
| n:1F4M:AD:sex | –0.188 (p = 0.214) | 0.061 (p = 0.566) | 0.305 (p = 0.337) | –0.073 (p = 0.787) |
| n:1F10M:AD:sex | –0.302 (p = 0.046) | 0.128 (p = 0.229) | 0.337 (p = 0.288) | 0.088 (p = 0.747) |
| age:sex | –0.001 (p = 0.637) | 0.006 (p = 0.658) | 0.004 (p = 0.510) | 0.008 (p = 0.782) |

| | OASIS-3 | | AIBL | |
|---|---|---|---|---|
| 10F1M:age:sex | −0.001 (p = 0.360) | 0.000 (p = 0.998) | 0.002 (p = 0.570) | −0.001 (p = 0.693) |
| 4F1M:age:sex | −0.001 (p = 0.363) | −0.000 (p = 0.964) | 0.000 (p = 0.887) | −0.001 (p = 0.700) |
| 1F4M:age:sex | 0.001 (p = 0.365) | −0.001 (p = 0.317) | 0.001 (p = 0.856) | 0.002 (p = 0.526) |
| 1F10M:age:sex | 0.002 (p = 0.159) | −0.000 (p = 0.676) | 0.000 (p = 0.939) | 0.003 (p = 0.308) |
| AD:age:sex | −0.000 (p = 0.987) | −0.018 (p = 0.461) | −0.004 (p = 0.617) | −0.065 (p = 0.124) |
| 10F1M:AD:age:sex | 0.007 (p < 0.001) | −0.004 (p = 0.008) | −0.001 (p = 0.794) | −0.004 (p = 0.246) |
| 4F1M:AD:age:sex | 0.004 (p = 0.071) | −0.003 (p = 0.059) | 0.000 (p = 0.926) | −0.004 (p = 0.333) |
| 1F4M:AD:age:sex | −0.004 (p = 0.044) | 0.003 (p = 0.088) | 0.000 (p = 0.968) | 0.000 (p = 0.952) |
| 1F10M:AD:age:sex | −0.008 (p < 0.001) | 0.004 (p = 0.007) | −0.000 (p = 0.933) | 0.002 (p = 0.684) |
| n:age:sex | 0.000 (p = 0.642) | 0.002 (p < 0.001) | −0.003 (p = 0.178) | 0.004 (p = 0.057) |
| n:10F1M:age:sex | 0.001 (p = 0.275) | 0.001 (p = 0.112) | −0.000 (p = 0.893) | 0.001 (p = 0.583) |
| n:4F1M:age:sex | 0.002 (p = 0.148) | 0.001 (p = 0.102) | 0.001 (p = 0.803) | 0.002 (p = 0.471) |
| n:1F4M:age:sex | −0.001 (p = 0.246) | 0.000 (p = 0.641) | 0.003 (p = 0.385) | 0.000 (p = 0.924) |
| n:1F10M:age:sex | −0.000 (p = 0.708) | −0.001 (p = 0.225) | 0.004 (p = 0.210) | −0.002 (p = 0.552) |
| n:AD:age:sex | 0.001 (p = 0.532) | −0.004 (p < 0.001) | 0.003 (p = 0.395) | −0.019 (p < 0.001) |
| n:10F1M:AD:age:sex | −0.004 (p = 0.040) | −0.001 (p = 0.555) | −0.000 (p = 0.979) | −0.002 (p = 0.596) |
| n:4F1M:AD:age:sex | −0.003 (p = 0.174) | −0.001 (p = 0.539) | −0.001 (p = 0.896) | −0.001 (p = 0.803) |
| n:1F4M:AD:age:sex | 0.003 (p = 0.214) | −0.001 (p = 0.589) | −0.004 (p = 0.386) | 0.001 (p = 0.769) |
| n:1F10M:AD:age:sex | 0.004 (p = 0.046) | −0.001 (p = 0.352) | −0.004 (p = 0.312) | −0.001 (p = 0.853) |

**Appendix 4—table 6.** Linear mixed model results for MSE and total outlier count (tOC) under age-skewed sampling for models trained with UKB and transferred to OASIS-3 or AIBL.
Models evaluate the influence of diagnosis (HC, AD), log-transformed and standardized sample size ($n$), age sampling strategy (Representative, Left-skewed, Right-skewed), age, sex, and their interactions on standardized deviation score outcomes. Continuous variables were standardized to allow comparison of effect sizes. Representative sampling, HC, and female sex are used as reference levels. Reported $\beta$ coefficients and corresponding p-values indicate the direction and significance of the effects.

| | OASIS-3 | | AIBL | |
|---|---|---|---|---|
| | MSE β (p-value) | tOC β (p-value) | MSE β (p-value) | tOC β (p-value) |
| Intercept | −0.475 (p < 0.001) | 1.421 (p = 0.448) | −0.475 (p = 0.345) | 7.329 (p = 0.072) |
| n | −0.172 (p = 0.004) | −0.261 (p < 0.001) | −0.439 (p = 0.154) | −1.094 (p < 0.001) |
| AD | −0.013 (p = 0.893) | 0.355 (p = 0.907) | −0.061 (p = 0.930) | −8.922 (p = 0.116) |
| left | 0.187 (p = 0.025) | 0.215 (p = 0.002) | −1.989 (p < 0.001) | −0.726 (p = 0.011) |

*Appendix 4—table 6 Continued on next page*

*Appendix 4—table 6 Continued*

| | OASIS-3 | | AIBL | |
|---|---|---|---|---|
| right | 0.009 (p = 0.918) | 0.397 (p < 0.001) | 1.258 (p = 0.004) | 5.751 (p < 0.001) |
| n:AD | 0.013 (p = 0.889) | 0.110 (p = 0.167) | 0.146 (p = 0.733) | 0.843 (p = 0.003) |
| n:left | −0.063 (p = 0.447) | 0.019 (p = 0.783) | 0.138 (p = 0.752) | 0.206 (p = 0.469) |
| n:right | 0.001 (p = 0.990) | 0.109 (p = 0.117) | −0.404 (p = 0.354) | 0.013 (p = 0.964) |
| left:AD | 0.031 (p = 0.818) | −0.213 (p = 0.058) | 4.092 (p < 0.001) | 0.250 (p = 0.528) |
| right:AD | −0.000 (p = 0.999) | 0.058 (p = 0.608) | −0.663 (p = 0.274) | −5.204 (p < 0.001) |
| n:left:AD | −0.002 (p = 0.987) | 0.041 (p = 0.712) | 0.040 (p = 0.947) | −0.094 (p = 0.812) |
| n:right:AD | 0.012 (p = 0.930) | 0.016 (p = 0.889) | 0.556 (p = 0.359) | −0.058 (p = 0.884) |
| age | −0.000 (p = 0.743) | −0.024 (p = 0.366) | −0.001 (p = 0.848) | −0.104 (p = 0.064) |
| left:age | 0.000 (p = 0.907) | −0.001 (p = 0.160) | 0.037 (p < 0.001) | 0.009 (p = 0.021) |
| right:age | 0.000 (p = 0.994) | −0.005 (p < 0.001) | −0.012 (p = 0.041) | −0.074 (p < 0.001) |
| AD:age | 0.000 (p = 0.850) | 0.002 (p = 0.955) | 0.001 (p = 0.930) | 0.122 (p = 0.117) |
| left:AD:age | −0.000 (p = 0.940) | 0.004 (p = 0.010) | −0.056 (p < 0.001) | −0.003 (p = 0.551) |
| right:AD:age | −0.000 (p = 0.993) | −0.001 (p = 0.662) | 0.009 (p = 0.264) | 0.072 (p < 0.001) |
| n:age | 0.000 (p = 0.628) | 0.002 (p = 0.003) | 0.002 (p = 0.599) | 0.014 (p < 0.001) |
| n:left:age | −0.000 (p = 0.857) | −0.000 (p = 0.778) | −0.005 (p = 0.444) | −0.002 (p = 0.527) |
| n:right:age | 0.000 (p = 0.994) | −0.001 (p = 0.161) | 0.004 (p = 0.454) | −0.001 (p = 0.887) |
| n:AD:age | −0.000 (p = 0.822) | −0.002 (p = 0.127) | −0.002 (p = 0.735) | −0.012 (p = 0.002) |
| n:left:AD:age | −0.000 (p = 0.960) | −0.001 (p = 0.665) | −0.000 (p = 0.959) | 0.001 (p = 0.810) |
| n:right:AD:age | −0.000 (p = 0.928) | −0.000 (p = 0.968) | −0.008 (p = 0.360) | 0.001 (p = 0.921) |
| sex | −0.013 (p = 0.708) | −0.983 (p = 0.377) | −0.088 (p = 0.773) | −3.181 (p = 0.197) |
| left:sex | −0.025 (p = 0.611) | −0.150 (p < 0.001) | 0.386 (p = 0.142) | 0.166 (p = 0.335) |
| right:sex | 0.001 (p = 0.988) | −0.152 (p < 0.001) | −0.102 (p = 0.697) | −1.733 (p < 0.001) |
| AD:sex | 0.017 (p = 0.775) | 1.334 (p = 0.488) | 0.125 (p = 0.768) | 10.016 (p = 0.004) |
| left:AD:sex | −0.038 (p = 0.654) | 0.320 (p < 0.001) | −1.816 (p < 0.001) | −0.124 (p = 0.605) |
| right:AD:sex | 0.002 (p = 0.982) | 0.079 (p = 0.270) | 0.351 (p = 0.338) | 3.313 (p < 0.001) |
| n:sex | 0.019 (p = 0.592) | 0.094 (p = 0.001) | 0.119 (p = 0.521) | 0.170 (p = 0.161) |
| n:left:sex | 0.015 (p = 0.759) | 0.009 (p = 0.835) | 0.170 (p = 0.520) | 0.042 (p = 0.808) |
| n:right:sex | 0.002 (p = 0.960) | −0.064 (p = 0.120) | 0.130 (p = 0.622) | −0.178 (p = 0.300) |
| n:AD:sex | −0.022 (p = 0.710) | −0.181 (p < 0.001) | −0.186 (p = 0.474) | −0.402 (p = 0.018) |

*Appendix 4—table 6 Continued on next page*

*Appendix 4—table 6 Continued*

|  | OASIS-3 |  | AIBL |  |
| --- | --- | --- | --- | --- |
| n:left:AD:sex | 0.007 (p = 0.933) | –0.005 (p = 0.938) | –0.263 (p = 0.472) | 0.020 (p = 0.934) |
| n:right:AD:sex | –0.013 (p = 0.881) | 0.032 (p = 0.654) | –0.333 (p = 0.364) | 0.257 (p = 0.284) |
| age:sex | 0.000 (p = 0.715) | 0.013 (p = 0.435) | 0.001 (p = 0.774) | 0.042 (p = 0.216) |
| left:age:sex | 0.001 (p = 0.484) | 0.002 (p = 0.005) | –0.005 (p = 0.138) | –0.002 (p = 0.403) |
| right:age:sex | –0.000 (p = 0.975) | 0.002 (p < 0.001) | 0.001 (p = 0.725) | 0.023 (p < 0.001) |
| AD:age:sex | –0.000 (p = 0.761) | –0.017 (p = 0.511) | –0.002 (p = 0.778) | –0.129 (p = 0.006) |
| left:AD:age:sex | 0.000 (p = 0.814) | –0.004 (p < 0.001) | 0.024 (p < 0.001) | 0.001 (p = 0.687) |
| right:AD:age:sex | –0.000 (p = 0.996) | –0.001 (p = 0.345) | –0.005 (p = 0.367) | –0.043 (p < 0.001) |
| n:age:sex | –0.000 (p = 0.608) | –0.001 (p = 0.020) | –0.002 (p = 0.522) | –0.002 (p = 0.174) |
| n:left:age:sex | –0.000 (p = 0.693) | –0.000 (p = 0.795) | –0.002 (p = 0.521) | –0.001 (p = 0.799) |
| n:right:age:sex | –0.000 (p = 0.971) | 0.001 (p = 0.157) | –0.002 (p = 0.637) | 0.002 (p = 0.335) |
| n:AD:age:sex | 0.000 (p = 0.688) | 0.002 (p = 0.002) | 0.002 (p = 0.493) | 0.005 (p = 0.024) |
| n:left:AD:age:sex | –0.000 (p = 0.991) | 0.000 (p = 0.891) | 0.004 (p = 0.471) | –0.000 (p = 0.956) |
| n:right:AD:age:sex | 0.000 (p = 0.897) | –0.000 (p = 0.666) | 0.004 (p = 0.379) | –0.003 (p = 0.311) |

**Appendix 4—table 7.** Linear mixed model results for MSE and total outlier count (tOC) under sex-imbalanced sampling for models trained with UKB and transferred to OASIS-3 or AIBL.

Models evaluate the influence of diagnosis (HC, AD), log-transformed and standardized sample size (*n*), sex ratio in the training set (1:1, 1F:4M, 1F:10M, 4F:1M, 10F:1M), age, sex, and their interactions on standardized deviation score outcomes. Continuous variables were standardized to allow comparison of effect sizes. The 1:1 ratio, HC, and female sex are used as reference levels. Reported $\beta$ coefficients and corresponding p-values indicate the direction and significance of the effects.

|  | OASIS-3 |  | AIBL |  |
| --- | --- | --- | --- | --- |
|  | MSE $\beta$ (p-value) | tOC $\beta$ (p-value) | MSE $\beta$ (p-value) | tOC $\beta$ (p-value) |
| Intercept | NA | 1.421 (p = 0.433) | –0.475 (p = 0.022) | 7.329 (p = 0.062) |
| n | –0.172 ($P < 0.001$) | –0.261 ($P < 0.001$) | –0.439 (p = 0.034) | –1.094 ($P < 0.001$) |
| AD | 0.000 (p = 1.000) | 0.355 (p = 0.904) | –0.061 (p = 0.831) | –8.922 (p = 0.103) |
| 10F1M | 0.014 (p = 0.822) | 0.323 ($P < 0.001$) | –0.026 (p = 0.929) | –0.038 (p = 0.844) |
| 4F1M | 0.006 (p = 0.929) | 0.216 ($P < 0.001$) | –0.027 (p = 0.925) | 0.033 (p = 0.864) |
| 1F4M | 0.015 (p = 0.821) | –0.009 (p = 0.880) | 0.088 (p = 0.763) | 0.435 (p = 0.025) |
| 1F10M | 0.016 (p = 0.809) | 0.027 (p = 0.647) | 0.072 (p = 0.805) | 0.425 (p = 0.029) |
| n:AD | 0.013 (p = 0.856) | 0.110 (p = 0.108) | 0.146 (p = 0.611) | 0.843 ($P < 0.001$) |
| n:10F1M | 0.006 (p = 0.920) | 0.013 (p = 0.821) | 0.091 (p = 0.756) | 0.130 (p = 0.504) |
| n:4F1M | 0.007 (p = 0.910) | –0.002 (p = 0.968) | 0.072 (p = 0.805) | 0.009 (p = 0.963) |

*Appendix 4—table 7 Continued on next page*

*Appendix 4—table 7 Continued*

|  | OASIS-3 |  | AIBL |  |
| --- | --- | --- | --- | --- |
| n:1F4M | –0.011 (p = 0.858) | –0.002 (p = 0.969) | –0.050 (p = 0.865) | –0.177 (p = 0.361) |
| n:1F10M | 0.003 (p = 0.964) | –0.009 (p = 0.878) | 0.130 (p = 0.656) | 0.029 (p = 0.883) |
| 10F1M:AD | 0.004 (p = 0.969) | –0.409 (*P* < 0.001) | 0.050 (p = 0.902) | 0.424 (p = 0.117) |
| 4F1M:AD | 0.000 (p = 0.999) | –0.267 (p = 0.006) | 0.021 (p = 0.959) | 0.347 (p = 0.200) |
| 1F4M:AD | –0.003 (p = 0.974) | –0.002 (p = 0.983) | –0.135 (p = 0.741) | –0.537 (p = 0.047) |
| 1F10M:AD | –0.000 (p = 0.997) | –0.015 (p = 0.873) | –0.128 (p = 0.754) | –0.756 (p = 0.005) |
| n:10F1M:AD | –0.008 (p = 0.937) | 0.015 (p = 0.878) | –0.105 (p = 0.796) | –0.513 (p = 0.058) |
| n:4F1M:AD | –0.004 (p = 0.969) | 0.049 (p = 0.611) | –0.047 (p = 0.909) | –0.301 (p = 0.265) |
| n:1F4M:AD | 0.014 (p = 0.893) | 0.019 (p = 0.847) | 0.130 (p = 0.750) | 0.198 (p = 0.465) |
| n:1F10M:AD | 0.004 (p = 0.968) | 0.037 (p = 0.698) | –0.108 (p = 0.791) | 0.127 (p = 0.640) |
| age | 0.000 (p = 1.000) | –0.024 (p = 0.350) | –0.001 (p = 0.641) | –0.104 (p = 0.055) |
| 10F1M:age | –0.000 (p = 0.943) | –0.004 (*P* < 0.001) | 0.001 (p = 0.849) | 0.001 (p = 0.677) |
| 4F1M:age | –0.000 (p = 0.963) | –0.003 (p = 0.003) | 0.001 (p = 0.895) | –0.000 (p = 0.994) |
| 1F4M:age | –0.000 (p = 0.932) | –0.000 (p = 0.968) | –0.001 (p = 0.826) | –0.006 (p = 0.027) |
| 1F10M:age | –0.000 (p = 0.977) | –0.000 (p = 0.567) | –0.000 (p = 0.921) | –0.006 (p = 0.028) |
| AD:age | –0.000 (p = 1.000) | 0.002 (p = 0.954) | 0.001 (p = 0.831) | 0.122 (p = 0.104) |
| 10F1M:AD:age | –0.000 (p = 0.988) | 0.006 (*P* < 0.001) | –0.001 (p = 0.905) | –0.005 (p = 0.156) |
| 4F1M:AD:age | 0.000 (p = 0.992) | 0.004 (p = 0.002) | –0.000 (p = 0.963) | –0.004 (p = 0.236) |
| 1F4M:AD:age | 0.000 (p = 0.967) | –0.000 (p = 0.895) | 0.002 (p = 0.738) | 0.007 (p = 0.050) |
| 1F10M:AD:age | 0.000 (p = 0.989) | 0.000 (p = 0.928) | 0.002 (p = 0.752) | 0.010 (p = 0.006) |
| n:age | 0.000 (p = 0.529) | 0.002 (*P* < 0.001) | 0.002 (p = 0.434) | 0.014 (*P* < 0.001) |
| n:10F1M:age | –0.000 (p = 0.980) | –0.000 (p = 0.831) | –0.001 (p = 0.747) | –0.002 (p = 0.475) |
| n:4F1M:age | –0.000 (p = 0.991) | 0.000 (p = 0.939) | –0.001 (p = 0.802) | –0.000 (p = 0.893) |
| n:1F4M:age | 0.000 (p = 0.862) | 0.000 (p = 0.887) | 0.001 (p = 0.852) | 0.002 (p = 0.378) |
| n:1F10M:age | –0.000 (p = 0.993) | 0.000 (p = 0.806) | –0.002 (p = 0.662) | –0.000 (p = 0.888) |
| n:AD:age | –0.000 (p = 0.770) | –0.002 (p = 0.076) | –0.002 (p = 0.614) | –0.012 (*P* < 0.001) |
| n:10F1M:AD:age | 0.000 (p = 0.949) | –0.000 (p = 0.894) | 0.001 (p = 0.800) | 0.007 (p = 0.062) |
| n:4F1M:AD:age | 0.000 (p = 0.978) | –0.001 (p = 0.619) | 0.001 (p = 0.913) | 0.004 (p = 0.271) |
| n:1F4M:AD:age | –0.000 (p = 0.886) | –0.000 (p = 0.859) | –0.002 (p = 0.746) | –0.003 (p = 0.463) |

*Appendix 4—table 7 Continued on next page*

*Appendix 4—table 7 Continued*

| | OASIS-3 | | AIBL | |
|---|---|---|---|---|
| n:1F10M:AD:age | –0.000 (p = 0.969) | –0.001 (p = 0.648) | 0.001 (p = 0.798) | –0.002 (p = 0.660) |
| sex | NA | –0.983 (p = 0.361) | –0.088 (p = 0.482) | –3.181 (p = 0.181) |
| 10F1M:sex | –0.004 (p = 0.907) | –0.165 (*P* < 0.001) | 0.016 (p = 0.926) | 0.133 (p = 0.259) |
| 4F1M:sex | –0.003 (p = 0.939) | –0.114 (p = 0.001) | 0.002 (p = 0.990) | 0.040 (p = 0.734) |
| 1F4M:sex | –0.003 (p = 0.943) | 0.016 (p = 0.650) | –0.045 (p = 0.801) | –0.157 (p = 0.181) |
| 1F10M:sex | –0.000 (p = 0.995) | 0.008 (p = 0.822) | –0.055 (p = 0.755) | –0.148 (p = 0.207) |
| AD:sex | NA | 1.334 (p = 0.474) | 0.125 (p = 0.473) | 10.016 (p = 0.002) |
| 10F1M:AD:sex | 0.001 (p = 0.988) | 0.335 (*P* < 0.001) | –0.003 (p = 0.989) | –0.107 (p = 0.514) |
| 4F1M:AD:sex | 0.001 (p = 0.989) | 0.213 (*P* < 0.001) | 0.008 (p = 0.975) | –0.105 (p = 0.520) |
| 1F4M:AD:sex | 0.001 (p = 0.989) | –0.022 (p = 0.715) | 0.086 (p = 0.728) | 0.308 (p = 0.060) |
| 1F10M:AD:sex | 0.003 (p = 0.970) | –0.002 (p = 0.968) | 0.111 (p = 0.652) | 0.472 (p = 0.004) |
| n:sex | 0.019 (p = 0.486) | 0.094 (*P* < 0.001) | 0.119 (p = 0.340) | 0.170 (p = 0.040) |
| n:10F1M:sex | 0.002 (p = 0.958) | –0.006 (p = 0.873) | –0.027 (p = 0.879) | –0.064 (p = 0.588) |
| n:4F1M:sex | 0.002 (p = 0.957) | 0.002 (p = 0.946) | –0.005 (p = 0.978) | 0.002 (p = 0.986) |
| n:1F4M:sex | 0.006 (p = 0.879) | 0.003 (p = 0.928) | 0.030 (p = 0.867) | 0.097 (p = 0.408) |
| n:1F10M:sex | –0.003 (p = 0.944) | 0.006 (p = 0.864) | –0.011 (p = 0.952) | 0.044 (p = 0.705) |
| n:AD:sex | –0.022 (p = 0.629) | –0.181 (*P* < 0.001) | –0.186 (p = 0.286) | –0.402 (*P* < 0.001) |
| n:10F1M:AD:sex | 0.001 (p = 0.992) | 0.005 (p = 0.933) | 0.036 (p = 0.885) | 0.288 (p = 0.079) |
| n:4F1M:AD:sex | 0.001 (p = 0.992) | –0.003 (p = 0.957) | –0.003 (p = 0.989) | 0.135 (p = 0.408) |
| n:1F4M:AD:sex | –0.002 (p = 0.977) | 0.016 (p = 0.794) | –0.067 (p = 0.784) | –0.085 (p = 0.604) |
| n:1F10M:AD:sex | –0.002 (p = 0.971) | 0.001 (p = 0.985) | 0.010 (p = 0.969) | –0.100 (p = 0.541) |
| age:sex | 0.000 (p = 1.000) | 0.013 (p = 0.421) | 0.001 (p = 0.485) | 0.042 (p = 0.199) |
| 10F1M:age:sex | 0.000 (p = 0.910) | 0.002 (*P* < 0.001) | –0.000 (p = 0.920) | –0.002 (p = 0.221) |
| 4F1M:age:sex | 0.000 (p = 0.937) | 0.001 (p = 0.006) | –0.000 (p = 0.986) | –0.001 (p = 0.666) |
| 1F4M:age:sex | 0.000 (p = 0.930) | –0.000 (p = 0.773) | 0.001 (p = 0.797) | 0.002 (p = 0.166) |
| 1F10M:age:sex | 0.000 (p = 0.989) | –0.000 (p = 0.962) | 0.001 (p = 0.748) | 0.002 (p = 0.175) |
| AD:age:sex | 0.000 (p = 1.000) | –0.017 (p = 0.497) | –0.002 (p = 0.494) | –0.129 (p = 0.004) |
| 10F1M:AD:age:sex | –0.000 (p = 0.981) | –0.005 (*P* < 0.001) | 0.000 (p = 0.987) | 0.002 (p = 0.488) |
| 4F1M:AD:age:sex | –0.000 (p = 0.984) | –0.003 (*P* < 0.001) | –0.000 (p = 0.974) | 0.001 (p = 0.505) |
| 1F4M:AD:age:sex | –0.000 (p = 0.980) | 0.000 (p = 0.682) | –0.001 (p = 0.729) | –0.004 (p = 0.070) |

*Appendix 4—table 7 Continued on next page*

*Appendix 4—table 7 Continued*

|  | OASIS-3 |  | AIBL |  |
| --- | --- | --- | --- | --- |
| 1F10M:AD:age:sex | −0.000 (p = 0.973) | 0.000 (p = 0.945) | −0.002 (p = 0.653) | −0.006 (p = 0.006) |
| n:age:sex | −0.000 (p = 0.505) | −0.001 (p = 0.007) | −0.002 (p = 0.341) | −0.002 (p = 0.047) |
| n:10F1M:age:sex | −0.000 (p = 0.948) | 0.000 (p = 0.858) | 0.000 (p = 0.871) | 0.001 (p = 0.568) |
| n:4F1M:age:sex | −0.000 (p = 0.941) | −0.000 (p = 0.953) | 0.000 (p = 0.970) | 0.000 (p = 0.966) |
| n:1F4M:age:sex | −0.000 (p = 0.839) | −0.000 (p = 0.872) | −0.000 (p = 0.871) | −0.001 (p = 0.414) |
| n:1F10M:age:sex | 0.000 (p = 0.960) | −0.000 (p = 0.794) | 0.000 (p = 0.953) | −0.001 (p = 0.706) |
| n:AD:age:sex | 0.000 (p = 0.601) | 0.002 (*P* < 0.001) | 0.002 (p = 0.308) | 0.005 (*P* < 0.001) |
| n:10F1M:AD:age:sex | −0.000 (p = 0.995) | −0.000 (p = 0.940) | −0.000 (p = 0.886) | −0.004 (p = 0.084) |
| n:4F1M:AD:age:sex | −0.000 (p = 0.996) | 0.000 (p = 0.928) | 0.000 (p = 0.989) | −0.002 (p = 0.400) |
| n:1F4M:AD:age:sex | 0.000 (p = 0.955) | −0.000 (p = 0.847) | 0.001 (p = 0.782) | 0.001 (p = 0.620) |
| n:1F10M:AD:age:sex | 0.000 (p = 0.981) | 0.000 (p = 0.930) | −0.000 (p = 0.979) | 0.001 (p = 0.573) |

## Appendix 5

### Control analysis for instability around $n \approx 300$

To determine whether the deviation observed at $n = 300$ under left-skewed sampling reflected a systematic modeling issue or stochastic sampling variability, we repeated the model-fitting procedure at sample size 300 using 20 independent random seeds. For each seed, training sets were newly generated following the same sampling schemes and models were re-estimated using the standard pipeline. Model performance in the HC test set was evaluated using the same metrics as in *Figure 3* (MSLL, SMSE, EV, Rho, lower-tail HC%, upper-tail HC%, and ICC), and variability was quantified across seeds (or across ROIs for ICC). The original sampling results were retained in the main manuscript for transparency and to reflect the variability inherent in individual sampling draws.

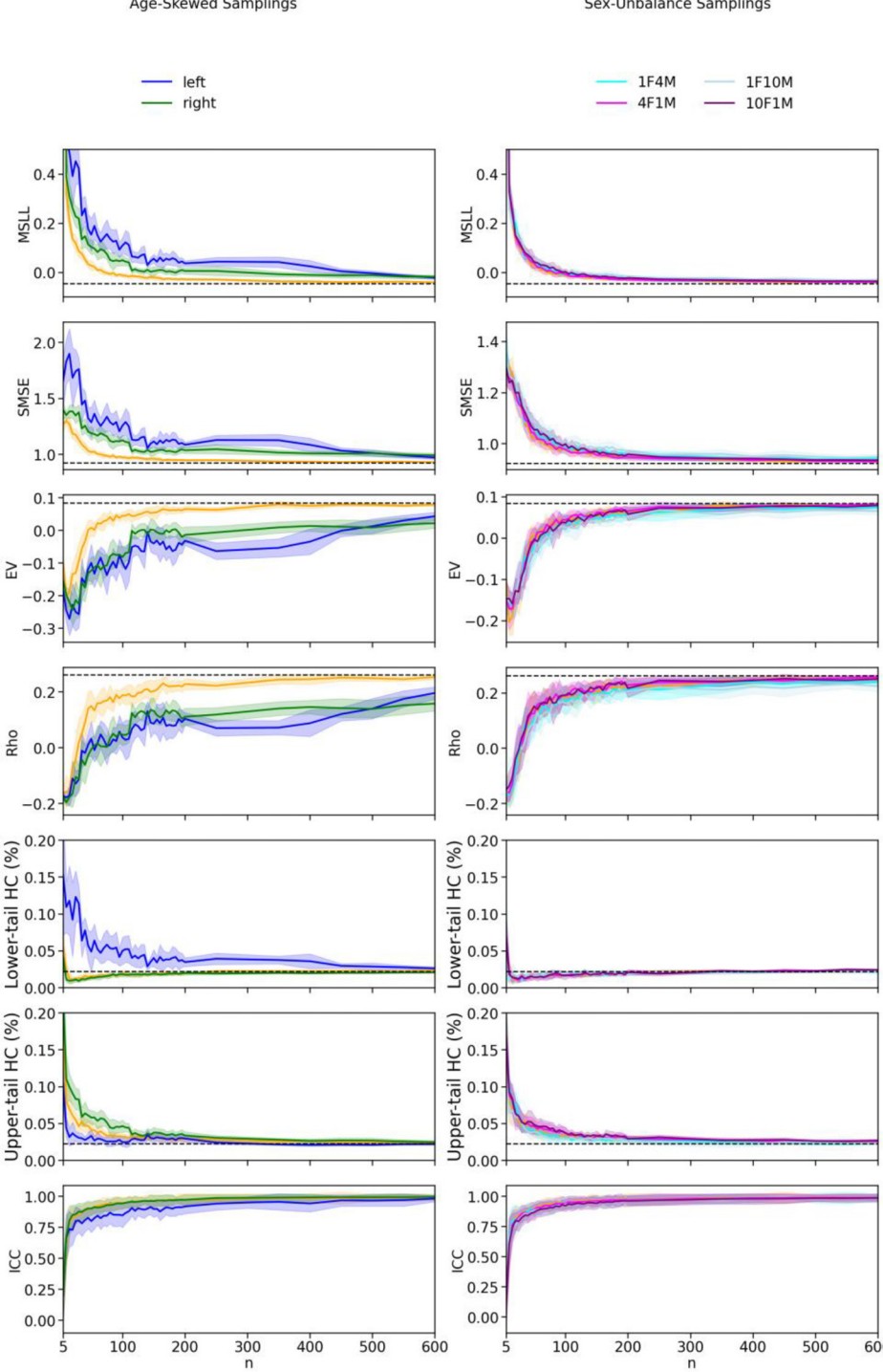

**Appendix 5—figure 1.** Evaluation of model fits in the HC test set across different sampling strategies of the training set for varying sample sizes (*n*), following re-estimation with 20 independent random seeds to assess the robustness of an apparent artifact observed at *n* = 300 in left-skewed sampling in the original analysis. Sampling strategies and metrics are identical to those shown in *Figure 3*. Solid lines represent the mean metric values across ROIs and random seeds, with shaded areas indicating the standard deviation across seeds for MSLL, SMSE, EV, Rho, the lower and upper tail HC%. For ICC, shaded areas represent variability across ROIs. Dashed lines indicate the mean metric values obtained with the full sample. The absence of a systematic deviation at *n* = 300 across random seeds indicates that the previously observed effect was driven by stochastic sampling variability rather than a stable modeling artifact.

