## [Editor Report · eLife Assessment]

This **important** manuscript evaluates how sample size and demographic balance of reference cohorts affect the reliability of normative models. The evidence supporting the conclusions is **convincing**. This work will be of interest to clinicians and scientists working with normative models.

---

## [Referee Report · Reviewer #1 (Public review)]

This is a well-designed and carefully executed study that delivers clear and actionable guidance on the sample size and representative demographic requirements for robust normative modelling in neuroimaging. The central claims are convincingly supported.

The study has multiple strengths. First, it offers a comprehensive and methodologically rigorous analysis of sample size and age distribution, supported by multiple complementary fit indices. Second, the learning-curve results are compelling and reproducible and will be of immediate utility to researchers planning normative modelling projects. Third, the study includes both replication in an independent dataset and an adaptive transfer analysis from UK Biobank, highlighting both the robustness of the results and the practical advantages of transfer learning for smaller clinical cohorts. Finally, the clinical validation effectively ties the methodological work back to real-world clinical application.

One dataset-dependent limitation worth noting concerns age-distribution coverage: the larger negative effects observed under left-skewed sampling reflect a mismatch between younger training samples and older test cohorts. Importantly, the authors explicitly quantify this effect using simulation-based coverage analyses and demonstrate that it accounts for the observed asymmetry in sampling performance. By identifying and empirically characterising this constraint, the study appropriately bounds the generalisability of its conclusions while strengthening their interpretability.

---

## [Referee Report · Reviewer #2 (Public review)]

Summary:

The authors test how sample size and demographic balance of reference cohorts affect the reliability of normative models in ageing and Alzheimer's disease. Using OASIS-3 and replicating in AIBL, they change age and sex distributions and number of samples and show that age alignment is more important than overall sample size. They also demonstrate that models adapted from a large dataset (UK Biobank) can achieve stable performance with fewer samples. The results suggest that moderately sized but demographically well-balanced cohorts can provide robust performance.

Strengths:

The study is thorough and systematic, varying sample size, age, and sex distributions in a controlled way. Results are replicated in two independent datasets with relatively large sample sizes, thereby strengthening confidence in the findings. The analyses are clearly presented and use widely applied evaluation metrics. Clinical validation (outlier detection, classification) adds relevance beyond technical benchmarks.The comparison between within-cohort training and adaptation from a large dataset is valuable for real-world applications.

The work convincingly shows that age alignment is crucial and that adapted models can reach good performance with fewer samples.

---

## [Author Response]

The following is the authors’ response to the original reviews

**Reviewer #1:**
Summary:Overall, this is a well-designed and carefully executed study that delivers clear and actionable guidance on the sample size and representative demographic requirements for robust normative modelling in neuroimaging. The central claims are convincingly supported.Strengths:The study has multiple strengths. First, it offers a comprehensive and methodologically rigorous analysis of sample size and age distribution, supported by multiple complementary fit indices. Second, the learning-curve results are compelling and reproducible and will be of immediate utility to researchers planning normative modelling projects. Third, the study includes both replication in an independent dataset and an adaptive transfer analysis from UK Biobank, highlighting both the robustness of the results and the practical advantages of transfer learning for smaller clinical cohorts. Finally, the clinical validation ties the methodological work back to clinical application.

We are grateful for the reviewer’s positive overall evaluation and for the constructive feedback, which has helped us refine and clarify the manuscript.

Weaknesses:There are two minor points for consideration:(1) Calibration of percentile estimates could be shown for the main evaluation (similar to that done in Figure 4E). Because the clinical utility of normative models often hinges on identifying individuals outside the 5th or 95th percentiles, readers would benefit from visual overlays of model-derived percentile curves on the curves from the full training data and simple reporting of the proportion of healthy controls falling outside these bounds for the main analyses (i.e., 2.1. Model fit evaluation).

We thank the reviewer for this helpful point. To address this, we implemented two complementary analyses that evaluate the accuracy of percentile estimates in the main evaluation (Section 2.1, Model fit evaluation).

(a) Percentage of healthy controls (HC) outside the extreme centiles (added to the main figure)

For each sampling strategy and sample size, we now report the proportion of healthy controls falling outside the predicted 2.5th and 97.5th percentiles, to remain consistent with the 1.96 threshold used throughout the study. Under perfect calibration, this proportion should be close to 2.5%. This metric was computed for every ROI, model run, sample size, and sampling condition. The results are now shown in the main model-fit figure alongside MSLL, EV, Rho, SMSE, and ICC, and the corresponding statistics have been added throughout. This directly quantifies how well the centile estimates capture tail behavior, which is essential for the clinical interpretation of normative deviations. See the added plots to Figure 2 and Figure 3 (see also Table 2-3 in the revised main manuscript and replication in AIBL and transfer leaning experiments in Supplementary Materials Figure S1, S10-11, S18-19, S2829, Table S1-2, S5-6, S9-10).

(b) Centile curve overlays (added to the Supplementary Figures)

To visually demonstrate calibration, we now include additional overlays of model-derived percentile curves against those obtained using the full training set. These are shown for key ROIs, multiple sample sizes and different sampling strategies in Supplementary Materials (Figure S9 and S27). These overlays illustrate where centile estimation diverges, particularly at age extremes.

Together, these additions provide both quantitative and qualitative evidence of percentile calibration across sampling regimes and sample sizes.

(2) The larger negative effect of left-skewed sampling likely reflects a mismatch between the younger training set and the older test set; accounting explicitly for this mismatch would make the conclusions more generalizable.

We agree with the reviewer that the large negative effect of left-skewed training reflects a mismatch between the training and test age distributions.

To characterize the expected age distributions produced by each sampling strategy, we simulated the procedures used in the main analyses by repeatedly drawing training samples under all sampling conditions (representative, left-skewed, right-skewed, and the predefined sex-ratio settings). Simulations were performed at a fixed sample size (n = 200), generating 1000 samples per condition, and the resulting age distributions were summarized separately for males and females (Supplementary Materials section 5.1). These simulated distributions show that left-skewed sampling produces a more pronounced shift toward younger ages than the corresponding shift toward older ages under rightskewed sampling, particularly in OASIS-3, with smaller differences observed in AIBL (Tables S14– S15).

To further quantify how these sampling-induced age profiles align with the empirical age structure of the test cohorts, we computed an age-bin coverage metric based on distribution intersection. Age was discretized into 20 quantile-based bins using the full training set of each dataset (OASIS-3 and AIBL) as reference.

For each sampling strategy (Representative, Left-skewed, Right-skewed), sample size, and dataset, we generated 1000 independent training samples using the same sampling procedures as in the main analyses. For each sampled training set, age-bin count distributions were computed and compared to the corresponding HC test-set age-bin counts.

Coverage was defined as:\begin{document}$$\displaystyle  =\frac{\sum_{i} \min \left(n_{\text {tain }}(i), n_{\text {tast }}(i)\right)}{\sum_{i} n_{\text {tast }}(i)}$$\end{document}

where, 𝑖 indexes age bins, 𝑛_train_ and 𝑛_test_ are the numbers of individuals in bin i in the sampled training set and HC test set, respectively. This metric quantifies the fraction of the test-set age distribution that is “covered” by the sampled training set and ranges from 0 (no test-set ages covered) to 1 (complete coverage of the test-set age distribution). For each condition, the mean and standard deviation of the coverage across repetitions were computed.

We show that under left-skewed sampling, age coverage remains markedly reduced across all sample sizes in OASIS-3 in comparison with AIBL dataset (see Figures S37). This suggests that the poorer performance observed with left-skewed training may stem from a reduced coverage of the test age range. We added the following in the Discussion (page 27):

“The left-skewed sampling had overall a greater effect than right-skewed sampling in both model evaluation and clinical validation, likely due to (1) the dataset’s original bias toward older individuals, making younger-skewed samples less representative, and (2) the older age structure of the AD population, which exacerbates mismatch when younger HC are used to calibrate models in the clinical population. This asymmetry is also reflected in the coverage analysis, where left-skewed sampling resulted in poorer age coverage of the target population at the same sample size (Supplementary Materials section 5.4.)”

**Reviewer #2:**
Summary:The authors test how sample size and demographic balance of reference cohorts affect the reliability of normative models in ageing and Alzheimer's disease. Using OASIS-3 and replicating in AIBL, they change age and sex distributions and number of samples and show that age alignment is more important than overall sample size. They also demonstrate that models adapted from a large dataset (UK Biobank) can achieve stable performance with fewer samples. The results suggest that moderately sized but demographically well-balanced cohorts can provide robust performance.Strengths:The study is thorough and systematic, varying sample size, age, and sex distributions in a controlled way. Results are replicated in two independent datasets with relatively large sample sizes, thereby strengthening confidence in the findings. The analyses are clearly presented and use widely applied evaluation metrics. Clinical validation (outlier detection, classification) adds relevance beyond technical benchmarks. The comparison between within-cohort training and adaptation from a large dataset is valuable for real-world applications.The work convincingly shows that age alignment is crucial and that adapted models can reach good performance with fewer samples. However, some dataset-specific patterns (noted above) should be acknowledged more directly, and the practical guidance could be sharper.

We are grateful for the reviewer’s positive overall evaluation and for the constructive comments that guided our revisions strengthened the manuscript.

Weaknesses:The paper uses a simple regression framework, which is understandable for scalability, but limits generalization to multi-site settings where a hierarchical approach could better account for site differences. This limitation is acknowledged; a brief sensitivity analysis (or a clearer discussion) would help readers weigh trade-offs.

We thank the reviewer for this insightful point. We agree that hierarchical Bayesian regression provides clear advantages in multi-site settings, particularly when site-level variability is substantial or when federated learning is required. In our case, both OASIS-3 and AIBL include only a small number of sites, and the primary aim of the study was to isolate the effects of sample size and covariate composition rather than to model site-related structure. For these reasons, implementing HBR was beyond the scope of the present work, but we fully acknowledge its relevance for studies with larger or more heterogeneous site configurations. To clarify this distinction, we added a dedicated paragraph in the Discussion (page 28) that situates warped BLR and HBR within different data scenarios and outlines the circumstances under which each approach is preferable.

“From a methodological perspective, the choice between warped BLR and HBR should primarily be guided by the structure of site effects and by computational constraints. HBR explicitly models sitelevel variation through hierarchical random effects, enabling information sharing across sites and supporting federated-learning implementations in which site-specific updates can be combined without sharing raw data (Bayer et al., 2022; Kia et al., 2021; Maccioni et al., 2025). This structure provides more stable estimates when site-specific sample sizes are small or acquisition differences are substantial. In contrast, wrapped BLR treats site as a fixed-effect covariate when site adjustment is required and does not implement hierarchical pooling, but offers simpler inference and substantially lower computational cost while accommodating non-Gaussian data distributions through the warping transformation (C. J. Fraza et al., 2021). These properties make wrapped BLR practical in settings where site heterogeneity is limited or adequately controlled, whereas HBR may be preferable in strongly multisite contexts or when federated learning is required for privacy-preserving data integration.”

Other than that, there are some points that are not fully explained in the paper:(1) The replication in AIBL does not fully match the OASIS results. In AIBL, left-skewed age sampling converges with other strategies as sample size grows, unlike in OASIS. This suggests that skew effects depend on where variability lies across the age span.Recommendation: Replication differences across datasets (age skew):In OASIS, left-skewed (younger-heavy) training harms performance and does not fully recover with more data; in AIBL, performance under left-skew appears to converge toward the other conditions as training size grows. Given AIBL's smaller size and older age range, please explain this discrepancy. Does this imply that the effect of skew depends on where biological variability is highest across the age span (e.g., more variability from ~45-60 in OASIS vs {greater than or equal to}60 in AIBL), rather than on "skew" per se? If so, the paper should say explicitly that skewness must be interpreted relative to the age-variability profile of the target population, not just counts.

We thank the reviewer for this thoughtful comment. To examine whether differences in age-related variability could explain the replication patterns, we quantified how regional variance changed with age by computing age-binned variance profiles in the HC training sets of OASIS-3 and AIBL. Age was discretized into 10 quantile-based bins for each dataset separately. For each ROI and each age bin, we calculated the sample variance of the ROI values within that bin. The bin center was defined as the mean age of individuals in the corresponding bin. We then summarized variance across ROIs by computing, for each age bin, the median variance and its interquartile range (25th–75th percentile). These summary profiles (median and IQR across ROIs as a function of bin-centered age) are shown in Author response image 1. As shown in this plot, OASIS-3 and AIBL display comparable levels of variance across their respective age ranges, and the profiles do not suggest pronounced shifts in variability that would account for the divergent behavior of the left-skewed models.

**Author response image 1. sa3fig1:** Median ROI variance across age bins for OASIS-3 and AIBL. Shaded areas represent variability across regions within each age bin.

Instead, the coverage analysis recommended by the reviewer in comment #5 and introduced in our response to Reviewer 1, comment #2 indicates that the replication differences between OASIS-3 and AIBL are primarily driven by the age coverage of the sampled training sets relative to the test cohorts. In AIBL, which has a narrower and predominantly older age range, left-skewed sampling shows slightly lower coverage than right-skewed sampling, but coverage increases steadily with sample size, and the strategies converge as n grows. In contrast, OASIS-3 spans a broader lifespan and is itself skewed toward older ages; under left-skewed sampling, coverage of the test-set age range increases more slowly and remains comparatively lower even at large n. This slower recovery of age coverage explains why leftskewed performance does not recover in OASIS-3 and why the discrepancies between left- and rightskewed sampling are more pronounced in this dataset. The corresponding age-coverage curves are reported in Supplementary Figures S37.

Furthermore, this difference is also reflected in the expected age distributions obtained from repeated simulations of the sampling procedures (Supplementary Materials section 5.1. Tables S14–S15), where left-skewed sampling induces a larger shift toward younger ages than right-skewed sampling induces toward older ages, especially in OASIS-3, with smaller differences observed in AIBL.

For more details on both analyses see also our response to Reviewer 1, comment #2.

(2) Sex imbalance effects are difficult to interpret, since sex is included only as a fixed effect, and residual age differences may drive some errors.Recommendation: Sex effects may be confounded with age:Because sex is treated only as a fixed effect, it is unclear whether errors under sex-imbalance scenarios partly reflect residual age differences between female and male subsets. Please report (or control for) age distributions within each sex-imbalance condition, and clarify whether the observed error changes are truly attributable to sex composition rather than age composition.

To address the concern that sex-imbalance effects could be driven by residual age differences we now explicitly report the age distributions by sex for the original training and test datasets, as well as the expected age distributions induced by each sampling condition, obtained by repeated simulation of the sampling procedure (Supplementary Materials section 5.1, Tables S13-15). Table S13 shows very similar distributions of age for HC train and test sets across sexes within each dataset. Tables S14–S15 further show that, within each sampling strategy, the age distributions of females and males are highly similar, including under sex-imbalanced conditions. These summaries confirm that the sampling procedures do not introduce systematic age-structure differences between sexes.

In addition, we extended the statistical models for tOC and MSE to explicitly include age, sex, and all higher-order interactions with the diagnosis, sample size, and sex-ratio sampling (Supplementary Materials section 5.2., Tables S17 for direct training, and S19 for transferred models). For completion we also included age and sex for age samplings models (Supplementary Tables S16 for direct training, S18 for transferred models). These analyses revealed no significant main effects of age under seximbalanced sampling and only very small effect sizes in isolated higher-order interactions. Together, these results indicate that age did not introduce residual confounding in our analyses.

We now report in the Results section (page 15) the following:

“Supplementary analysis (Tables S17,19) also showed that main effect of age was not significant for either MSE or tOC, and no significant age × sex-ratio interactions were observed. While some higherorder interactions involving age, diagnosis, and sex-ratio reached statistical significance, all associated effect sizes were very small and inconsistent across outcomes, indicating that the observed error changes are not driven by residual age confounding.”

And in the Methods section (page 36):

“Age distributions were summarized separately for males and females in the original training and test sets (Supplementary Table S13) and the expected age distributions resulting from the skewed-age sampling and the sex-imbalance sampling procedures were obtained by repeated simulations at a fixed sample size and are reported in Supplementary Tables S14–S15.”

(3) In Figure 3, performance drops around n≈300 across conditions. This consistent pattern raises the question of sensitivity to individual samples or sub-sampling strategy.Recommendation: Instability around n ≈ 300 (Figure 3):Several panels show a consistent dip in performance near n=300. What drives this? Is the model sensitive to particular individuals being included/excluded at that size, or does it reflect an interaction with the binning/selection scheme? A brief ablation (e.g., alternative sub-sampling seeds or bins) would help rule out artefacts.

We thank the reviewer for highlighting this point. To assess whether the observed dip at n=300 reflected sensitivity to the specific individuals selected or to the sub-sampling scheme, we re-ran the analysis at n = 300 using 20 independent random seeds (Supplementary Materials sections 5.3.). This ablation showed no systematic decrease in performance across repetitions, indicating that the original effect was driven by stochastic sampling variability rather than a stable model instability or binning interaction. We now report this control analysis in the Supplementary Materials (Figure S36). We have clarified this point in the Results page 10:

“A consistent dip in performance was observed around n = 300 for the left-skewed sampling condition in the original analysis (Figure 3). To assess whether this reflected sensitivity to the specific subsampling or stochastic sampling variability, we repeated the analysis for this specific sample using 20 independent random seeds (Figure S36); the absence of a consistent effect across repetitions indicates that the original pattern was driven by sampling variability rather than a systematic model artifact.”

(4) The total outlier count (tOC) analysis is interesting but hard to generalize. For example, in AIBL, left-skew sometimes performs slightly better despite a weaker model fit. Clearer guidance on how to weigh model fit versus outlier detection would strengthen the practical message.Recommendation: Interpreting total outlier count (tOC):The tOC findings are interesting but hard to operationalize. In AIBL, even for n>40, left-skewed training sometimes yields slightly better tOC discrimination and other strategies plateau. Does this mean that a better model fit on the reference cohort does not necessarily produce better outlier-based case separation? Please add a short practical rule-set: e.g., when optimizing for deviation mapping/outlier detection, prioritize coverage of the patient-relevant age band over global fit metrics; report both fit and tOC sensitivity to training-set age coverage.

We thank the reviewer for this important point. Apparent improvements in tOC-based separation under left-skewed training should not be interpreted as indicating a better model or superior deviation mapping. In particular, in AIBL, left-skew can sometimes yield slightly larger group differences in tOC despite weaker overall model fit. This reflects an inflation of deviation magnitude in AD rather than improved separation per se. Crucially, relative ranking between HC and AD remains preserved across sampling strategies, as shown by the classification analysis in the main manuscript (Figure 5C), indicating that enhanced tOC contrast under left-skew does not translate into improved case discrimination. Instead, it reflects a systematic shift in deviation scale due to age-mismatched training.

We now clarify this distinction in the Discussion of the main manuscript on page 26:

“Importantly, apparent increases in HC–AD separation in total outlier count should not be interpreted as evidence of superior model quality. Age-mismatched training can rescale deviation magnitudes and inflate tOC in specific subgroups without improving true case–control separability, as shown by classification task (Figure 5C). Model fit metrics and outlier-based measures, therefore capture complementary but distinct aspects of normative model behavior and should be interpreted jointly rather than in isolation.”

(5) The suggested plateau at n≈200 seems context dependent. It may be better to frame sample size targets in relation to coverage across age bins rather than as an absolute number.Recommendation: "n≈200" as a plateau is context-dependent:The suggested threshold for stable fits (about 200 people) likely depends on how variable the brain features are across the covered ages. Rather than an absolute number, consider reporting a coverageaware target, such as a minimum per-age-bin coverage or an effective sample size relative to the age range. This would make the guidance transferable to cohorts with different age spans.

We agree that the observed performance plateau around n≈200 is context dependent and may shift with the covered age range, anatomical variability, and feature of interest. In the present study, this stabilization was evaluated within the specific datasets and age spans considered and extending it to broader lifespan or different biological contexts will require dedicated future work.

To clarify this point, we added an explicit age-coverage analysis in the Supplementary Materials (section 5.4.) as introduced in response to reviewer 1 on comment #2. This analysis shows that, under representative sampling, the point at which age coverage becomes complete closely coincides with the saturation of model fit and stability metrics. At the same time, we note that normative models operate in continuous covariate space, such that reliable interpolation can still be achieved even when intermediate age ranges are less densely sampled, provided that surrounding age ranges are sufficiently represented. This makes rigid minimum per-bin requirements difficult to define in a generalizable way.

Rather than proposing a universal sample-size threshold, we now emphasize that both learning-curve analyses and age-coverage assessments offer a more transferable way to identify when performance approaches saturation for a given dataset. This clarification is now included in the Discussion on page 25:

“This is further supported by the coverage analysis reported in the Supplementary Materials (section 5.4), which shows that under representative sampling, the point of full age coverage closely coincides with the saturation of model fit and stability metrics. Rather than proposing a universal sample size threshold, we therefore encourage readers to perform learning-curve analyses, complemented by age coverage assessments, in their own datasets to empirically assess when performance approaches saturation for their specific age range and population.”

And we also address it in the limitations page 29:

“In addition, the observed stabilization of model performance around 200–300 participants was evaluated within the specific age ranges and cohorts examined here and may shift in broader lifespan settings or in populations with different sources of biological variability.”

(5) Minor inconsistency in training-set size:The manuscript mentions 691 in Methods, but the figures/scripts label is 692. Please correct for consistency.

Thank you for pointing out this inconsistency, the error in the methods section has been corrected.